# Determination of the TROPOMI-SWIR instrument spectral response function

Richard M. van Hees[1], Paul J. J. Tol[1], Sidney Cadot[1,2], Matthijs Krijger[1,3], Stefan T. Persijn[5], Tim A. van Kempen[1], Ralph Snel[1,4], Ilse Aben[1], and Ruud W. M. Hoogeveen[1]

[1]SRON Netherlands Institute for Space Research, Utrecht, the Netherlands
[2]Jigsaw B.V., Delft, the Netherlands
[3]Earth Space Solutions, Utrecht, the Netherlands
[4]Science and Technology B.V., Delft, the Netherlands
[5]VSL Dutch Metrology Institute, Delft, the Netherlands

**Correspondence:** Dr Richard van Hees (r.m.van.hees@sron.nl)

**Abstract.** The Tropospheric Monitoring Instrument (TROPOMI) is the single instrument on board of the ESA Copernicus Sentinel-5 Precursor satellite. TROPOMI is a nadir-viewing imaging spectrometer with bands in the ultraviolet and visible, the near infrared and the short-wave infrared (SWIR). An accurate instrument spectral response function (ISRF) is required in the SWIR band where absorption lines of CO, methane and water vapor overlap. In this paper, we report on the determination of the TROPOMI-SWIR ISRF during an extensive on-ground calibration campaign. Measurements are taken with a monochromatic light source scanning the whole detector, using the spectrometer itself to determine the light intensity and wavelength. The accuracy of the resulting ISRF calibration key data is well within the requirement for trace-gas retrievals. Long-term in-flight monitoring of SWIR ISRF is achieved using five on-board diode lasers.

## 1   Introduction

The Tropospheric Monitoring Instrument (TROPOMI) is the single payload of the Copernicus Sentinel-5 Precursor (S5P) satellite mission. The instrument maps the Earth's atmosphere using two spectrometer modules behind a common telescope, one covering the ultraviolet/visible (270–495 nm) and near-infrared (675–775 nm), and the other covering the short-wave infrared (SWIR) spectral range 2305–2385 nm. The spectral resolution of the SWIR spectrometer is about 0.25 nm with a spectral sampling interval of typically 0.1 nm. The TROPOMI instrument measures sunlight reflected by the surface and atmosphere of the Earth via the radiance port. Direct sunlight is measured via the irradiance port and internal diffuser for calibration purposes (Veefkind et al., 2012). The SWIR spectrometer (developed by SSTL, United Kingdom) consists of a slit, collimator mirror optics, an immersed grating (developed by SRON, van Amerongen et al., 2012), anamorphic prism and camera optics consisting of multiple lenses, and a HgCdTe detector (developed by Sofradir, France). The detector has 1000 columns in the spectral dimension and 256 rows in the spatial dimension of which about 975 columns and 217 rows are illuminated.

The TROPOMI-SWIR band is used for the retrieval of the atmospheric trace gases carbonmonoxide, methane and water vapor. Simulations have shown that in particular the methane retrieval is very sensitive to errors in the instrument spectral response function (ISRF or instrument line shape). As a result, the requirement on the ISRF is formulated that it should be known with an accuracy of 1% of its maximum where the ISRF is greater than 1% of its maximum (Hu et al., 2016). To reach the required accuracy, data have been measured with high spectral resolution using a scanning monochromatic light source covering the SWIR band. For a comprehensive overview of various approaches applied to determine the ISRF for relevant past and future space-borne missions we refer to Sun et al. (2017). In summary, for the pioneering mission GOME on ESA's ERS-1 satellite and for the SCIAMACHY instrument on board ESA ENVISAT, no high-resolution ISRF was measured at all on ground. Information was derived from an on-board spectral light source with discrete line emissions (Schrijver et al., 2009). For the later OMI instrument on board NASA's EOS Aura satellite a white light source followed by a high-resolution monochromator was used to create a comb of narrow spectral lines (Dobber et al., 2006). A similar approach was followed for the other three bands of TROPOMI (Kleipool et al., 2018). The NASA OCO instruments had the ISRF measured with a monochromatic line source (Day et al., 2011; Sun et al., 2017).

Often no distinction is made between a *spread* function and a *response* function. In this paper, the functions are defined as follows: a spread function maps an object to image space, which involves many detector pixels; a response function maps an image to object space, which is a property of a given detector pixel. The instrument spectral *spread* function (ISSF) is measured simply by illuminating the spectrometer slit homogeneously with a monochromatic source and taking a detector image (frame). In the spectral dimension, about 5 pixels have significant signal, as expected from the spectral oversampling. This is the spread function of the instrument for this wavelength. In Fig. 1b it is shown as a red cross section. When the wavelength is scanned in small steps over a set of frames, the signal in those frames for a given pixel forms an ISRF, with an arbitrarily fine sampling. This is shown as a green cross section in Fig. 1c. There is an infinite number of ISSFs (one at each wavelength) and a finite number of ISRFs (one for each pixel). The ISSF consists of one sample from each ISRF of a few neighboring pixels on a row. If the ISRF varies negligibly between these pixels, the ISSF is a sparsely sampled version of this ISRF. Figure 1a shows that the samples taken with increasing column index are ISRF points from the right side of the peak to the left side: the ISSF samples a mirrored ISRF, indicated by the light-green line in Fig. 1b.

By design, the ISRF and dispersion of the TROPOMI instrument should vary only smoothly in the spectral and spatial dimension. This assumption will be validated in this study by determining the local ISRF for many pixels of the SWIR detector. The assumption is also used in the data analysis to interpolate the ISRF to pixels for which there are no reliable measurements and to reduce the effect of outliers.

The ISRF is determined from measurements using the radiance port and the irradiance port. Although differences between both data sets are not expected, they could not be ruled out. Light entering via the irradiance port follows a different path, via a diffuser.

The ISRF measurements are part of the on-ground calibration campaign performed at the Centre Spatial de Liège (CSL) in Belgium (Kleipool et al., 2018). The limited time slot did not allow to perform the measurements with a very accurate wavelength meter and wait for a stable laser at a given wavelength. Instead, the laser scanned slowly during data taking and

each frame is treated as the measurement of an ISSF per row, where the column index of the fitted center is used as a wavelength label for the row data. The ISRF of a pixel, based on data from a set of frames, is then the normalized signal as a function of wavelength in pixel units (non-integer values). It should not be confused with the ISSF, which has basically the same horizontal scale but then in integer pixel units. Only at the end of the algorithm is the ISRF of a given pixel converted to a function of wavelength in nm, using the wavelength assignment derived from an independent wavelength calibration measurement. This results in the ISRF calibration key data (CKD) which are used in trace-gas retrievals.

The measurements used for the ISRF characterization are presented in Sect. 2. A description of the method and algorithm used to derive the ISRF for all illuminated pixels is presented in Sect. 3. Details of the algorithm are discussed in Sect. 4, as well as the ISRF results based on the on-ground calibration measurements. Comparison of the ISRF as determined from irradiance, radiance and on-board diode-laser measurements are discussed and the choice for the CKD is motivated. The validation of the SWIR ISRF is also part of the discussion (Sect. 4). The in-flight monitoring of the ISRF is briefly described in Sect. 5. This is followed by the conclusions in Sect. 6.

## 2 Calibration measurements

### 2.1 Measurements with the external laser

The light source employed in the ISRF characterization measurements is a 2 W continuous-wave optical parametric oscillator (OPO), custom-built by VSL (Delft, the Netherlands). The OPO is pumped by a single-frequency distributed feedback (DFB) fiber laser operating at 1064 nm which is amplified to 10 W by an ytterbium fiber amplifier. The OPO wavelength is set coarsely between 2290 nm and 2390 nm by manually setting the temperature of the periodically poled lithium niobate crystal and rotating the etalon mounted on a galvo. The wavelength is scanned continuously over a range of about 2 nm by applying a changing piezo voltage to the fiber laser and simultaneously changing the crystal temperature with a predetermined dependence on the piezo voltage. The setup for the radiance and irradiance measurements is shown in Fig. 2.

During the radiance measurements, the power entering the instrument has been reduced with a neutral density filter, just after the OPO, to avoid saturation of the SWIR detector. To suppress speckle patterns on the detector, the light is sent to an integrating sphere via a high-speed spinning mirror with a small angle between the rotation axis and the normal. The light exits the integrating sphere and is collimated with a field stop and an off-axis parabolic mirror. The beam corresponds to a swath-angle coverage of $1.1°$, illuminating approximately 2 pixels in the swath direction. To scan all swath angles, the instrument is mounted on a cradle. The automated wavelength scans were repeated 109 times to cover the range of $108°$ around nadir. At each swath position, the wavelength was scanned in the opposite direction. The detector covers the wavelength range 2300–2389 nm, but due to time constraints, measurements have been collected of wavelengths between 2304 nm and 2386 nm, which covers the performance range 2305–2385 nm.

The field stop and parabolic mirror were replaced by a set of collimating lenses during the irradiance measurements, in which the whole swath is illuminated at once via the on-board solar diffuser. As there was no need to repeat measurements at different

swath angles, each automated wavelength scan was repeated with increasing and decreasing wavelength scan direction. The irradiance data covers the full wavelength range.

Each automated wavelength scan of about 2 nm, or 20 spectral pixels, took about 165 seconds. The data is collected at 10 Hz, yielding 1650 ISSF samples in total and about 80 samples per pixel. The laser scan speed was not constant, despite the small adjustments of the piezo voltage during the wavelength scan. Due to the large number of samples taken, this has no negative impact on the ISRF determination. During the measurements, a dedicated quick-look facility was available to monitor the conditions of the instrument and to show the measured signals in real-time. A wavelength meter was used by the operators to set the start wavelength of each automated wavelength scan. The operators kept a log during the measurement campaign to reported on issues during the measurements. Overall, more problems were reported during the radiance measurements, mostly due to instability of the source due to drift and mode hopping. Detailed data analysis was performed after the measurement campaign was finalized, denying the possibility to improve or redo certain measurements.

## 2.2 Measurements with the on-board diode-lasers

TROPOMI's on-board calibration system includes five distributed feedback lasers (Nanoplus, Germany) to monitor stray light and ISRF (Veefkind et al., 2012). The wavelength is scanned by tuning the temperature of each laser using the thermo-electric cooler integrated in the laser housing. These diode lasers are tuneable over a range of 7 nm (about 70 pixels), but due to operational constraints monitoring is restricted to 0.6 nm (about 6 pixels). Analysis revealed that the lasers are very stable and can perform very precise wavelength scans are scanned in 430 seconds, taking over 700 samples per pixel (at 10 Hz). The distribution of the five lasers is listed by their operational wavelength (at the center of the scan) and corresponding pixel (between brackets): 2311.8 nm (154), 2328.2 nm (341), 2340.0 nm (471), 2357.5 nm (659) and 2372.2 nm (813). The lasers illuminate the SWIR spectrometer via a dedicated diffuser. The speckle in the laser signal can be suppressed by oscillation of the diffuser around the nominal angle. However, as the diffuser mechanism is a life-limited item, only during the on-ground calibration campaign and during the in-flight commissioning phase, measurements will be performed with a moving diffuser.

## 2.3 Data preparation

The three measurement data sets (irradiance, radiance and diode-lasers) are corrected for detector features such as memory and pixel response non-uniformity (PRNU), using the operational level-1b processor developed by KNMI (Kleipool et al., 2018). Changes of the background signal are removed by dedicated background measurements, which include offset, detector dark current and thermal background signal. These measurements are performed regularly during the measurements with the external laser, by blocking its signal with a shutter (Fig.2). For the on-board lasers the background signal is measured before and after each wavelength scan. A non-linearity correction is not implemented in the operational processor. It is also not needed for the ISRF characterization, as the error is small: detector non-linearity was measured to be about 0.1% to 0.2% (Hoogeveen et al., 2013). In irradiance measurements, where the light is imaged as a vertical line, the ISRF at one row could be affected by stray light from other rows. Hence, these data are corrected for stray light using the operational processor. In radiance

measurements, where the light is imaged as a spot, there is no stray light from other rows and the stray-light correction is not applied. The diode laser measurements are also not corrected for stray light, as they are intended to monitor stability.

Calibration key data for pixel quality have been derived from on-ground measurements. The pixel quality is based on several tests to identify pixels with too high dark current or noise. Most of the pixels with low quality exhibit high noise. About 260 pixels with a very low pixel quality ("bad pixels") are rejected from the analysis.

During data analysis it was found that a significant amount of irradiance measurements had to be rejected from the analysis due to detector saturation, as a result of an unexpectedly large variation of the laser intensity. From the radiance measurements partly illuminated rows were rejected.

## 3 Methodology

In this section, the method is described to determine the TROPOMI-SWIR ISRF from the on-ground measurements presented in the previous section. A general description is given first, followed by the details of the method.

The first step in the analysis is to obtain the wavelength and intensity of the signal for each frame measured in each wavelength scan. As a first approximation, the ISSF shape corresponds to an image of the slit on the detector with the optics blurring the image. The mathematical model for the ISSF at this stage is a convolution of a normal distribution and a uniform distribution. Non-linear least squares minimalization is used to fit this function to the data. As a result, a wavelength can be assigned to each measurement and the signals can be normalized by the fitted signal intensity. The wavelength is expressed as a non-integer column distance. The wavelength assignment in nm is performed in the final step of the ISRF determination.

The shape of an ISRF at location $(r, c)$, where $r$ is along the spatial dimension and $c$ along the spectral dimension, is given by the normalized signal measured by detector pixel $(r, c)$ as a function of wavelength. Typical examples of ISRFs obtained from resp. irradiance, radiance and diode-laser measurements are presented in Figs. 3–5, taken across the SWIR detector from top left to bottom right. The locations vary slightly between the measurement types to avoid measurements affected by strong saturation effects or laser instabilities. In the upper panels, the ISRF data points are shown as black dots. The irregular distances between the dots clearly show that the scan speed of the external laser is not constant, as explained in Sect. 2.1. This has no negative effect on the curve fitted through the data points. Furthermore, the fit is also quite robust against single outliers and missing data points. The peak width due to the projection of the slit on the detector is constant when expressed as a wavelength interval, but expressed as a column distance it decreases towards larger columns (longer wavelengths), because the spectral dispersion changes. The peak height increases to keep an integrated area of 1. The ISRF in the panels $(a_1)$ and $(a_2)$ are clearly skewed, while the other three are, by eye, symmetrical. The signal-to-noise in all three measurements is sufficient to determine the signal accurately up to 4.5 pixels from the center. In the calibration key data the ISRF will be defined in this range only. The ISRF outside this range will be set to zero. Any remaining signal is considered as stray light, in line with the stray-light correction algorithm (Tol et al., 2018).

The one-step approach described so far would actually result in ISRF fit residuals much larger than shown in Figs. 3–5. The residuals would show systematic oscillations with a period of one pixel. These occur when a simplified model is applied to a

poorly sampled ISSF, leading to errors depending on whether the peak is at a measured point or between two measured points (see Fig. 1b). In addition, the residuals of an asymmetric ISRF would show significant left-right differences, mostly negative residuals to the left and positive residuals to the right, when a symmetric model is used in the ISSF fit.

A TROPOMI-SWIR ISSF measurement has typically 5 spectral pixels with sufficient signal-to-noise, not enough to fit all parameters of the complete ISRF model including a description for skewness and tails. However, the ISRF derived with the approach so far is closer to the true ISRF than the simplified model used for the first ISSF fit. Using the mirrored shape of the ISRF at a given detector location as the shape of the ISSF at that location, the ISSF is fitted again to yield an updated wavelength and intensity as input to the determination of the ISRF (Sect. 3.2). The residuals of the resulting ISRF fit are much smaller. This procedure should be repeated until the residuals are no longer improving.

So far, a general approach is presented to determine high-resolution ISRFs for a spectrometer that measures the ISSF with only a few spectral pixels, using a scanning monochromatic light source. Essential is to use the instrument itself to determine the wavelength and intensity of the light. This method enables the necessary measurements to be taken within reasonable time (days) even for detectors with more than 100,000 pixels. Next, a model is defined for the TROPOMI-SWIR ISRF and a practical implementation is shown of the iterative approach.

## 3.1 ISRF model

The mathematical model for the ISRF should be flexible enough to represent the range of ISRF shapes adequately with the smallest number of parameters. The TROPOMI-SWIR ISRF is modeled by the weighted sum of functions for the peak and the tails. The peak function is a skew-normal distribution convolved with a uniform distribution. This corresponds to a (possibly asymmetric) image of the slit on the detector, with the optics blurring the image. Beirle et al. (2017) use an asymmetric version of the exponential power distribution ('Super Gaussian'). This function is computationally less demanding and suitable for general ISRF simulations, but in our case the fit residuals are larger than with the convolution above. For the tails a function is needed with an adjustable tail weight. A suitable function is found to be the Pearson type VII distribution, which is a generalization of the Gauss and Lorentz distributions. It can represent the wings of SWIR spectral stray light satisfactorily (see Tol et al., 2018, Fig. 9).

The normal distribution with mean 0 and standard deviation $\sigma$ is given by

$$\mathcal{G}(c;\sigma) = \frac{1}{\sigma\sqrt{2\pi}} \exp\left(-\frac{c^2}{2\sigma^2}\right). \tag{1}$$

The skew-normal distribution is a generalization including an extra skewness parameter $s$:

$$\mathcal{N}_1(c;\sigma,s) = \left[1 + \text{erf}\left(\frac{s\,c}{\sigma\sqrt{2}}\right)\right] \mathcal{G}(c;\sigma). \tag{2}$$

This function has mean $\sigma\,\delta$ and standard deviation

$$d = \sigma\sqrt{1-\delta^2}, \tag{3}$$

with

$$\delta = \frac{\sqrt{2}\,s}{\sqrt{\pi(1+s^2)}}.$$

(4)

To interpret fitting results more easily, the skew-normal distribution is written in terms of $d$ instead of $\sigma$ and a parameter $c_0$ is included for the mean:

$$\mathcal{N}_2(c;d,s,c_0) = \mathcal{N}_1\!\left(c - c_0 + \frac{d\,\delta}{\sqrt{1-\delta^2}}; \frac{d}{\sqrt{1-\delta^2}}, s\right)$$

(5)

$$= \frac{\sqrt{\frac{1}{2}\pi + \frac{1}{1+s^2} - 1}}{\pi d}\left[1 + \mathrm{erf}\left(\frac{s\,\xi_0}{\sqrt{2}}\right)\right]\exp\left(-\tfrac{1}{2}\xi_0^2\right),$$

(6)

with

$$\xi_0 = \frac{\sqrt{1-\delta^2}}{d}(c - c_0) + \delta.$$

(7)

Peak function $\mathcal{S}$ is the convolution of skew-normal distribution $\mathcal{N}_2(c;d,s,c_0)$ and a uniform distribution with mean 0 and full
width $w$ (the 'block width'):

$$\mathcal{S}(c;d,s,w,c_0) = \frac{1}{w}\int\limits_{c-w/2}^{c+w/2} \mathcal{N}_2(u;d,s,c_0)\,\mathrm{d}u.$$

(8)

This can be written as

$$\mathcal{S}(c;d,s,w,c_0) = \frac{\mathrm{erf}\left(\xi_+/\sqrt{2}\right) - \mathrm{erf}\left(\xi_-/\sqrt{2}\right)}{2w}$$
$$- 2\frac{T(\xi_+,s) - T(\xi_-,s)}{w}$$

(9)

with

$$\xi_\pm = \frac{\sqrt{1-\delta^2}}{d}(c - c_0 \pm \tfrac{w}{2}) + \delta$$

(10)

and using Owen's T function (Patefield and Tandy, 2000)

$$T(z,s) = \frac{1}{2\pi}\int\limits_{0}^{s}\frac{\exp\left(-\frac{1}{2}z^2(1+t^2)\right)}{1+t^2}\,\mathrm{d}t.$$

(11)

Tail function $\mathcal{P}_7$ is the Pearson type VII distribution

$$\mathcal{P}_7(c;\gamma,m,c_0) = \frac{\Gamma(m)}{\gamma\sqrt{\pi}\Gamma(m-\frac{1}{2})}\left(1 + \frac{(c-c_0)^2}{\gamma^2}\right)^{-m}$$

(12)

with $m > 1/2$ and $\gamma > 0$. This distribution is a generalization of the Lorentz distribution where the tail shape can be changed; the specific case $\mathcal{P}_7(c;\gamma,1,c_0)$ is the Lorentz distribution with half width at half-maximum $\gamma$. ISRF function $\mathcal{R}(c;d,s,w,\eta,\gamma,m,c_0)$

consists of a peak function with three shape parameters $d$, $s$ and $w$, and a tail function with two shape parameters $\gamma$ and $m$:

$$\mathcal{R}(c;d,s,w,\eta,\gamma,m,c_0) = (1-\eta)\,\mathcal{S}(c;d,s,w,c_0)$$
$$+ \eta\,\mathcal{P}_7(c;\gamma,m,c_0). \tag{13}$$

The mean is at $c_0$, the integral is 1 and the integral over the tail part only is $\eta$.

The requirement on the TROPOMI-SWIR ISRF states that the ISRF should be known with an accuracy of 1% of its maximum. A stringent implementation of the requirement would be that the absolute value of all residuals should be less than 1% of the maximum of the ISRF (about 0.004), but that leaves no room for outliers. An alternative measure of the fit quality turns out to be an rms value calculated as the square root of the sum of the squared difference between the ISRF fit residuals, using points where the fit function is larger than 6% of its maximum, divided by the number of ISRF data points minus the number

of free fit parameters. The threshold of 6% is arbitrary, but a lower value would include more of the tails where the residuals are always very small, which would make this measure less sensitive. The advantage of this measure is that it is less sensitive to small outliers, and sensitive to large outliers which can corrupt the fit procedure. Therefore, we use this rms as a measure of the fit quality.

## 3.2    ISRF parameter iteration

In the iterative approach as introduced above we start with a simplified model for the ISSF to determine the central wavelength and intensity of the laser. Many ISSF fits are used to estimate the (local) ISRF, so the ISRF can be determined with many parameters. For the next iterations we use the fact that the ISSF is equal to the mirrored local ISRF, and the only free parameters in the successive iterations in the ISSF fit are the wavelength and the intensity.

     This procedure and the number of iterations have been verified using synthetic ISRF data. About 2000 synthetic ISRFs

have been constructed using the ISRF model (Eq. 13) and combinations of the 7 parameters covering most of the parameter space. This data set has been generated with and without realistic noise. The simulations confirmed our guess that not all ISRF parameters can be fitted at the same time, because the tail parameters $\gamma$ and $m$ are not independent and the fit of these parameters is further complicated by the small signal of the tails. In early computations to determine the TROPOMI-SWIR ISRF, the tail parameter $m$ was fixed to 1.2, based on the shape of the tails found in stray-light measurements. The simulations

with synthetic data shows it is better not to fix $m$, because its true value is poorly known and it improves the convergence of the other parameters. The best convergence towards the true ISRF, both in speed and in accuracy, is achieved by fixing peak width parameter $w$. However, in this case $w$ needs to be known better than 1%, which is unrealistic. Second best is to fix tail fraction $\eta$. The tail fraction is nearly constant across the detector and from detailed analysis of the ISRF determined after the first iteration its value is usually be between 0.1 and 0.12. Therefore, assuming a tail fraction of 0.11 would work for most

ISRF fitting. To overcome problems in case the true tail fraction is outside this range, the ISRF fit is performed twice. First only fixing the tail fraction (from the previous iteration), then only fixing $w$ (improved guess from the previous ISRF fit). We refer to each repeated step of the method as "stages", because each step contains several computations: one ISSF fit and two ISRF fits.

This approach generates consistent good results after 4 stages. Synthetic data show that the differences between the determined ISRF and the true ISRF are within 0.0005 (about $0.125\%$ of the ISRF maximum). Adding realistic noise has a very minor effect, due to the large number of measured ISSF samples: the residuals are doubled and the determined ISRF is more slightly symmetrical. Because the true ISRF is generated with the same ISRF model used for the fit, one would expect that the determined ISRF matches the true ISRF perfectly. However, this is not the case due to the fact that details of the true ISRF are lost in the fit of the poorly sampled ISSF. According to an extra simulation, the determined ISRF matches the true ISRF nearly perfectly when the ISSF signal is measured with 10 instead of 5 spectral pixels.

The convergence of the method is illustrated in Fig. 6, where the ISRF shapes presented in Figs. 3–5 are simulated. For all five simulated ISRFs a quick convergence is achieved. The large residuals between the asymmetric ISRF after stage 1 and true ISRF are due to a shift in the peak position, while the shape of the final ISRF agrees with the true ISRF, see Table 1.

## 3.3 ISRF parameter smoothing

The goal of the on-ground calibration is to determine the SWIR ISRF for all pixels, because trace-gas retrieval needs the ISRF for the whole SWIR spectral range and for all swath angles. However, the ISRF could not be measured for all pixels due to measurements with too strong laser signal (rejected before analysis), bad pixels and laser instability. Therefore, it is necessary to interpolate the local ISRFs. Minor problems with the OPO are still present in the ISRF data points (see Figs. 3–5), which may affect the ISRF fit yielding minor deviations of the fit parameters. Assuming that these deviations are random and that the shape of the ISRF varies only smoothly over the surface of the detector (as it is determined by the spectrometer optics), then the quality of the ISRF would benefit when the ISRF parameters are smoothed and interpolated using bivariate polynomial fitting. Any high-frequency detector features still present in the data due to imperfect calibration or local changes in the detector PSF should then be visible in the difference between the local and smoothed parameter values.

The selected bivariate polynomial fit uses Chebyshev polynomials of the first kind $T_n$ to minimizes the problem of Runge's phenomenon. The model at a location specified by row $r$ and column $c$, where the first row and column are mapped to $-1$ and the last row and column are mapped to $+1$ is given by

$$E_{\text{fit}}[r,c;\boldsymbol{a}] = \sum_{m=0}^{M} \sum_{n=0}^{m} a_{mn} T_{m-n}\left(2\frac{r}{n_{\text{row}}-1}-1\right) T_n\left(2\frac{c}{n_{\text{col}}-1}-1\right), \tag{14}$$

where $n_{\text{row}} = 256$ and $n_{\text{col}} = 1000$ for the SWIR detector. The order $M$ is the maximum of the sum of the exponents.

The bivariate polynomial fitting is sensitive to obvious outliers. Therefore, ISRF fits have been rejected when the fit quality is low: rms $> 0.003$, skew parameter $|s| > 5$, tail parameter $\gamma$ is outside the range [0, 3] or tail parameter $m$ is outside the range [0.5, 3]. The order $M$ of the bivariate polynomial fit is optimized for each ISRF parameter to minimize the variance of the difference between the raw values and the fitted value. An order $M = 6$ is used for skew parameter $s$, $M = 4$ for peak parameters $d$ and $w$, and $M = 2$ for tail parameters $\gamma$ and $m$.

The smoothed ISRF parameters are employed to calculate the ISRF calibration key data for each and every pixel. Residuals are examined to check whether the excluded local variations are small enough to ignore. Actually, the smoothed ISRF param-

eters are calculated at the end of each stage, because the ISSF fit of the next stage, in general, benefits when erroneous ISRF fit results are not propagated to the next stage.

## 4  Discussion of results

Of the 2000 simulations with synthetic data performed, the results of those that closely resemble the measured ISRF examples as shown in Figs. 3–5 will be used for illustration. The convergence towards a true ISRF from synthetic measurements is presented in Fig. 6. These simulations are performed with an ideal laser: constant wavelength scan speed (using the average 0.00125 nm/s from the measurements) and constant laser signal. As expected, the fit residual and the deviation from the true ISRF is still significant after the first stage, but the procedure quickly converges in subsequent stages. The final residuals and differences with respect to the true ISRF are much smaller than the requirement on the ISRF knowledge. Table 1 shows that all ISRF parameters converge towards their true values during the ISRF parameter iteration.

The local ISRFs determined from irradiance measurements (Fig. 3c) and radiance measurements (Fig. 4c) have typical fit residuals smaller then 0.004, except for a few outliers. These outliers correspond with the presence of (small irregular) wavelength jumps during the measurements. Some of the fit residuals show systematic features, but they are not consistent between both datasets. Therefore, the ISRF model (Eq. 13) is considered a good representation of the ISRF shape. The fit residuals determined from the on-board diode-laser measurements (Fig. 5c) are less noisy, because the speed of the wavelength scan was much lower and the diode lasers were better behaved.

Several fit thresholds are introduced in section 3.3, to optimize the bivariate polynomial fitting of the ISRF parameters. Low quality fits are identified by a high rms value. The median rms value is about 0.0017 for irradiance measurements, hence a reasonable threshold value is set at 0.003. Saturated measurements are rejected from analysis, but some ISRF fits are still affected by nearby saturation. Therefore, ISRF fits with an exceptionally large skew-normal width value are rejected.

Of 211,575 pixels of the SWIR detector in the operational wavelength and swath ranges, we have ISRF data measured through the irradiance port of 195,892 pixels. For 7 % of the pixels ISRF data was not available or rejected due to possible saturation. The ISRF parameter smoothing at the end of stage 1 is based on about 116,000 local ISRF, because 73,616 ISRF fits were rejected due to the rms condition and about 6000 due to too large skew and skew-normal width. The number of used ISRF fits increased per stage to about 164,000 at stage 4, with only 28,900 rejected due to the rms condition and about 2,400 due to too large skew and skew-normal width. The rms values of the irradiance ISRF are shown in Fig. 7a.

The number for the radiance measurements are significantly lower: ISRF data is available for 150,000 pixels. For 30 % of the pixels ISRF data was not available or rejected due to partial illumination of the pixels by the laser spot. The ISRF parameter smoothing at the end of stage 1 is based on about 63,000 local ISRFs, because 84,300 ISRF fits were rejected due to the rms condition and about 3000 due to too large skew and skew-normal width. The number of used ISRF fits increased per stage to about 92,500 at stage 4, with only 51,300 rejected due to the rms condition and about 2,000 due to too large skew and skew-normal width. The rms values of the radiance ISRF are shown in Fig. 7b.

The difference between the ISRF data points and the ISRF model are very small, as the median of the rms is 0.0015 for both irradiance and radiance measurements, but the coverage of the first is much better. Therefore, the smoothed ISRF derived from irradiance measurements are used to generate the TROPOMI-SWIR ISRF calibration key data. The ISRF model parameters are presented in Figs. 8–13. The skew-normal width parameter is sensitive to the photo-sensitivity of the SWIR detector (Fig. 6 of Hoogeveen et al., 2013). These patterns are easily recognizable in the residual plot (Fig. 8b), where the difference is shown between the ISRF parameter $d$ and its bivariate poynomial fit.

Figure 14 shows the differences between the smoothed ISRF and local ISRFs determined from irradiance, radiance and diode-laser measurements at the five locations presented in Figs. 3–5. The fit parameters are compared in Table 2. The differences between the ISRF data points and the smoothed ISRF are for most measurements within the requirement on the ISRF. In general, all local ISRF fits with rms values less than 0.002 in Fig. 7 have also small residuals against the smoothed ISRF. Therefore, we conclude that smoothed ISRF, used to derive the ISRF CKD, agrees well with the local ISRF data.

## 5   In-flight monitoring of ISRF

As knowledge of the ISRF is critical for the science results of the SWIR band, it has been decided to include means to identify possible changes in the ISRF between the on-ground calibration campaign and the first measurements in space, and to monitor the ISRF stability during the 7 years of operational lifetime. For this, five tunable DFB diode lasers, spread evenly over the SWIR wavelength range, are included in the on-board calibration unit. Technical details of the on-board lasers are given in Sect. 2.2. During the in-flight commissioning phase, the on-ground measurements with the on-board lasers will be repeated with a moving diffuser. Hence, these ISRFs can be compared. The early in-flight measurements also act as a "reference ISRF" for the ISRF monitoring.

During the operational phase, dedicated measurements are planned to monitor ISRF stability. These measurements will be performed with each laser, roughly once per month with a fixed diffuser, because the diffuser mechanism is a life-limited item. The ISRF determined from these measurements ("monitoring ISRF") is less accurate as it is affected by laser speckle patterns. It is used for monitoring only, not for trace-gas retrieval. If recalibration of the ISRF is necessary, then the wavelength scan of the on-board diode lasers has to be maximized and a moving diffuser should be used.

The monitoring ISRF is determined with the algorithm presented in Sect. 3 without iterations. It starts with the ISSF fit of stage 4 using the parameters of a reference ISRF, determined earlier in the mission with the same diode laser. Then, for each illuminated row, ISRF data of several spectral pixels are combined to generate one ISRF up to 5 pixels from its center. Speckle patterns are reduced by a median calculated from all ISRF data points. This monitoring ISRF will be compared with a monitoring ISRF obtained early in the commissioning phase and used in trend analyses. It is expected that this method is sensitive enough to be used for long-term monitoring, and being able to distinguish between changes in the real instrument ISRF and changes in the speckle pattern.

## 6 Conclusions

An approach is presented to determine high-resolution ISRFs for a spectrometer that measures the ISSF with only a few spectral pixels, using a scanning monochromatic light source. The instrument itself is used to determine the wavelength and intensity of the light, which makes it possible to perform the necessary measurements within a reasonable time (days) even for detectors with more than 100,000 pixels.

The wavelength and intensity of the signal are determined with a fit to the measurements, but the model used for this ISSF fit is decisive for the accuracy of the ISRF determination, as is shown using simulations with synthetic ISRF data. Based on the simulations, an iterative approach is developed to improve the ISSF model from a simple model in the first iteration to the mirrored ISRF model in later iterations. The simulations show satisfactory convergence to the true ISRF in 4 iterations.

The TROPOMI-SWIR ISRF is modeled by the weighted sum of functions for the peak and the tails. The peak function is a skew-normal distribution convolved with a uniform distribution. This corresponds to a (possibly asymmetric) image of the slit on the detector, with the optics blurring the image. For the tails a function is needed with an adjustable tail shape. A suitable function is the Pearson type VII distribution, which is a generalization of the Gauss and Lorentz distributions. It can represent the wings of SWIR spectral stray light satisfactorily. Each of the five ISRF shape parameters are smoothed by fitting a bivariate polynomial to derive the ISRF calibration key data for all SWIR wavelengths and swath angles.

The ISRF measured through the irradiance port using the solar diffuser has been compared with the equivalent ISRF measured via the radiance port. The differences are very small, and are mostly due to measurement details, not instrument details. Calibration key data for the ISRF have been derived from the larger irradiance data set. The determined ISRF meets the requirement on ISRF knowledge and should thus be sufficient for SWIR trace-gas retrievals.

The on-board calibration unit contains five diode lasers in the SWIR wavelength range. Accurate measurements with these diode lasers before and after launch will reveal whether the ISRF remained stable and the ISRF calibration key data can be applied for data retrieval. During operations, the lasers will be used to monitor the long-term stability of the optical properties of the SWIR module.

## 7 Data availability

The underlying data of the figures presented in this publication can be found at ftp://ftp.sron.nl/open-access-data/richardh.

*Competing interests.* The authors declare that they have no conflict of interest.

*Acknowledgements.* The authors would like to thank the teams of Airbus Defence and Space Netherlands and KNMI for organizing the calibration campaign and in particular the operators for the tireless data acquisition. The authors would also like to thank the anonymous referees for their thorough reviews of the original manuscript.

TROPOMI is a collaboration between Airbus Defence and Space Netherlands, KNMI, SRON and TNO, on behalf of NSO and ESA. Airbus Defence and Space Netherlands is the main contractor for the design, building and testing of the instrument. KNMI and SRON are the principal investigator institutes for the instrument. TROPOMI is funded by the following ministries of the Dutch government: the Ministry of Economic Affairs, the Ministry of Education, Culture and Science, and the Ministry of Infrastructure and the Environment.

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

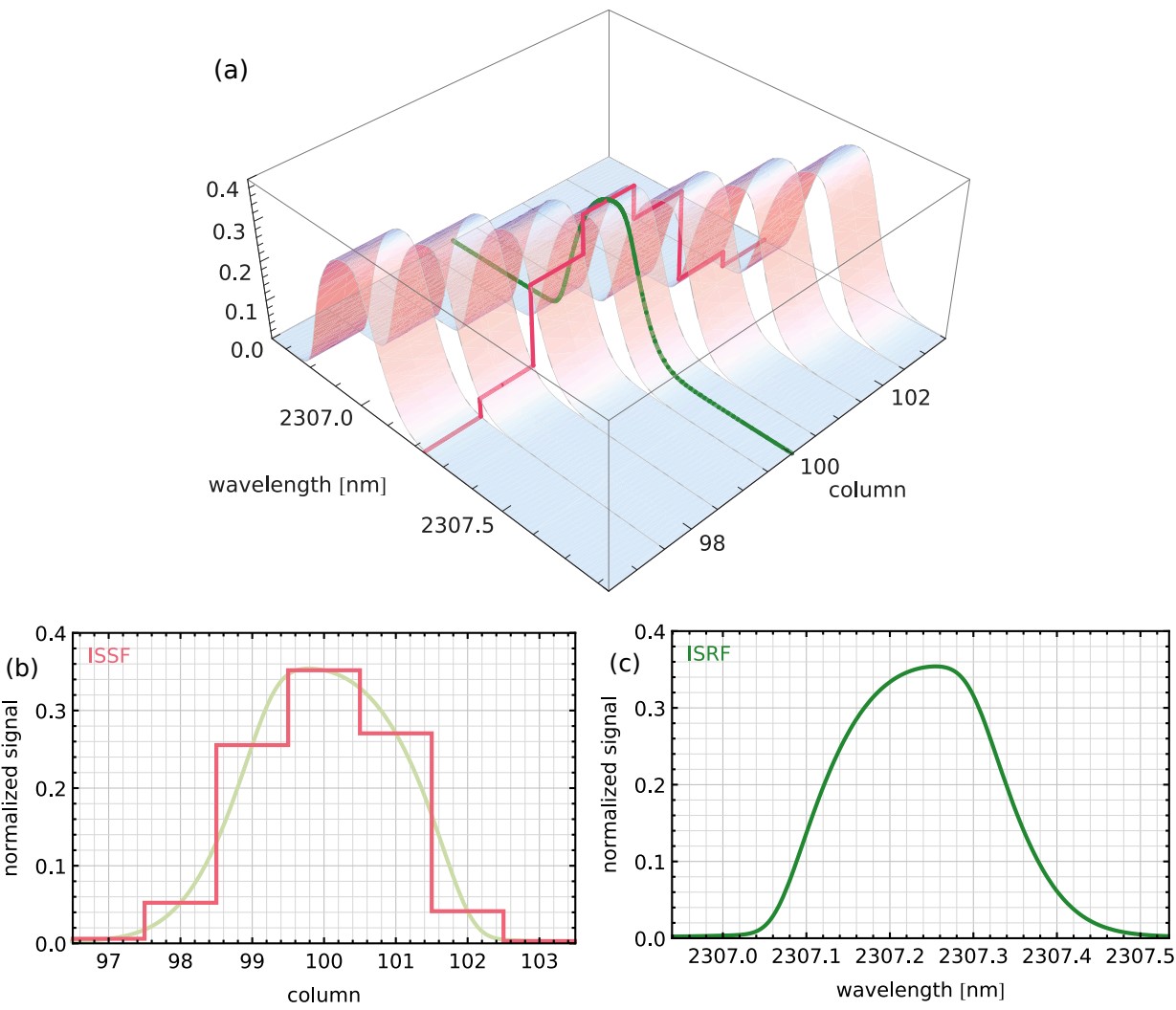

**Figure 1.** Normalized signal as a function of source wavelength and pixel on an arbitrary row, with two cross sections: the ISSF at 2307.24 nm (red) and the ISRF of the pixel in column 100 (green). In the plot of the ISSF, a mirrored version of the ISRF is shown in light green. The skew of the ISRF has been exaggerated to show the mirroring.

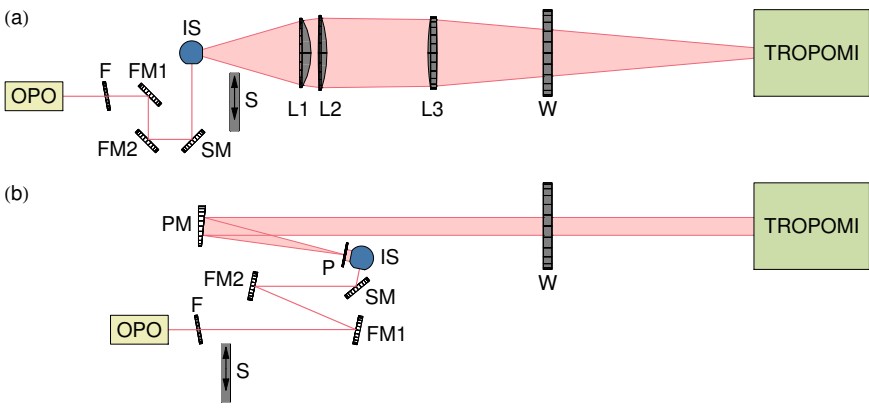

**Figure 2.** The setup for (a) irradiance measurements and (b) radiance measurements. The elements after the OPO are: neutral density filter F, folding mirrors FM1 and FM2, spinning mirror SM, integrating sphere IS, shutter S, lenses L1, L2 and L3, field stop P, parabolic mirror PM and window W of the vacuum chamber containing TROPOMI. The light enters (a) the Sun port or (b) the Earth port of the instrument.

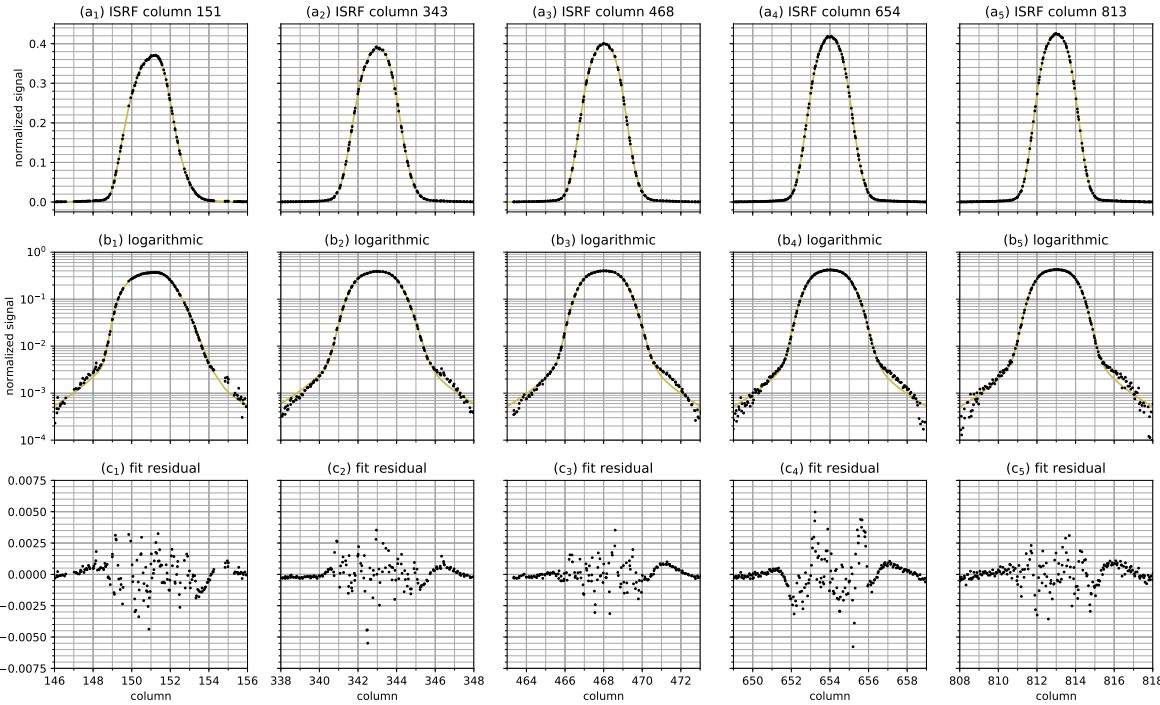

**Figure 3.** Five typical SWIR ISRFs determined from on-ground measurements with an external laser through the TROPOMI irradiance port. The upper panels (a) show the shapes of the ISRF at location: columns 151, 343, 468, 654 and 813, resp., and rows 24, 76, 118, 155 and 191, resp. The middle panels (b) show the same data on a logarithmic scale. The lower panels (c) show the difference between the ISRF data points and the ISRF fit (end of stage 4).

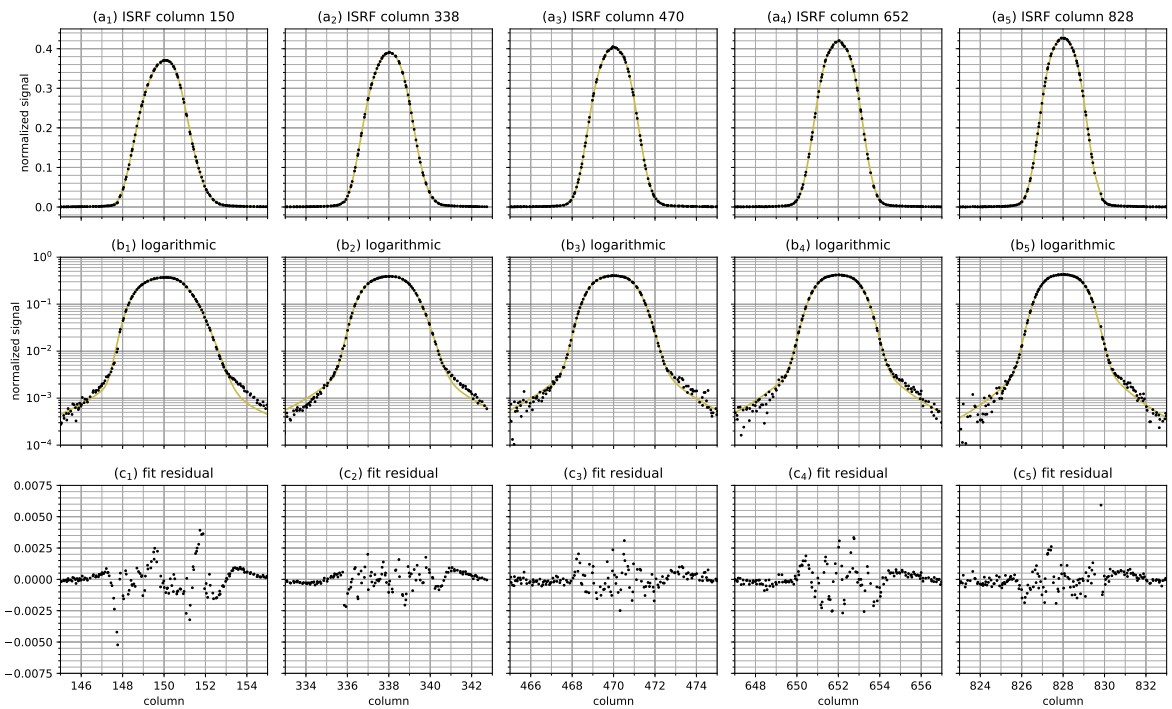

**Figure 4.** Five typical SWIR ISRFs determined from on-ground measurements with an external laser through the TROPOMI radiance port. The upper panels (a) show the shapes of the ISRF at location: columns 150, 338, 471, 652 and 828, resp., and rows 24, 76, 118, 155 and 191, resp. The middle panels (b) show the same data on a logarithmic scale. The lower panels (c) show the difference between the ISRF data points and the ISRF fit (end of stage 4).

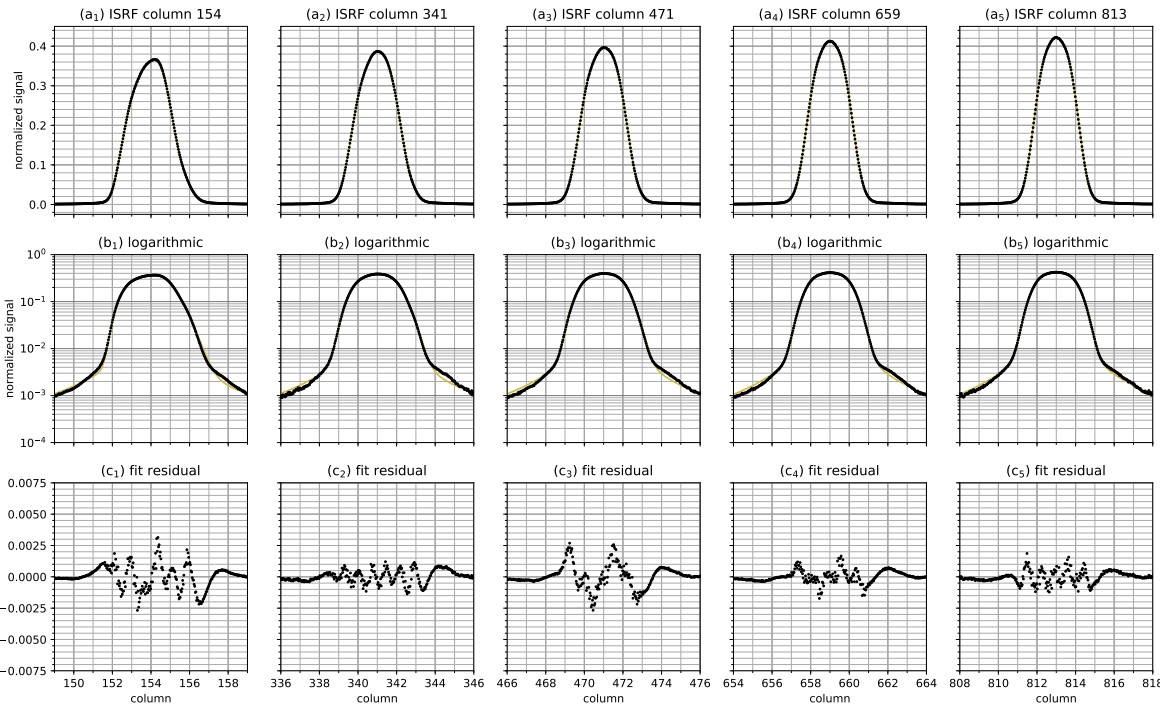

**Figure 5.** SWIR ISRF determined from on-ground measurements with the five on-board diode lasers. The upper panels (a) show the shapes of the ISRF at location: columns 154, 341, 471, 659 and 813, resp., and rows 24, 76, 118, 155 and 191, resp. The middle panels (b) show the same data on a logarithmic scale. The lower panels (c) show the difference between the ISRF data points and the ISRF fit (end of stage 4).

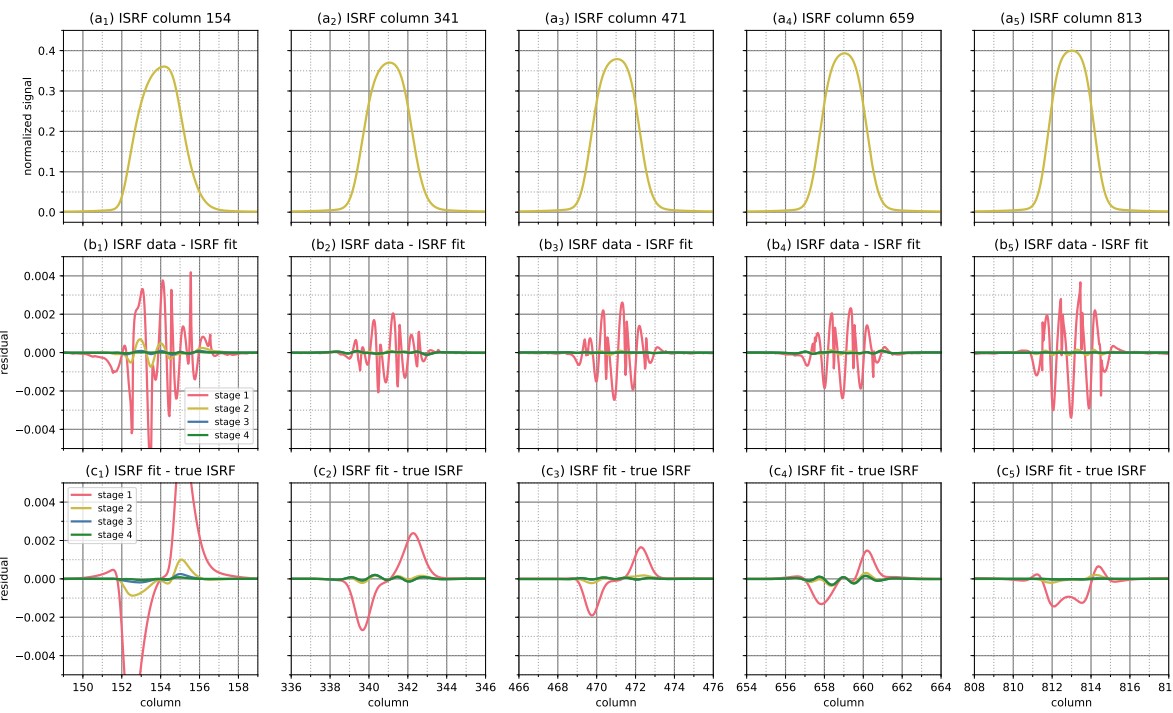

**Figure 6.** Convergence of five synthetic ISRF determinations in four stages. The upper panels (a) show the shapes of synthetic ISRF closely resemble the ISRF examples as shown in Figs. 3–5. The middle panels (b) show the difference between the ISRF data points and the ISRF fit. The lower panels (c) show the difference between the ISRF data points and the true ISRF.

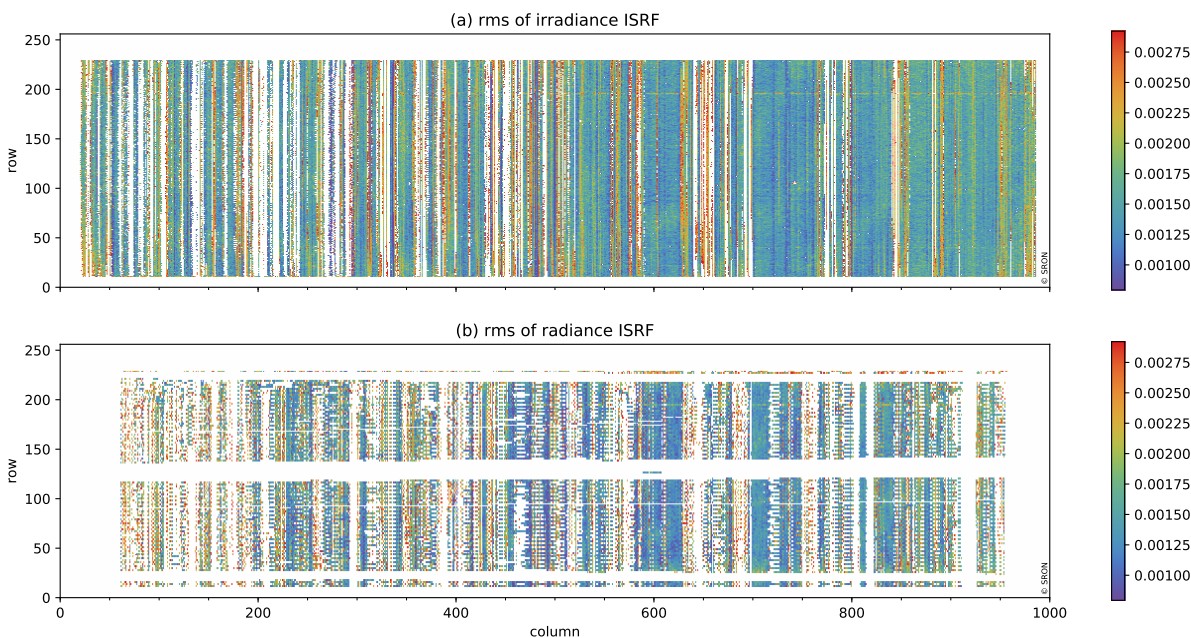

**Figure 7.** Fit quality of the local ISRF using the (a) irradiance port and (b) radiance port. No measurements are performed in the white edges. Panel (a) white vertical stripes are due to saturation in the measurements, and red vertical stripes are due to nearby saturation or laser instabilities. Panel (b) white areas are due to partly illuminated rows.

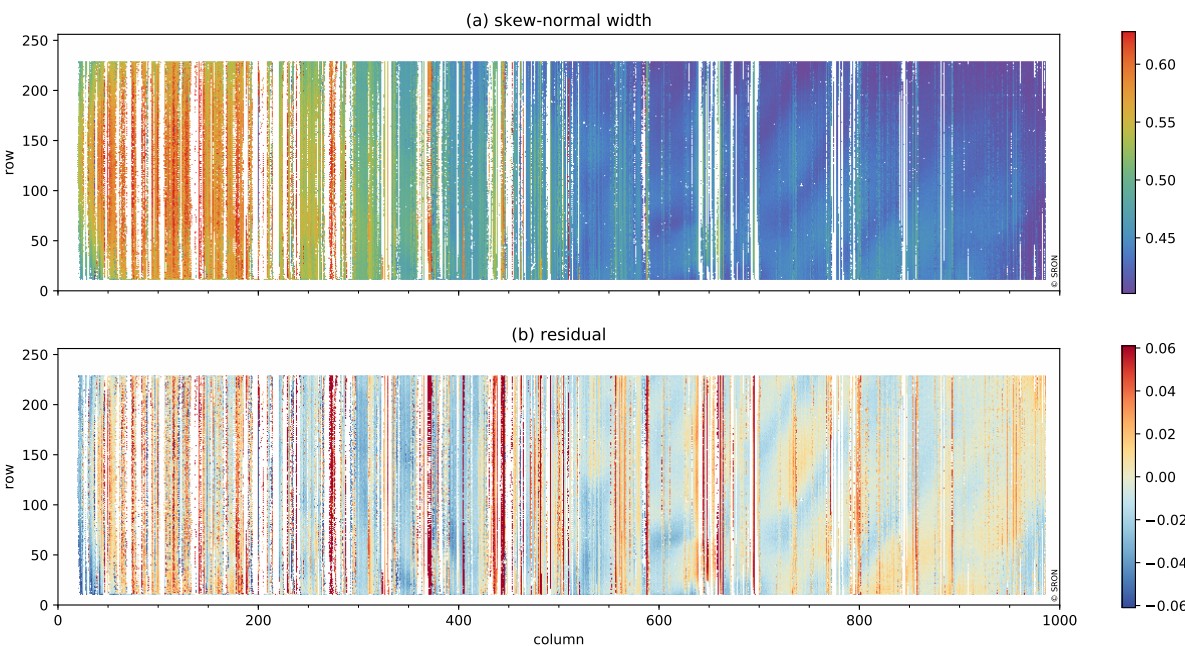

**Figure 8.** ISRF parameter skew-normal width $d$ (irradiance port). Panel (b) shows the difference between $d$ and its bivariate poynomial fit. No measurements are performed in the white edges. White vertical stripes are due to saturation in the measurements. Red vertical stripes are due to nearby saturation or laser instabilities.

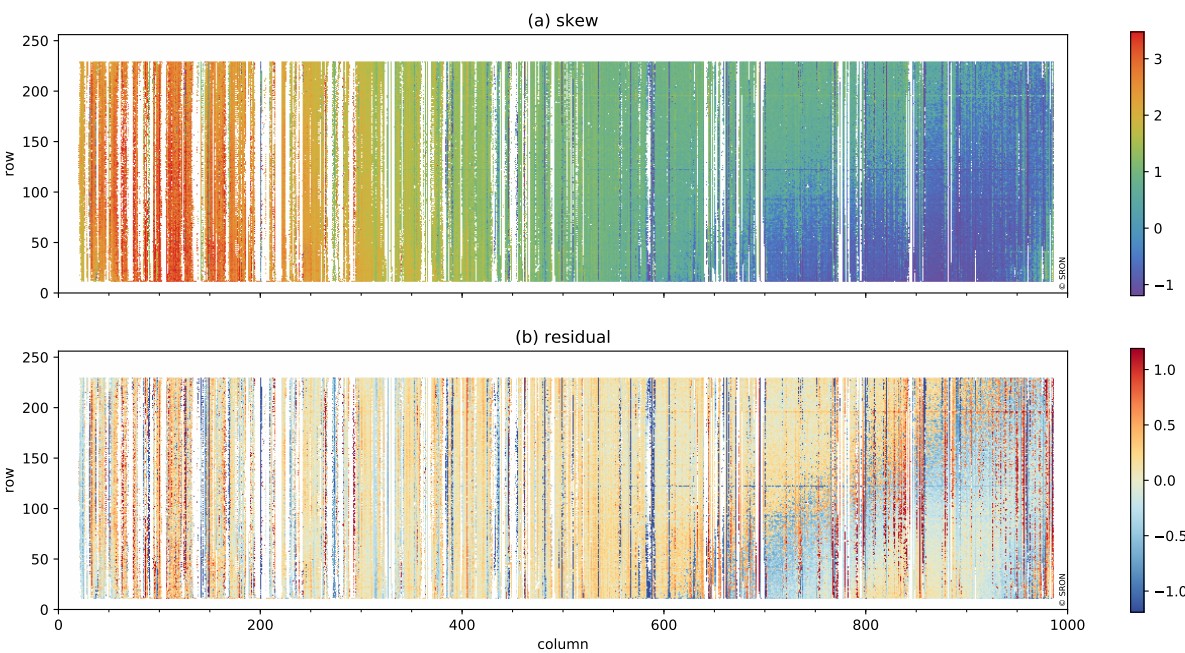

**Figure 9.** ISRF parameter skew $s$ (irradiance port). Panel (b) shows the difference between $s$ and its bivariate poynomial fit. No measurements are performed in the white edges. White vertical stripes are due to saturation in the measurements. Red vertical stripes are due to nearby saturation or laser instabilities.

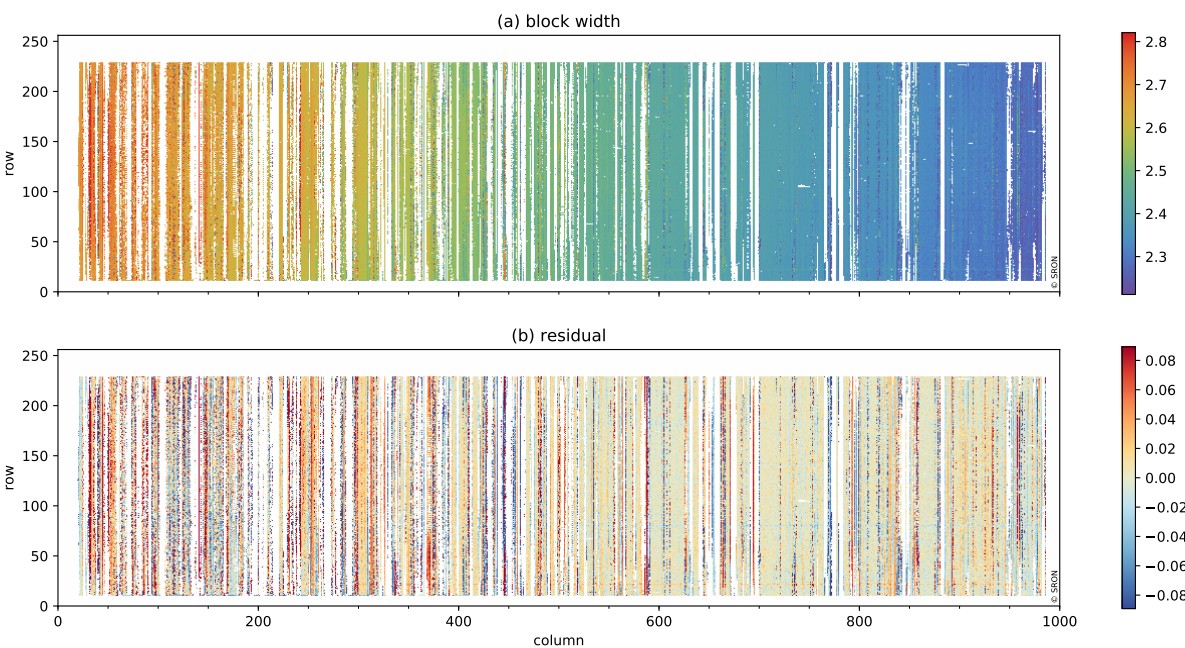

**Figure 10.** ISRF parameter block width $w$ (irradiance port). Panel (b) shows the difference between $w$ and its bivariate poynomial fit. No measurements are performed in the white edges. White vertical stripes are due to saturation in the measurements. Red vertical stripes are due to nearby saturation or laser instabilities.

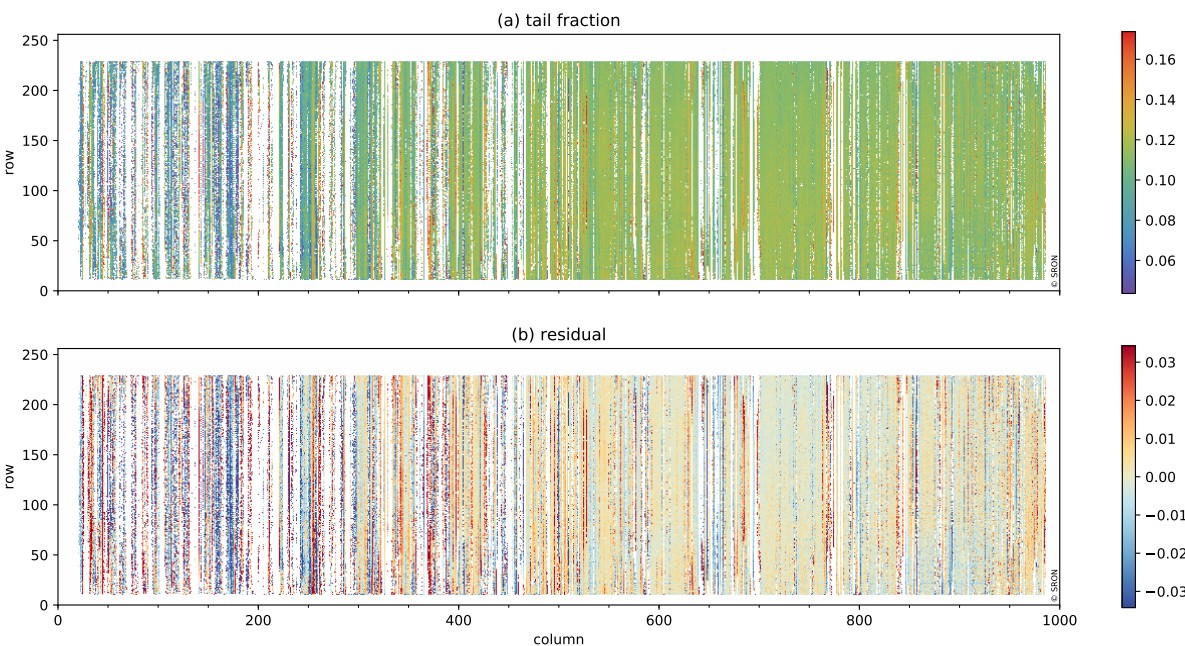

**Figure 11.** ISRF parameter tail fraction $\eta$ (irradiance port). Panel (b) shows the difference between $\eta$ and its bivariate poynomial fit. No measurements are performed in the white edges. White vertical stripes are due to saturation in the measurements. Red vertical stripes are due to nearby saturation or laser instabilities.

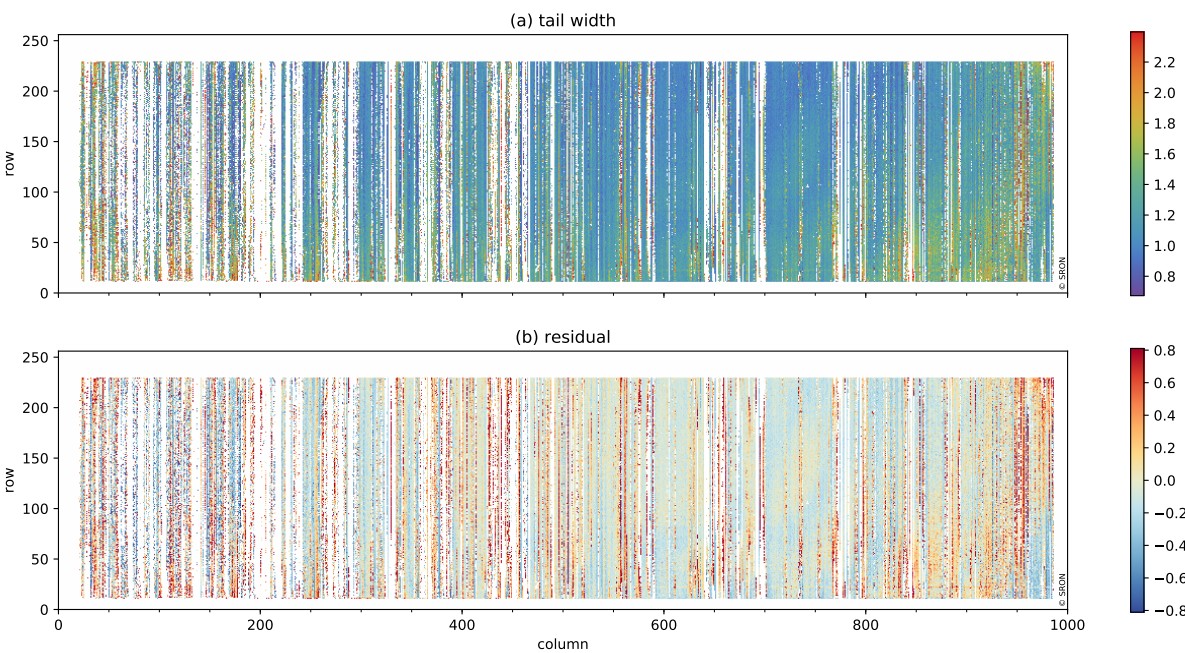

**Figure 12.** ISRF tail parameter $\gamma$ (irradiance port). Panel (b) shows the difference between $\gamma$ and its bivariate poynomial fit. No measurements are performed in the white edges. White vertical stripes are due to saturation in the measurements. Red vertical stripes are due to nearby saturation or laser instabilities.

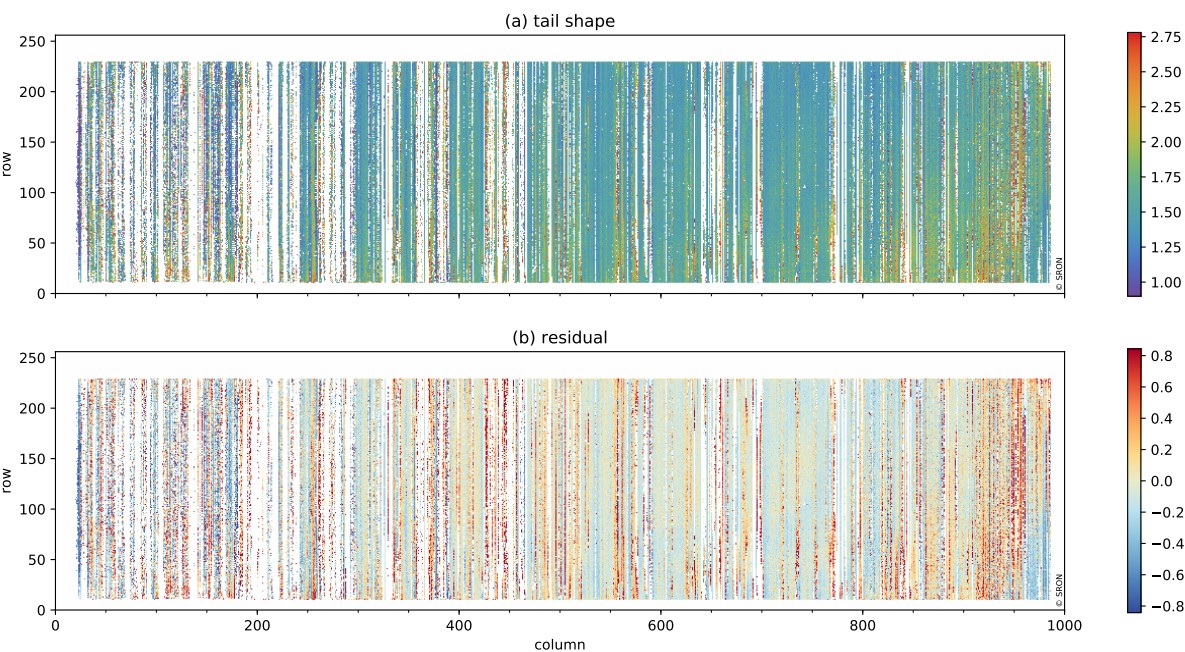

**Figure 13.** ISRF tail parameter $m$ (irradiance port). Panel (b) shows the difference between $m$ and its bivariate poynomial fit. No measurements are performed in the white edges. White vertical stripes are due to saturation in the measurements. Red vertical stripes are due to nearby saturation or laser instabilities.

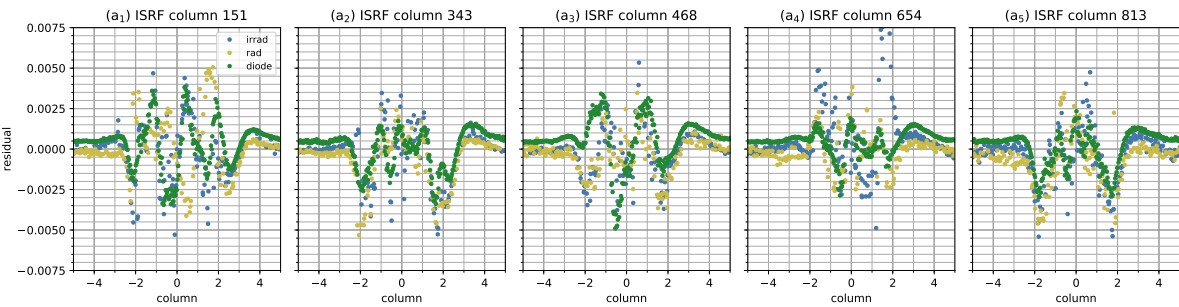

**Figure 14.** Differences between the local ISRF and the smoothed ISRF for irradiance, radiance and diode-laser measurements. This figure illustrates that smoothed ISRF data agree well with the local ISRF data presented in Figs. 3–5.

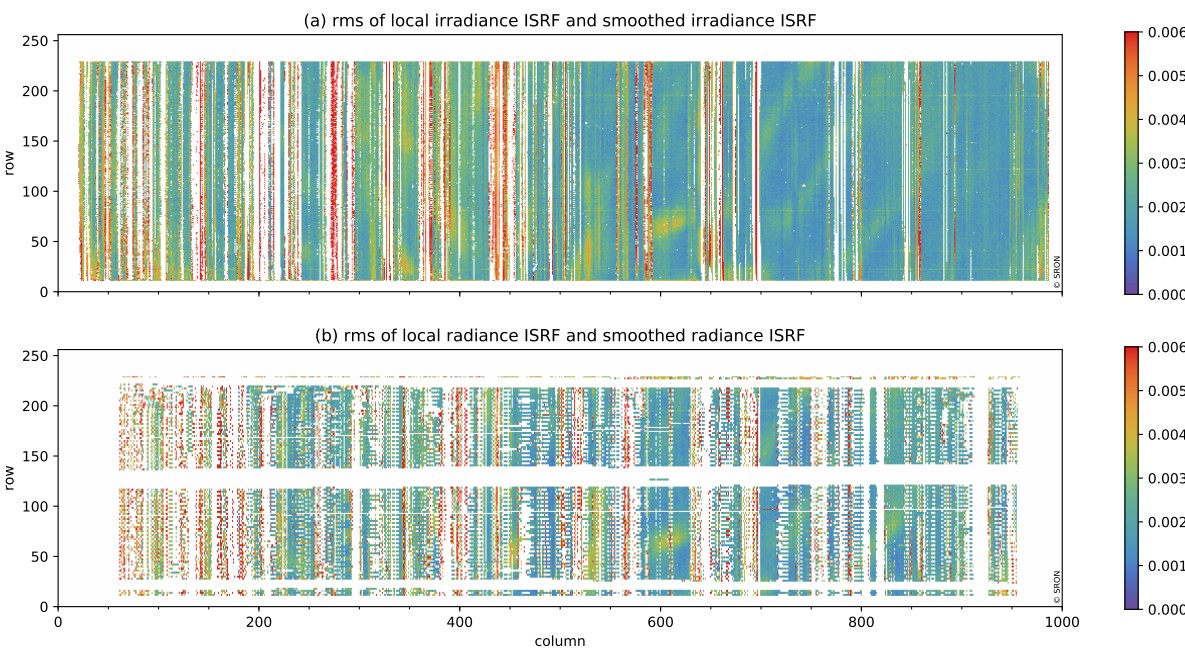

**Figure 15.** Fit quality of the smoothed ISRF using the (a) irradiance port and (b) radiance port. No measurements are performed in the white edges. Panel (a) white vertical stripes are due to saturation in the measurements, and red vertical stripes are due to nearby saturation or laser instabilities. Panel (b) white areas are due to partly illuminated rows.

**Table 1.** The convergence of the ISRF parameter iteration is shown by listing the determined ISRF parameters at the end of four stages. The simulations are performed on five synthetic ISRFs representative for the determined TROPOMI-SWIR ISRF, equally spaced at positions on the SWIR detector from top left to bottom right. The synthetic data are generated without noise. The shape parameters for the true ISRF are listed in italic.

| pixel | stage | $c_0$ | $d$ | $s$ | $w$ | $\eta$ | $\gamma$ | $m$ |
|-------|-------|-------|-----|-----|-----|--------|----------|-----|
| (47, 154) | 1 | 0.0227 | 0.5671 | 2.6562 | 2.6480 | 0.12 | 1.6715 | 2.0396 |
| | 2 | 0.0011 | 0.5683 | 2.7282 | 2.6544 | 0.1038 | 1.3396 | 1.6791 |
| | 3 | 0.0002 | 0.5696 | 2.7292 | 2.6499 | 0.1016 | 1.4137 | 1.6835 |
| | 4 | 0.0 | 0.5703 | 2.7257 | 2.6481 | 0.1004 | 1.4171 | 1.6812 |
| | true | *0.0* | *0.5709* | *2.7202* | *2.6464* | *0.0989* | *1.4142* | *1.6701* |
| (79, 341) | 1 | 0.0102 | 0.5178 | 1.5651 | 2.5561 | 0.12 | 1.4770 | 1.9236 |
| | 2 | 0.0002 | 0.5156 | 1.5839 | 2.5664 | 0.1152 | 1.2991 | 1.6972 |
| | 3 | 0.0 | 0.5155 | 1.5826 | 2.5669 | 0.1133 | 1.2654 | 1.6497 |
| | 4 | 0.0 | 0.5157 | 1.5813 | 2.5664 | 0.1123 | 1.254 | 1.6317 |
| | true | *0.0* | *0.5173* | *1.5768* | *2.5621* | *0.1083* | *1.2404* | *1.599* |
| (118, 471) | 1 | 0.0036 | 0.4724 | 0.991 | 2.4808 | 0.12 | 1.5179 | 2.0329 |
| | 2 | 0.0002 | 0.4695 | 1.0147 | 2.4958 | 0.1137 | 1.2356 | 1.6602 |
| | 3 | -0.0 | 0.4690 | 1.0151 | 2.4984 | 0.1119 | 1.1794 | 1.5879 |
| | 4 | -0.0 | 0.4688 | 1.0152 | 2.4993 | 0.1112 | 1.1585 | 1.5612 |
| | true | *0.0* | *0.4680* | *1.0163* | *2.5015* | *0.1122* | *1.1470* | *1.5525* |
| (155, 659) | 1 | 0.0013 | 0.4375 | 0.7156 | 2.3934 | 0.12 | 1.597 | 2.1619 |
| | 2 | 0.0003 | 0.4384 | 0.7553 | 2.4083 | 0.1111 | 1.2278 | 1.6427 |
| | 3 | 0.0 | 0.4354 | 0.7575 | 2.4109 | 0.1085 | 1.1607 | 1.557 |
| | 4 | 0.0 | 0.4353 | 0.7578 | 2.412 | 0.1078 | 1.1363 | 1.5267 |
| | true | *0.0* | *0.4318* | *0.7615* | *2.4215* | *0.1145* | *1.1173* | *1.5400* |
| (191, 813) | 1 | 0.0003 | 0.4292 | 0.4438 | 2.3396 | 0.12 | 1.5864 | 2.1123 |
| | 2 | 0.0001 | 0.4275 | 0.4879 | 2.3519 | 0.1121 | 1.2543 | 1.6513 |
| | 3 | 0.0 | 0.4279 | 0.4917 | 2.3539 | 0.1096 | 1.1972 | 1.5776 |
| | 4 | 0.0 | 0.4279 | 0.4924 | 2.3549 | 0.1088 | 1.1777 | 1.553 |
| | true | *0.0* | *0.4258* | *0.4940* | *2.3607* | *0.1131* | *1.1564* | *1.5544* |

**Table 2.** Listing of ISRF parameters (Eq. 13) of five typical SWIR ISRFs determined from irradiance (irr), radiance (rad) and diode-laser (ld) measurements, respectively. The location of the ISRF are (approximately) equally spaced at positions on the SWIR detector from top left to bottom right.

| pixel | type | $d$ | $s$ | $w$ | $\eta$ | $\gamma$ | $m$ | rms |
|-------|------|------|--------|---------|--------|--------|--------|--------|
| (47, 151) | irr | 0.5573 | 3.3047 | 2.66357 | 0.1000 | 1.3028 | 1.5631 | 0.0016 |
| (47, 150) | rad | 0.5771 | 2.2230 | 2.67038 | 0.0605 | 0.9152 | 1.1827 | 0.0015 |
| (47, 154) | ld | 0.5681 | 3.1603 | 2.64913 | 0.0860 | 0.9437 | 0.9104 | 0.0011 |
| (79, 343) | irr | 0.4923 | 1.4446 | 2.57038 | 0.1184 | 1.1696 | 1.5006 | 0.0016 |
| (79, 338) | rad | 0.4941 | 1.5636 | 2.58129 | 0.1005 | 0.9051 | 1.2890 | 0.0009 |
| (79, 341) | ld | 0.4961 | 1.6075 | 2.58128 | 0.1126 | 0.9854 | 0.7520 | 0.0006 |
| (118, 468) | irr | 0.4539 | 1.2030 | 2.51046 | 0.1125 | 1.1190 | 1.4994 | 0.0012 |
| (118, 470) | rad | 0.4558 | 1.0764 | 2.50779 | 0.1037 | 0.9629 | 1.3429 | 0.0011 |
| (118, 471) | ld | 0.4561 | 1.2055 | 2.52954 | 0.1170 | 1.0469 | 0.8045 | 0.0013 |
| (155, 654) | irr | 0.4569 | 1.0076 | 2.42063 | 0.1001 | 1.0806 | 1.4331 | 0.0022 |
| (155, 652) | rad | 0.4341 | 0.5962 | 2.42946 | 0.1026 | 0.9643 | 1.3882 | 0.0016 |
| (155, 659) | ld | 0.4330 | 0.7617 | 2.42022 | 0.1117 | 0.9818 | 0.7381 | 0.0007 |
| (191, 813) | irr | 0.4122 | 0.7598 | 2.36538 | 0.1097 | 1.0196 | 1.4315 | 0.0015 |
| (191, 828) | rad | 0.4142 | -0.3014 | 2.35932 | 0.1015 | 1.0025 | 1.5010 | 0.0013 |
| (191, 813) | ld | 0.4168 | 0.6230 | 2.36022 | 0.1088 | 0.9447 | 0.7110 | 0.0007 |