# Peer review of "Determination of the TROPOMI-SWIR instrument spectral response function"

_Atmospheric Measurement Techniques, 2017_

## Referee Comment (RC1) · Anonymous Referee #1 · 25 Jan 2018

#############

General comments

#############

The paper addresses the determination of Instrument Spectral Response Functions (ISRF) of the recently launched Tropomi/Sentinel-5P mission. ISRF uncertainty is a notorious limitation of past and future space-borne atmospheric chemistry missions (e.g. GOME-2, OMI, Sentinel-4, Sentinel-5), as well as missions targeting greenhouses gases (OCO-2, MicroCarb, Sentinel mission). The SWIR band of the Tropomi/Sentinel-5P is used for retrieval of CO and CH4, where ISRF knowledge is most critical (together

with the NIR band) because of strong, narrow absorption features. In this sense, the paper addresses a critical aspect of trace and greenhouse gas retrieval and therefore the topic is of high scientific relevance.

The introduction of a new method for ISRF characterisation by on-ground calibration measurements, as promised by the title, would be of high interest for planned future missions. However, the manuscript fails to deliver key elements for introducing and making a case for a new method. Neither do the authors describe other techniques, nor do they motivate the introduction of a new one, explain the difference, or compare it with previous calibration/validation measurements. If the objective of the paper is the introduction of novel method for ISRF determination, as suggested by the title, the differences to previous instrument calibrations (e.g. SCIAMACHY, OMI, OCO-2) have to be pointed out. Ideally, a comparison shall be offered pointing to the advantages of the new approach.

In fact the reader is left wondering which are the new elements of the proposed approach: On the instrumental aspect: Is it the first time an OPO was used ? If so, why is it expected to yield better performance (than e.g. monochromator) ? What are the instrumental limitations of the approach ? The impact of key parameters of the measurement data (e.g. signal-to-noise ratio) is not discussed. No details on the instrumental setup are provided, and the quoted reference does not contain them either. On the modelling aspect: Is the novelty of the approach the mathematical model for the ISRF in terms of a peak and a tail function ? Then, why were these particular functions chosen and not any other ? Are there physical reasons why the ISRF tails should follow a Pearson type VII distribution ? All these questions are not addressed in the manuscript. In large parts, the text resembles an technical document (or ATBD), which describes a mathematical algorithm without explaining, why certain steps are taken.

The fitting procedure to determine the ISRF seems to suffer from an under-determined equation system, although this is never mentioned. This is mitigated by "fixing" some parameters selectively in the inversion during four iterative steps. Again, the authors

do not justify the presented sequence of partial fits, other than it results in "good fits" (defined by low residuals). This is particularly worrying as the approach is verified with two synthetic ISRF shapes, which were presumably computed using the same mathematical model as in the inversion (albeit this is not clear from the text). In theory, if the information content of the measurements would be sufficient to estimate all model parameters, this should result in perfect agreement between fit and forward model. However, the fit residuals clearly exhibit systematic features, indicating a weakness in the fit procedure. Nevertheless, the authors conclude "compliance" as the fit residuals' amplitude is below the accuracy threshold (1% requirement).

The iterative fit procedure yields ISRF shapes for all detector pixels, which are not uniform and clearly show high-frequency variation. At this point the authors argue that the ISRF is only determined by the spectrometer optics (PSF), which varies smoothly with wavelength and swath angle. This assumption is not in line with the definition of ISRF used in other publications, and even with the introduction of the present manuscript. Defined as the spectral response of a single pixel as a function of wavelength, the ISRF includes the detector PSF (convolved with the slit boxcar and the optical PSF). The detector PSF is mainly driven by cross-talk, which may indeed introduce a pixel-to-pixel or column-to-column variation. While such systematic features are clearly visible in the estimated parameters across the detector (e.g. Fig. 5), the authors indirectly dismiss them as artefacts (by the above assumption) and eliminate them with an elaborate "smoothing procedure".

The smoothing equations (bi-variate polynomial fit) are reported, without justifying the choice of parameters (merely stating to yield "smoother and better" results). It is impossible to judge the impact (and therefore significance) of the smoothing, since no statistics are provided as of how much the individual parameter have been smoothed. The difference w.r.t the original fit results is not presented. Pixel-dependent effects (e.f. cross-talk) are "smoothed out" by this procedure, but presence of such effects does not mean that the individual ISRFs determined in the previous step were less accurate.

[Figure]

The use of smoothing procedures gives the impression that the authors do not have much confidence in their technique. They regard the smoothed ISRF parameters as more accurate, and report the observation that the individual rms deviation from the measurements have increased as "counter-intuitive" (although seems to be expected).

It is understood that for practical purposes (operational Level-2 processing), calibration key data cannot take into account ISRF variations at the pixel-to-pixel level. However, the authors should clearly state where such compromises are made (e.g. between level of detail and computational resources) and justify them by quantifying or estimating the impact on Level-2 accuracy. Currently, the ISRF is defined such as to match the fit and smoothing procedures, effectively establishing a definition which is only valid for the presented approach. This compromises the ability to compare with other instrument calibrations and the applicability for future instruments.

Any local variation is interpreted as resulting from not further specified "laser artefects", which are smoothed out by a bi-variant polynomial fitting. This needs to be further justified, answering the following questions: - Are the laser artefacts repeatable ? Have measurements been repeated for some of these "bad data" ? - Do they occur at given angles and wavelengths (patterns) or are they randomly distributed ? - What is the likely instrumental root cause (e.g. speckle or wavelength instability ?) ? - Why are such instrumental effects absent in the "good data" ? Could these also be affected by "laser artefacts" The paper lacks a discussion of error sources of the new method including instrument effects (only fit residuals are considered). A true error analysis of the technique would involve a rigorous analysis of instrumental error sources, such as - SNR of the laser measurements - laser stability - speckle amplitude (of integration spheres, diffusers) - straylight correction efficiency - non-linearity (resp. knowledge thereof) - pixel-to-pixel cross talk variation

The methodology for evaluating the suitability of the approach is questionable. The only figure-of-merit considered is "good fit quality". The amplitude of fit residuals is interpreted as the accuracy of the method, compared with the requirement of 1% of the
ISRF peak. However, fit residuals only provide information about the consistency be-tween the mathematical model and the measurement. If the measurement is affected by systematic instrument error (say, from straylight), the fit "absorbs" this error into the estimated parameters. This does, however, not mean that the true ISRF was deter-mined more accurately. In fact, the authors identify "different treatment of straylight" as the likely cause of discrepancies between the ISRF for radiance and irradiance ports (although the correction technique and its accuracy is not presented), which in theory should be identical. But instead of interpreting this as a limitation, the two different straylight backgrounds are fitted by the model and both ISRFs are regarded as true ones. It is clear that instrument effects are fundamentally unavoidable (and obviously present here). They should be identified as such, and not "absorbed" into parameters of the model and declared part of the "true" ISRF.

Due to a poor (not further explained) laser performance, the irradiance measurements were used to compute key data , while the radiance fits classified as "good" were used for validation. Some validation results are reported, but only median values for selected columns, from which the conclusion is drawn, that radiance and irradiance ISRF are identical. This raises the question, if ISRF characterisation (which is typically a cost driver for imaging spectrometers), can be restricted to irradiance measurements only. This would greatly reduce effort (one wavelength scan versus $\sim$ 100 fpr each spatial sample) and cost. A discussion and quantitative analysis would greatly enhance the impact of the paper.

The paper includes a short section on in-flight calibration of the ISRF with Tropomi's on-board calibration system, comprising five tuneable laser diodes. This part of the paper has the potential to significantly raise its impact, as this aspect is relevant for several upcoming missions (see list above). However, this opportunity is missed as no comparison between on-board and external diffuser/laser ISRFs is provided. The authors merely the state, that "The ISRF measured with the diode lasers is in close agreement with the ISRF calibration data,. . .", without presenting any plot, table, or

statistics. Since data using the on-board calibration system were acquired and are available, it is strongly suggested to extend this part of the paper by providing quantitative comparison.

Finally, the authors do not give adequate credit to previous work. The reference list is rather short and limited to Tropomi-related publications. Tropomi/S5P not the first grating spectrometer for which extensive ISRF calibration has been performed, and also not the first covering the SWIR spectral range (e.g. SCIAMACHY, OCO-2). A more extensive discussion on previous work (actually needed for the introduction of a new method) should include a literature review covering the following missions (non-exhaustive list): - SCIAMACHY - GOME-2 - OMI (Dobber et al. 2004) - OCO-2 (Crisp et al., lasers used for ISRF calibration in SWIR and NIR) They should also mention upcoming missions for which a new technique may become relevant, like e.g. - Sentinel-4 - Sentinel-5 - FLEX - MicroCarb - future Copernicus mission for anthropogenic CO2 Also, the definition of ISRF and ISSF functions in the introduction, although accurate, needs to make reference to previous discussions in the literature (see detailed comments).

Overall, the draft paper in its current status cannot be considered for publication in a peer-reviewed scientific journal. However, as the topic under study is highly relevant for interpreting results from Tropomi/S5P, as well as for a wealth of future space-borne push-broom imaging spectrometers, they are encouraged to thoroughly rework the manuscript along the lines indicated in the above comments, and the more detailed ones below.

##############

Detailed comments

############# Abstract: From the abstract it is not clear what is the difference between "deriving" and ISRF, as proposed in this paper and applied to the SWIR band of the TROPOMI instrument on one hand, and "measuring an ISRF", which is reported to

be done for all bands. Please rephrase or add a sentence, clarifying the topic of the work presented here.

It is not clear from the abstract (and in large paerts of the paper), if a new generic technique for determining an ISRF is proposed, or if the it is simply reported how it has been done for the Tropomi instrument. If the main topic is the introduction of a new technique for ISRF derivation, which can be used for future instruments, then the abstract shall contain quantitative statements about the advantages of the new technique.

Will the proposed new method or be used to derive the in-flight ISRF ?

#############

1.) Introduction

L. 13: incomplete sentence '"The latter…" L. 14: "entrance slit"; also add that light from ground scene is collected by a common telescope (not mentioned)

L. 19: Why does CO "have to be measured" with this acc. ? If this is a scientific requirement, please clarify and reference it (e.g. from a study)

- add a sentence clarifying the term ISRF (in US literature instrument line shape) and how it is defined

- Better clarify the link between the requirement for XCH4 accuracy (1%) and the ISRF requirement. Is this error the only contributor to the budet ?

The discussion of the difference between ISRF and ISSF on p. 2 (L. 2-17) is important. It is basically paraphrasing a similar discussion in: J. Caron, et al.: THE CARBONSAT CANDIDATE MISSION: RADIOMETRIC AND SPECTRAL PERFORMANCES OVER SPATIALLY HETEROGENEOUS SCENES ICSO 2014 Tenerife, Canary Islands, Spain International Conference on Space Optics 7 - 10 October 2014 Please cite this paper in the introduction

- It shall be pointed out what is the role of the ISSF, which is composed of of ISRFs of neighbouring detector pixels. Is only the ISRF useful for

L. 27-28: Previously it has not been mentioned that the instrument also measures solar irradiance. Please clarify that in the introduction, so that the reader understands the difference in "irradiance ISRF" and "radiance ISRF".

"In the spectral dimension, about 4–5 points have significant signal. This is the spread function of the instrument for this wavelength.' If "This" refers to 4-5 points of signal, it must be replaced by "these" ? It shall be added, that an ISSF cannot be measured continuously, but only sampled, while the ISRF can be measured continuously.

Recommendation: Cite a publication about the SCIAMACHY instrument, which pioneered space based measurements in the SWIR spectral range. Also cite the NASA mission OCO-2, which also deployed tuneable lasers in ISRF calibration.

#############

GENERAL REMARK The introduction (and abstract) fail to motivate a new in the technique for ISRF determination, as is promised by the title. Why is a new approach needed ? Is the 1% accuracy requirement not reachable by previous techniques ? What is the basic idea of the new approach ? Please add a section motivating the new approach and its basic idea (and why it is only applied to the SWIR). #############

#############

2.) Calibration measurements

- Please provide a figure depicting the calibration setup (incl. OPO, integrating sphere, collimating optics, etc.). This would make it easier to follow the given description

- Please explain in the text why the "irradiance ISRF" is expected to be different from the "radiance ISRF" of the same spectrometer. Theoretically, the ISRF is a spectrometer property determined by the optics after the entrance slit. The only difference is the

pre-slit illumination via the Sun diffuser (which btw. is not mentioned in the instrument description given in the introduction). The cited paper (Tol et al. 2017) does not provide a sketch neither.

- The sequence of measurements is not clear from the sentence in L. 11-12. With 165 sec. of measurements at 10 Hz we have 1650 acquisitions, so 20 for each wavelength ? What is the spectral sampling of the calibration measurement (2 nm / 80 = 0.025 nm, equally spaced) ? Is the wavelength adjusted in steps or continuously (by temperature variation ?). Please clarify in the text.

- The wording "100 wavelength scans per manual wavelength setting...to cover the swath..." is somewhat confusing. Is the manual setting a coarse adjustment which is then scanned in finer steps ? Or is it that 100 swath positions are manually aligned and then scanned over wavelength ?

- 1.1 deg. of the FoV corresponds indicates that more then one detector pixel is illuminated (216 pixels / (108 deg / 1.1 deg) = 2.2 pixels). In the introduction it is stated that the ISRF is determined for individual pixels. How does image distortion (smile, frown) influence the ISRF measurements over >1 pixel ?

- P. 3, L. 14: "Calibration measurements via the radiance port and irradiance port have been performed to verify that they are identical." Were the irradiance and radiance ISRF measurements identical ? To what extent ? If this is reported later, indicate it in the text.

- P. 3; L. "...was performed up and down" -> better "...was performed with increasing and decreasing wavelength scan direction."

#############

3.1 ISRF shape

p. 3; L. 25: "block distribution" is not a commonly used expression, recommended to use the term "boxcar function" instead.

p. 3; l. 28: "In the end, the Pearson type VII distribution resulted in the best fit." What criteria determines what is the best fit ? In general, the criteria for the quality of the determined ISRF are not clearly defined at this stage.

Define or reference "Pearson type VII distribution"

- Please clarify why the particular representation of the ISRF was chosen. The idea of the Peak function is sketched (convolution of a perfect slit image with a function representing image blur by the optics. However, a Gaussian is not a perfect representation of the spectrometer PSF and the detector cross talk, and other functions may be used. Please comment on how the shape components were chosen (e.g. why a Pearson distribution). How robust are the results w.r.t. other representations ?

It is also not shown (nor stated) that eq. (2) and (3) represent a convolution of a skew-normal distribution with a boxcar function (which is suggested on p. ; L. 1-2). Please clarify.

#############

3.2 Data preparation

p. 4, l. 21: "The measurement data are corrected for. . .and stray light (irradiance only)" Why are the irradiance data straylight-corrected, and radiance data (apparently) not ? Generally, the straylight correction raises the question of definition of straylight, which is linked to the definition of the ISRF. In some ESA and NASA missions, straylight is defined by the spectral extent of the ISRF. E.g. for OCO-2, spectral straylight is defined as the light detected at wavelengths beyond 6 times of the FWHM of the ISRF. Inside this range it is part of the ISRF. The authors should highlight the relation between straylight and ISRF definition, and explain the definition used for Tropomi.

Question: Have the data been corrected for detector non-linearity as well ? This could be important for measurement of the far wings of the ISRF, where signal levels are low.

p. 4 l. 22: "Readouts from bad pixels are discarded in the analysis." The manuscript

often uses terms from requirement definitions, which may not be familiar to all readers. Please briefly clarify the term "bad pixels" (e.g. reduced spectral detection efficiency).

p. 4; l. 23-24: "Frames where the light source was off or very weak are discarded." Switched—off light source is not expected in s nominal calibration procedure. Specify how many data were lost or, if insignificant, remove sentence.

p. 4; l. 24-25: "In each remaining frame, the column with the maximum average signal is determined and the columns up 25 to 7 pixels from this peak column are selected, to include the faint signal of the tails."

Are 7 pixels sufficient to capture the "faint signal of the tails" ? The ISRF could be deliberately saturated to increase the sensitivity further out to the wings. The criteria should be the definition of the ISRF boundary (1% of the peak). Please comment (and add in text) that 7 pixels cover that range.

This short (5 lines) section should be merged with the next one, or even the next two, which can be combined in a Section "ISSF and ISRF fitting".

#############

3.3 ISSF fit

p. 5; l. 4: "The ISSF is assumed to be the mirrored version of an ISRF,..." With growing wavelength distance (and changing optical PFS), the measured ISSF should differ from a mirrored ISRF centered at c0. Deviations from the symmetry assumptions could potentially affect the far wings of the ISRF. Please comment on the assumption and justify be demonstrating negligible error. If possible, present data (plots) supporting the validity of the assumption.

p. 5; l. 4: "which can be modelled with the function AR(c;d,$-$s,w,$\eta$,$\gamma$,m,c0) using Eq. (1); only its skew parameter s has the opposite sign" The introduction of the function AR in this sentence is confusing. Just define it by R with reversed skew parameter to reflect the mirror shape w.r.t. the ISRF. Maybe even add the defining equation AR(s) =

[Figure]

R(-s).

p. 5; l. 5: "In each frame, the ISSF of an illuminated row is fitted to the ISRF shape to normalize the signals and to find the wavelength peak position expressed in pixel units (Fig. 1b)." The process of fitting the ISSF to the ISRF shape involves the estimation of the 8 parameters indicated in Eq. 1-4. In a previous sentence, the number of useful pixels of a measured ISSF frame is only about 5: p. 2; l. 9: "In the spectral dimension, about 4–5 points have significant signal." Therefore the inversion of an ISRF profile seems underdetermined. Please explain how the ISRF parameters can be still be estimated (assumptions on constant parameters ?).

p. 5; l. 18: "The square root of the fit variance is the rms value." This can be assumed to be known by the reader (recommended to be removed).

In general, the description of the fit procedure is rather sparse. It reports processing steps, which are not further justified, leaving the reader wondering why a certain step is taken in a particular way: For example: 1) "As the laser-wavelength scan is not regular, the ISRF data points are not on a regular grid. Therefore, the points in the scan range are collected in bins of 1/32 of a spectral pixel and a median is applied to the data points in each bin. Empty bins are discarded." Does the sampling (1/32) automatically follow from the non-regularity of the grid ? How irregular are the wavelength steps of the laser ? Why can't the fit be performed on a non-regular grid, when the functional shape and the relative wavelength are prescribed anyway ?

2) "The quality of the fit is determined by calculating the fit variance, the sum of the squared fit residuals where the fit function is larger than 6% of the maximum, divided by the number of degrees of freedom (number of points minus the free fit parameters)." Please explain (or reference) the fit quality parameter. In absence of explanation it appears to be an arbitrary choice (e.g.why 6% threshold).

Since a new method shall be introduced here (according to the title), please extend the discussion of the fit procedure, explaining and justifying all steps.

[Figure]

############

3.5 ISRF parameter smoothing

p. 5, l. 21: "However, the ISRF fits are valid locally (at location (r, c)) and not available for all pixels" A previous sentence seems to indicate that the fit is done for all pixels: p. 2, l. 24: "A description of the method and algorithm used to derive the ISRF for all illuminated pixels is presented in Sect. 3" Please clarify.

p. 5; l. 21-22: "It is expected that the fit parameters that define the local ISRF vary only smoothly over the surface of the detector as this is determined by the spectrometer optics. " -> This is only true if detector effects can be neglected. This assumes that all pixel-to-pixel effects (like PRNU, DRNU) are perfectly calibrated. Cross-talk effects, which determine the detector PSF (one component of the ISRF and therefore not eliminated by calibration), may vary from pixel to pixel. Please justify the assumption that the ISRF is determined by optics only. If possible, provide empiric evidence for constant cross-talk.

p. 5; l 25, Eq. 5: Similar to the previously described fit procedure, this particular smoothing function seems to "fall from the sky". Please provide justification for this particular function and how its parameetrs are determined (e.g. are the constants 255 and 999 the detector pixels in spatial and spectral dimension ?)

p.6; l. 1-3: "To obtain good results for the ISRF parameter fitting, obvious outliers in the individual ISRF-fit results should be rejected before the bivariate polynomial fit is performed. Given the distribution of outliers (in columns at the same wavelength), it is judged that most of them are caused by laser artefacts". - Typo: Replace "artifacts" by "artefacts"

- Comment: Each ISRF has been determined by a multitude of ISSF measurements at many wavelengths, which have been used to fit a composite shape function with relatively few parameters. Random laser effects (which ones?) should already be

smoothed out by this procedure. It is somewhat surprising, that these obtained parameters need to be further smoothing. Please elaborate (in the text) on: - the size and distribution of outliers - why it is judged that the outliers are caused by laser artefacts - the likely cause of "laser artefacts" (e.g. can better lasers improve the method ?)

p.6; l. 8: "unrealistic curve-fit solutions are rejected..." Please report the fraction of rejected fits.

p.6; l. 9-10: Please report the fraction of rejected fits. And again, please justify the numbers of the rejection filter (why rms <= 0.0065 and not any other number ?)

Question: What was the impact of "bad" and "dead" pixels in the procedure ? This may provide important guidelines for detector cosmetics requirements

p.6; l. 12: "all automated scans are performed twice: scanning up and down in wavelength. " Please report if a systematic difference was observed between the two scan directions (hysteresis effect).

p. 6; l. 16-17: "The irradiance data has a better coverage in both spectral and spatial directions,..." -> Why is this the case ? The difference between Sun and Earth ports is the diffuser before the slit. This may extend the illumination in the spatial direction (full slit illuminated), but not in the spectral. "...so a higher order M = 7 could be applied on the parameters d and s, which show much more structure than the other fit parameters." -> It would be useful to compare (in plots and by statistics) the variability of the different ISRF parameters/

p. 6; l. 21-25: "The quality of the parameter fitting is determined by comparing the measured ISRF data points with the ISRF that results from the parameter model." -> This implies that the measurements are independent from the chosen mathematical model of the ISRF. However, each "measurement point" is already the result of fitting the ISRF shape model to the measured ISSF. " In general, the parameter smoothing will result in better and smoother ISRF calibration key data due to averaging and interpolation" -> This seems to imply that "smoother" automatically means "better". However, exaggerated smoothing could systematically affect the accuracy of the derived ISRF shapes. Please comment on the chosen balance between smoothing and accuracy. Have measurements been repeated at outlier positions to verify they are due to random instrumental artefacts ? "Possibly counter intuitive, the rms value will be slightly larger as the ISRF data points are now compared with a smoothed ISRF instead of an optimized local ISRF that might be influenced by measurement imperfections." -> What does "slightly" mean ? Please provide numbers. This "counter-intuitive" observation actually may indicate exaggerated smoothing. Since the indivudual fits are closer to the truth, I would certainly expect the rms to increase.

Please report on the overall statistics of the smoothing procedure. What is the scatter of the original ISRF parameters around the smoothed value used in the key data ? Please provide a table for all ISRF parameters.

#############

3.6 ISRF parameter iteration

p. 6; l. 31: "block width" -> "boxcar width"

1) "Once the ISRF has been fitted, the skew and tails are known approximately, and can be included as fixed properties..." 2) "Therefore, the refitted block width as a function of row and column is smoothed and used as a fixed property in the final ISRF fit"

This section provides some explanation for the question raised above: How can 8 parameters be estimated from ISSF of 4-5 significant pixels? However, the approach of the "passes or stages" cannot solve an under-determined measurement problem. Parameters are estimated in stages, limiting the number of unknowns in each step to avoid under-determination. However, the results of each stage impact the ones of the next one (by fixing parameters estimated from an incomplete model). Each stage necessarily yields errors (incomplete description of then shape profile by fixing

parameters), and these are propagated into the next stage. In addition, smoothing seems to be applied between the 4 steps (see 2)), which also introduces systematic errors, propagated into the next step.

Please include a discussion, justifying the choice of the sequence, in which parameters are estimated.

p. 7; l. 3-8: How have the "realistic ISRFs" been simulated > Using the same parametric model ? This paragraph seems to describe a convergence test of the fit procedure for ideal data (no noise, no parametrisation errors, same ISRF). What are the start values chosen before stage 1 ? How far from the known true values can they be ? Please perform a test to demonstrate the convergence range of the technique.

p. 7; l. 9-14 The discrepancy of the derived ISRF from the known true one may indeed indicate the numerical problem highlighted above and result from the approach (underdetermined problem "solved" by fixing parameters and repeated iterations).

p. 7; l. 14: "However, the differences between the true ISRF and the derived ISRF are less than 0.25% and are considered acceptable" Acceptability could be stated if this were the maximum possible error. However, it seems to be assumed that the true ISRF is computed by Eq. 1-4, so it is consistent with the mathematical model and no modelling error is included. Please test the approach with ISRF profiles deviating from the chosen mathematical model to demonstrate robustness of the approach. Please also quantify the impact/sensitivity to measurement noise. This latter would give useful information on the required quality of the calibration system.

#############

4 Discussion of results

p. 7; l. 22: "A median has been taken over all rows illuminated. " What does this mean (a medium of what parameter) ?

p. 7; l. 22: "From visual inspection of the displayed ISRFs, one can conclude that:

(i) the ISRF is sharper and higher at higher column number (longer wavelength)" A large ($\sim$20% change) in ISRF width (2.3-2.7 pixels) should have been predicted by the optical design analysis. Is the magnification changing in spectral direction ? See also comment on Figure 3 and 4.

"(ii) the ISRF fit resembles the ISRF data very well, e.g. the residuals are very small, except where small artifacts can be identified in the ISRF data" Correct typo "artifacts". The log plots show significant discrepancy in the wings and the residual plots show periodic structures, whose peak-to-peak amplitude correspond to almost half of the requirement (1%). I would change "very well" to "satisfactory" to "compliant". Since a new method is proposed (according to the title), the question arises if it provides superior performance over previous calibration campaigns.

"(iii) the fit residuals of the irradiance ISRF are nearly a factor 2 smaller compared with the radiance ISRF." Please provide explanation.

p. 7; l. 27: "The difference is likely due to differences in stray light in these measurements." -> It was stated before that all measurements (both for radiance and irradiance) are based on straylight corrected data. Now straylight is identified as the cause for a discrepancy between radiance and irradiance ISRFs. This needs to be commented to avoid confuciotn. How accurate is the straylight correction ? Is the observed discrepancy (apparently averaged across the entire detector) explainable by the limitation of straylight correction. All this would be part of an accuracy analysis of the "new method", which is currently missing.

p. 7; l. 28: "In all subsequent fitting, shape parameter m is fixed to 1.25 to enhance convergence of the curve-fitting routine..." -> Why is this value fixed whereas all others are (partially, sequentially) fitted ? If it represents the "straylight level", as suggested in the text, why should it be constant ? How much do the fitted values of m vary and deviate from the median value ? In absence of this discussion this appears an arbitrary reduction of parameters for better convergence. Please comment and justify.
"It has to be noted that the contribution of the tail to the ISRF is small (< 10 %) and only significant 1-2 pixels away from the peak" -> This seems in contrast to studies analysing the impact of ISRF shapes on CH4 retrieval, for which ISRF far wings are very important.

"It has been verified that fixing the m parameter has negligible effect on the resulting ISRF and the fit residuals expressed in the rms value." Please provide evidence. Level-2 processing (not visual inspection) defines when an effect on the resulting ISRF is negligible. Has this been verified by retrieval simulations ?

p. 8; l. 10: "Block width w of the ISRF is determined by the projection of the slit onto the detector and therefore decreases as a function of wavelength." Again, please explain why a 20% reduction of slit width image is expected. In fact, the spectral sampling requirement (>2.5 pixel) seems to be violated across a wide spectral range.

p. 9; l. 19: "However, width parameter d has been designed such that no errors are introduced by the ISRF parameter fit." What does it mean to "design" a parameter ? Is it the choice of xi in Eq. 2 ?

p. 8; l. 28: "The quality of the ISRF fits as determined with the parameters from the bivariate parameter-fitting models shown in Fig. 6b." Typo, "is" is missing.

p. 8; l. 30: "There are a few small regions which coincide with the fine-scale structures visible in the skew-normal width, see for example around row 50 at columns 525 and 610." It is very difficult (if not impossible) to see the described fine-scale structures in Fig. 5 and 6 (a single row is not visible).

p. 9; l. 3-4: "In general, the laser performed worse during the radiance measurements, yielding radiance ISRF measurements of poorer quality than the irradiance measurements." Please provide details as to why and how much the laser has performed worse

p. 9; l. 13-14: "On the left side of the detector, the block width of the radiance ISRF tends to be smaller than that of the irradiance ISRF. This subtle difference is attributed

to the non-optimal scanning of the laser at these wavelengths."

Please indicate in which way the laser-scanning was non-optimal. Provide recommendation for optimum laser scanning.

Based on the poorer quality of the radiance fits, irradiance measurements are used for key data generation, while the radiance measurements merely serve for validation. What was the calibration time partition between irradiance and radiance measurements ? Noting that - significantly more time was spent on radiance (100 scans versus 1) and - the differences are stated to be negligible one conclusion could be that ISRF calibration can be reduced to measuring irradiance only. Please comment.

#############

5 In-flight Monitoring of ISRF

It is appreciated that a section on in-flight calibration is included in this paper. However, more detailed information shall be given here. As a minimum, the text should provide - type of laser diodes (Distributed Feedback (DFB) - their distribution over the SWIR range This is particularly important since the ISRF changes significantly over the spectral range - scan range in nm (not roughly pixels) - mention of a negligible laser bandwidth Also, reference to publications shall be made here, which describe the instrument design of Tropomi.

The comparison between ISRFs from the ISRF calibration campaign with external laser sources and measurements using the on-board lasers is of high interest in the context of future missions (e.g. Sentinel-5). Therefore the authors should elaborate on the quantitative comparison between the ISRFs derived form the two sources. The curent discussion is too qualitative and does not allow an evaluation of on-board ISRF monitoring. Adding a quantitative discussion (with plots) would significantly enhance the impact of this paper.

p. 9; l. 26-27: " The scanning range is about 6 spectral pixels so that the ISRF can

be monitored for one or two wavelength pixels per laser." This is confusing, since the term "wavelength pixels" is not defined, or at least not discriminated to "spectral pixels" in the same sentence. I assume the authors mean the spectral range corresponding to two FWHM of the ISRF. Please rephrase.

p. 9; l. 25-26 "The laser wavelengths are scanned by tuning the temperature of the laser using a built-in thermo-electric cooler" Built in what: the (DFB?) laser or the calibration unit? Please describe more precisely.

p. 9; l. 28: "As the diffuser is not moved during the measurements, there will be speckle." Inadequate wording for a science paper: - Replace "there will be speckle." by more precise formulation, e.g. "...the measurements will be affected by speckle patterns due to the coherent laser light." Also, the connection to "moving diffusers" might not be clear to every user. Please add a sentence like: "Such patterns can be reduced by moving (e.g. rotating) the detector during the acquisition. However, this is not foreseen for in-flight calibration due to mechanical contraints (e.g. micro-vibrations)."

p. 9; l. 29: "Most speckle is removed by taking the median of the data of all illuminated rows." This is not understood and needs more explanation: I assume that the illumination of the on-board diffuser illuminates the Is the median taken over all rows (swath direction) to yield only one ISRF for the entire focal plane ? Why the median and not the mean (affected by outliers) ? The latter makes more sense for reasons of energy conservation. Shouldn't the on-board calibration enable the determination of the ISRF across the entire swath width at 5 spectral positions ?

p. 9; l. 29: "During the commissioning phase, in-flight measurements with the on-board lasers will be performed with a moving and a fixed diffuser" Before it was mentioned that the on-board diffuser cannot be moved in flight. This may be different during commisioning phase, but deserves a sentence of explanation. Please provide details to improve clarity.

p. 9; l. 31: "The ISRF obtained from these measurements can be compared with the ISRF measured on ground using the external laser and the on-board diode lasers to detect any possible changes" - Remove "any", as this suggests infinite accuracy

p. 9; l. 31: "The monitoring ISRF is of sufficient quality to check for any degradation of the instrument but cannot be applied in trace-gas retrieval." - Remove "any", as this suggests infinite accuracy - Please explain why the on-board ISRF "cannot be applied in trace-gas retrieval". It may actually be useful to correct for launch effects and thermo-mechanical effects (de-focus). If such correction is made, it will be indirectly used in Level-2 processing.

In general, be more quantitative in the comparison and evaluation of the on-board ISRF measurements. What is the expected and obtained accuracy, and over which spectral range ?

p. 10; l. 3: "With an oscillating diffuser. . ." - Please explain what is meant by "oscillating" . I suppose that a calibration disk is moved back and forth by a few degrees (which is not quite oscillation), but the reader has to guess. Provide more details (see comment above).

p. 10; l. 4-5: ". . .except that ISRF parameter smoothing (Sect. 3.5) is calculated from the ISRF fits of the few columns scanned per diode laser." - Why is smoothing necessary here ? I would assume that the ISRF is determined for the five ISRFs corresponding to the center wavelengths of the diode laser scan ranges. The ISRF fit procedure probably takes the on-ground parameters as start values, so large outliers should not be expected.

p. 10; l. 4-5: "The column dependence of the shape parameters is neglected and the row dependence is smoothed by a second order polynomial." Does a square law (second order polynomial) decribe the variation of optical effects in swath direction ?

p. 10; l. 6-7: "Then the median ISRF is calculated from all ISRF data of the central

one/two fully-scanned columns, neglecting any row dependence." - replace "one/two" by "one to two" - Why is row dependence neglected (it has even been "smoothed" by a polynomial anyway) ? Why taking a median ISRF, not a mean ? Why are the 5 in-flight ISRFs not determined for all spatial samples across the swath ? The approach to in-fight ISRF characterisation appears somewhat immature.

p. 10; l. 9-10: Insert "the" between "moving" and "on-board"

p. 10; l. 10-11: "The ISRF measured with the diode lasers is in close agreement with the ISRF calibration data, thus proving the usability of the method and validating the calibration data." -> Be more quantitative here. No plot nor table is provided for this important comparison. Please plot the 5 ISRFs measured with the on-board diffuser together with the on-ground ISRF, corresponding to the same detector pixels. Perform the comparison for both, moved and stationary on-board diffuser to quantify the impact of speckle patterns.

p. 10; l. 12: "The monitoring ISRF deviates from the ISRF calibration data as could be expected." -> This is in contradiction to the sentence before.

p. 10; l. 12-13: "However, it is believed the method is sensitive enough to be used on board for long-term monitoring, being able to distinguish between changes in the real instrument ISRF and changes in the speckle pattern." -> Again, please provide a quantitative comparison, to substantiate this "believe".

Editorial: Straylight is written inconsistently in two ways: "stray-light" and "stray light". Any of them is fine (as well as one word), but be consistent.

#############

6 Conclusions

p. 10; l. 15-16: "A new and accurate method using a scanning OPO has been developed and applied to characterize the TROPOMI-SWIR ISRF. " -> Remove "accurate", as this is a qualitative statement, that's need to be substantiated (How accurate? More

accurate than other methods ?). In fact, its accuracy should be a results reported quantitatively here.

p. 10; l. 18-20: "An iterative scheme to derive the SWIR ISRF has been developed, where the ISRF determined in a previous iteration is used to improve the ISSF model in the current iteration. The required accuracy of the ISRF is obtained within 4 iterations." -> It shall added here that the "iterative scheme developed" is not estimating free parameters in every iteration (as would be expected for an over-determined problemS selectively fixing part of the parameters in every iteration is characteristic to the proposed new approach, so it has ot be repeated here (and the consequences as well).

p. 10; l. 18-20: "The ISRF measured through the irradiance port using the solar diffuser has been compared with the equivalent ISRF measured via the radiance port. The differences between the ISRFs derived from both data sets are very small,..." ->Please be quantitative here

"...and largely due to differences in stray-light treatment and laser scan imperfections." -> The statement that the discrepancy in ISRFs is "largely due to differences in stray-light treatment and laser scan imperfections", is an assertion, not a finding. It is suspected, but not demonstrated in this paper (it is even unclear, in which way straylight was treated differently).

p. 10; l. 23-25: "The derived ISRF meets the requirement on ISRF knowledge and should thus be sufficient for methane retrievals." The claim that the derived ISRF meets the requirement on ISRF knowledge is based only on the claim that the fit residuals are smaller than 1% of the ISF peak. However, this only means that the parameters of the chosen mathematical representation can tuned to match the observed shape. It does not mean that the observed shape is accurate. An example is the straylight, which apparently affects the measurements differently in the radiance and irradiance ports. Does it mean that the true ISRF of a system depends on the quality of straylight correction ? By fitting the straylight into the line shape (parameter m), it becomes a

feature of the true ISRF.

It is proposed to include a critical appraisal of the approach and results in the conclusions. This should outlined also the limitations of the approach. Accuracy (in terms of deviation from a true ISRF) shall clearly be distinguished from consistency between a fit and a measured curve.

##############

Figures and Tables

Table 1: Why is the parameters m kept at 1.25

Table 3: Reference (Beers et al.). Please explain why a a publication on "Measures of Location and Scale for Velocities in Clusters of Galaxies" is relevant, resp,. applicable to describing the ISRF variation.

Fig. A1 and A2: - Axis labels are missing ("Pixel No.") - Figure captions should be understandable without reading the text. Please extend the figure captions, briefly explaining the difference between "ISRF fit" and "ISRF parameter fit".

Comment: These plots indicate that there are systematic (not random) features being smoothed by the polynomial fit procedure, especially in the spectral dimension. Without evidence it is not obvious that they result from "laser artefacts". Speckle effects (not mentioned in the text) should affect the spatial component stronger due to smoothing by spectral dispersion.

The lower panel of Fig. A2 suggests that the "block width", representing the image width of the entrance slit, varies from 2.3 to 2.7 (pixels ?). This ∼20% change over the spectral range (for the entire swath) should be readily verifiable by optical analysis (diffraction and spot size PSF). Please check (and report) the plausibility of the result with the optical performance analysis.

Fig. 3 and 4: Reporting the parameter values in the caption is difficult to associate to

the 15 plots and does not provide useful information. Better include in them into the plot legends.

The fit residuals (bottom row) clearly exhibit systematic (periodic) structure. This indicates a shortcoming of the ISRF shape model, which does not allow for periodic components. Are there physical reasons why periodic components in the ISRF shape are ruled out ? Please comment on this and possibly propose an improvements.

- Please include plots showing the difference between radiance and irradiance ISRF and discuss the reasons for differences.

Fig. 5: Unclear Fig. caption: "In the white area, the ISRF fit failed (vertical stripes), the light is blocked by the entrance slit of the spectrometer (top and bottom) or a shield at the detector (left and right)." It does not seem logical, that an entrance slit blocks light. Improve clarity by adding "white area at the edge" or changing color. The term "white areas" is confusing with most of the middle panel being white (not only the edge). Proposed to change color scale.

Fig. 7: The number of "good" fits is drastically lower for radiance than for irradiance. Please provide explanation why this is the case.

Fig. A3 and A4: The plots show large variability of the resulting tail fraction and width from the ISRF fits. However, the ISRF parameter fit ("model") seems to assume a single value across the detector. Has this also been fixed to the median value (as parameter m) ? What is the justification, given the large, systematic variability ? Convergence ? It should be clarified (already in Section X) which parameter have been fixed to avoid the impression that the ISRF shape model has 8 free parameters.

#############

References Buscaglione: ESA-SRDs should have no author name

---

## Referee Comment (RC2) · Anonymous Referee #2 · 2 Feb 2018

############# General comments #############

The paper "New method to determine the instrument spectral response function, applied to TROPOMI-SWIR" by R. M. van Hees et al. addresses the determination of the Sentinel-5p / TROPOMI instrument spectral response function (ISRF) in the SWIR spectral region. The authors claim that the accuracy of the derived ISRF is well within the requirements for accurate trace-gas retrievals, which is stated to be known with an accuracy of 1% of its maximum where the ISRF is greater than 1% of its minimum.

The paper addresses an important topic, as accurate knowledge of the ISRF shape and FWHM is essential to avoid systematic errors in trace gas retrievals, especially for

missions with stringent requirements on spall systematic errors, e.g. greenhouse gas missions such as OCO-2/3 MicroCarb, GeoCarb, and the upcoming future European CO2 monitoring missions. The paper describes an iterative approach to accurately retrieve the ISRF shape from a series of measurements performed with an optical parametric oscillator (OPO) during the TROPOMI on ground spectral calibration measurements at CSL. The topic is in general of high interest to be published in AMT. However, my impression is, that this paper resembles in large parts a technical document or an ATBD describing the applied mathematical algorithm without explanation or deeper analysis of the applied steps. I agree in this regard with anonymous Referee #1 that this Manuscript should only be published in AMT after substantial revision. As a very comprehensive review is already given by Referee #1 addressing most of the issues I found in this paper, I will only briefly address some additional issues and my major points of concern.

To avoid further confusion, I will use in the following review the terms ISRF and ISSF as defined in this manuscript.

- The authors claims to introduce a new method for ISRF characterization without giving any evidence for the case. To underline the issue, the authors should perform a more comprehensive literature review on the topic and should cite for instance previous literature like K. Sun et al. 2017, R.A.M. Lee et al. 2017, J.O. Day et al. 2011, Beirle et al. 2017, Liu et al. 2015, Dirksen et al. 2006 and others.

- The authors are representing an iterative approach to derive the high resolution ISRF from a series of ISSF measurements, claiming that the high resolution ISRF could not be measured directly.

However this is only true, if the spectral accuracy, linewidth and "intensity" of the used optical stimulus (in this case the OPO) is insufficiently known. I have the impression, that this the case for the used OPO setup as the authors stated in two cases: "During the on-ground calibration measurements, the absolute wavelength of the source is not

measured accurately enough" and "The ISRF parameters cannot be retrieved directly from the measurements, because the wavelength and intensity of the signal are unknown and have to be determined via the ISSF". If this is the case, the authors should clearly state in the introduction section of the manuscript, that the iterative approach presented in this paper is required due the insufficient accuracy of the used spectral calibration equipment and is therefore the novelty of the described procedure. However a carefully design of the calibration stimulus should be able to overcome this problem and should allow a direct high resolution measurement of the ISRF for each detector pixel. Nevertheless, construction of the ISSF from such measurements could be tricky, as in addition, detector issues, as for instance differences in pixel to pixel crosstalk, PRNU etc. needed to be considered.

- The authors fail to justify, why for instance the Pearson VII distribution is used or why the given iterative approach is chosen. There is no comparison with other possible distributions, see for instance Beirle et al. 2017. Also the use of filter parameters is not sufficiently justified. For instance rms filtering of ISRF fits with an rms larger than 0.0065 is applied. Why not 0.005 or 0.008?

- The authors claim in the abstract and the conclusion that "The accuracy of the derived ISRF is well within the requirement for accurate trace-gas retrievals". However, the method described in this paper presents only a fit procedure able to fit the measurements with a given accuracy. The authors lack to provide a comprehensive error budget, including effects on PRNU, detector non linearity and other mostly detector related effects to underline that claim. The paper also lacks to provide an independent verification for that claim, see for instance Frankenberg et al. 2015 (doi:10.5194/amt-8-301-2015) for comparison.

- The use of the terms ISRF and ISSF as defined by the authors is confusing and not consisted with the paper of Hasekamp et al. 2016. cited for justification of the ISRF knowledge requirements. Typically the (derived) ISRF function is used to convolute a theoretical high resolution RTM spectrum to lower spectral resolution in the

retrieval. The (optical) ISRF is typically defined as the spectrometer response to a uniform monochromatic stimulus and approximated as the convolution of the slit image (typically represented by a boxcar function) with the PSF (or more accurately, the LSF) if the detector properties are neglected, see for comparison also Caron et al. 2017. This definition of the ISRF function is the mirror function of the ISRF function as defined in this paper, when assuming the changes of the (optical) ISRF are smooth over the image plane.

- Section 5 needs a deeper analysis to justify statements given in this section. For instance statements as: "However, it is believed the method is sensitive enough to be used on board for long-term monitoring, being able to distinguish between changes in the real instrument ISRF and changes in the speckle pattern" needs to be justified by analysis or removed.

############# Specific comments #############

Section 2, P.3 L.1-6: (calibration measurements): Is a Wavemeter and a (spectral response and linearity) calibrated monitoring detector used in the setup ? Is there any other type of direct laser wavelength monitoring integrated in the setup. If yes, what is the accuracy? What is the laser linewidth?

P.2 L.23-24: "using the wavelength assignment derived from an independent wavelength calibration measurement." - State how accurate the independent wavelength measurements are, as this has impact on the accuracy of the derived ISRF shape and how is it done? Is a different setup used?

P.3 L.25: convolution of block distribution should be exchanged by convolution with a boxcar function in the entire manuscript.

P.3 L.26: The optics is "blurring" the image by the spectrometer PSF (LSF) which could be asymmetric and often has also oscillations in the wings. Using a normal distribution for approximation of the PSF/LSF is only a first order approximation.

P.4 L.21: The measurement data is corrected for background, PRNU and straylight. Why is the data not corrected for detector non linearity as the used MCT detector can have nonlinearity in the % range over the dynamic range ?

P.5 L.2-3: Wavelength and intensity of the signal are unknown: are they really completely unknown? What is the wavelength accuracy/knowledge and stability of the used OPO ?

P.5 L.11-12: The laser wavelength scan is not regular -> why ?

P.5 L.17: larger than 6 % - > justify 6%, why not 5% or 7% ?

P.5 L.21-23: "It is expected that the fit parameters that define the local ISRF vary only smoothly over the surface of the detector as this is determined by the spectrometer optics. Therefore, a bivariate polynomial fitting is used to smooth and to interpolate the ISRF fit parameters." – This can only be expected for the optical system in case additional detector effects are neglected. However in a real life scenario, the effective ISRF is additionally compromised by insufficiently corrected detector effects. I guess, if all detector effects could be corrected to the required level, the resulting derived ISRF would be a smooth function. Therefore this statement contradicts the claim by the authors that the accuracy of the derived ISRF is well within the requirement for accurate trace-gas retrievals. Also the statement that most of the outliers are at same wavelengths and are caused by laser artefacts / scan imperfections contradicts that claim, as laser artefacts and scan imperfections need to be considered in the total error budget of the ISRF. See also discussion on P9. L.10-13. For a better understanding it would be helpful to show ISRF cross sections measurements from compromised rows, where the fit procedure fails.

P.9 L.3-7: A sketch of the radiance and irradiance calibration setup would be very helpful. Is the on-board diffuser illuminated via the integrating sphere during irradiance measurements or directly by the OPO ? I would expect from the text and as a ND filter is used in front of the OPO (P3,L10), that the OPO is directly illuminating the

on-board diffuser. This would result in a better SNR as more light is entering the instrument but also in more spectral structures introduced by the laser + on board diffuser combination. The authors state that the integration sphere in combination with the spinning mirror is used to avoid speckles. So please justify, why the irradiance measurement which obviously should have more spectral structures is used for the key-data. This furthermore triggers the discussion what is physically more meaningful, a smaller spread in the data as observed in figure 8 and also stated on P.9 L9-10 for the radiance measurements or a better RMS of the fit as observed for the irradiance measurements, which can for instance be caused by a larger number of fit parameters defining the degree of freedom of the fit.

P.9 L.14: How is it judged that the observed difference is attributed to the non-optimal scanning of the laser? What is an optimal scan? In fact, the wavelength accuracy and intensity of the laser are imperfectly known as previously stated.

P.9 L.15: Is a difference of the block width between radiance and irradiance measurements of $\sim 5$ % as shown in Fig 8 c for the left side of the detector really negligible in comparison to the requirements ?

P.9 L.29: Would taking the median over an entire row not imply the assumption, that there could not be a relative ISRF change along the row ?

P.10 L.1: The statement is contradicting to the previous statement. In fact, if the laser can be used to recalibrate the ISRF for a significant part of the detector, they can be used for trace gas retrievals.

P. 13 Figure 1 (a): it should be added to the axis caption that "column" is the detector spectral direction and wavelength is the wavelength derived for each ISSF measurement by the fitting procedure.

P.17 Figure 5: The figure caption should be clearer. For instance "white areas on top and bottom of the detector blocked by the slit for stray light correction and DC

monitoring" or something similar.

P.22 Table 3: How meaningful is the fit of the skew parameter s with > 100 % error ?

---

## Author Comment (AC1) · 24 May 2018

Our reply to the comments of Referee #1 is given in the supplement.

Please also note the supplement to this comment:
https://www.atmos-meas-tech-discuss.net/amt-2017-438/amt-2017-438-AC1-supplement.zip
———————————————————

---

## Author Comment (AC2) · 24 May 2018

Our reply to the comments of Referee #2 is given in the supplement.

Please also note the supplement to this comment:
https://www.atmos-meas-tech-discuss.net/amt-2017-438/amt-2017-438-AC2-supplement.zip

———————————————

---

## Author Response (AR1)

**1 *Author comment on* "New method to determine the instrument spectral response function, applied to TROPOMI-SWIR" *by Richard M. van Hees et al.*, manuscript amt-2017-438, Anonymous Referee #1**

We would like to thank Referee #1 for the very useful comments to improve our manuscript. In this document, we provide our reply to the comments. The original comments made by the referee are numbered and typeset in red. Page, line and figure numbers refer to the old version of the manuscript. After the reply we provide a revised version of the manuscript, each section is adjusted according to the review as described below. In this process, the original text has been substantially rewritten and/or reorganized. The output of 'latexdiff' is considered confusing and nevertheless provided. The revised version of the manuscript is part of this authors comment.

A brief overview of the major changes of the revised manuscript compared to the original manuscript:

- Title has been changed to "Determination of the TROPOMI-SWIR instrument spectral response function". We do no longer claim to present a new method.

- Clear distinction is made between what has been measured, what is determined from these measurements, and what is delivered. Inconsistent usage led to confusion.

- Thanks to the reviewer question to explain "laser artefacts", we discovered that:

  - a significant amount of irradiance measurements had to be rejected from the analysis due to an unexpectedly large variation of the laser signal yielding detector saturation.
  - we rejected radiance measurements from partly illuminated rows.

- The Pearson VII shape parameter $m$ was fixed to 1.25 in our analysis presented in the original manuscript, because then the shape of the model matched the far wings observed in TROPOMI SWIR stray light very well. However, simulations – requested by the reviewer – showed that this assumption introduces significant errors in case the true ISRF has wings with a different shape. A new approach has been developed and verified with simulations that requires two successive ISRF fits per stage: first only fixing the tail fraction (from the previous stage), then only fixing $w$ (improved guess from the previous ISRF fit). The method requires an initial guess of the tail fraction, which can be obtained from stray light or ISRF data (determined in stage 1).

- As a result of the above 2 points the analysis was improved such that the irradiance ISRF and radiance ISRF are more consistent.

- Analysis of the ISRF as determined from on-ground measurements with the on-board diode-lasers is included in the manuscript.

**1.1 General comments**

The paper addresses the determination of Instrument Spectral Response Functions (ISRF) of the recently launched Tropomi/Sentinel-5P mission. ISRF uncertainty is a notorious limitation of past and future space-borne atmospheric chemistry missions (e.g. GOME-2, OMI, Sentinel-4, Sentinel-5), as well as missions targeting greenhouses gases (OCO-2, MicroCarb, Sentinel mission). The SWIR band of the Tropomi/Sentinel-5P is used for retrieval of CO and CH4, where ISRF knowledge is most critical (together with the NIR band) because of strong, narrow absorption features. In this sense, the paper addresses a critical aspect of trace and greenhouse gas retrieval and therefore the topic is of high scientific relevance.

GC-i The introduction of a new method for ISRF characterisation by on-ground calibration measurements, as promised by the title, would be of high interest for planned future missions. However, the manuscript fails to deliver key elements for introducing and making a case for a new method. Neither do the authors describe other techniques, nor do they motivate the introduction of a new one, explain the difference, or compare it with previous calibration/validation measurements. If the objective of the paper is the introduction of novel method for ISRF determination, as suggested

by the title, the differences to previous instrument calibrations (e.g. SCIAMACHY, OMI, OCO-2) have to be pointed out. Ideally, a comparison shall be offered pointing to the advantages of the new approach.

**Adjusted.** We do no longer claim to present a new method, updated the title, abstract, section "Methodology" and conclusions accordingly. Citations have been added where relevant. A brief discussion on ISRF determination of several relevant other missions is added to the Introduction.

GC-ii In fact the reader is left wondering which are the new elements of the proposed approach: On the instrumental aspect: Is it the first time an OPO was used? If so, why is it expected to yield better performance (than e.g. monochromator)? What are the instrumental limitations of the approach? The impact of key parameters of the measurement data (e.g. signal-to-noise ratio) is not discussed. No details on the instrumental setup are provided, and the quoted reference does not contain them either. On the modelling aspect: Is the novelty of the approach the mathematical model for the ISRF in terms of a peak and a tail function? Then, why were these particular functions chosen and not any other? Are there physical reasons why the ISRF tails should follow a Pearson type VII distribution? All these questions are not addressed in the manuscript. In large parts, the text resembles an technical document (or ATBD), which describes a mathematical algorithm without explaining, why certain steps are taken.

**Adjusted.** The manuscript presents the determination of the TROPOMI-SWIR ISRF. All questions and remarks are addressed in the updated manuscript and in the detailed comments below.

GC-iii The fitting procedure to determine the ISRF seems to suffer from an under-determined equation system, although this is never mentioned. This is mitigated by "fixing" some parameters selectively in the inversion during four iterative steps. Again, the authors do not justify the presented sequence of partial fits, other than it results in "good fits" (defined by low residuals). This is particularly worrying as the approach is verified with two synthetic ISRF shapes, which were presumably computed using the same mathematical model as in the inversion (albeit this is not clear from the text). In theory, if the information content of the measurements would be sufficient to estimate all model parameters, this should result in perfect agreement between fit and forward model. However, the fit residuals clearly exhibit systematic features, indicating a weakness in the fit procedure. Nevertheless, the authors conclude "compliance" as the fit residual's amplitude is below the accuracy threshold (1% requirement).

**Adjusted.** We obviously have failed to present the method to determine the ISRF clearly, leaving the reader confused. Therefore, we have reworked our manuscript to cover these questions. The comment and questions are repeated as detailed comments below.

GC-iv The iterative fit procedure yields ISRF shapes for all detector pixels, which are not uniform and clearly show high-frequency variation. At this point the authors argue that the ISRF is only determined by the spectrometer optics (PSF), which varies smoothly with wavelength and swath angle. This assumption is not in line with the definition of ISRF used in other publications, and even with the introduction of the present manuscript. Defined as the spectral response of a single pixel as a function of wavelength, the ISRF includes the detector PSF (convolved with the slit boxcar and the optical PSF). The detector PSF is mainly driven by cross-talk, which may indeed introduce a pixel-to-pixel or column-to-column variation. While such systematic features are clearly visible in the estimated parameters across the detector (e.g. Fig. 5), the authors indirectly dismiss them as artefacts (by the above assumption) and eliminate them with an elaborate "smoothing procedure".

**Partly adjusted.** We only partly agree with the reviewer. By assuming that the ISRF is determined by spectrometer optics (PSF) –which should varies smoothly with wavelength and swath angle by design– we can identify detector PSF features. This is illustrated in Fig. 5b, where clear features of possible SWIR detector PSF are visible in the residuals (Fig. 8b in de revised manuscript). However, these are not the obvious column-to-column effects (the vertical stripes), present in Fig. 5, which are measurement artefacts, because these are absent (or at different locations) in other measurements (radiance and/or diode laser measurements). Variation of the detector PSF across the array is reflected in the light blue areas in the dark blue background on the right side of the detector. The same patterns are seen in the photo-sensitivity of the SWIR detector (Fig. 6 of Hoogeveen et al., 2013). An estimate of the error introduced by the

bivariate polynomial fit, which neglect detector PSF, is preseted in Fig. 15 in the revised manuscript. However, it can be shown that this error is small, see Sect. 4 "Discussion of results".

**GC-v** The smoothing equations (bi-variate polynomial fit) are reported, without justifying the choice of parameters (merely stating to yield "smoother and better" results). It is impossible to judge the impact (and therefore significance) of the smoothing, since no statistics are provided as of how much the individual parameter have been smoothed. The difference w.r.t the original fit results is not presented. Pixel-dependent effects (e.f. cross-talk) are "smoothed out" by this procedure, but presence of such effects does not mean that the individual ISRFs determined in the previous step were less accurate.

**Adjusted.** We agree with the reviewer that our description (in Sect. 3.5) is confusing and have updated the manuscript (Sect. 3.3) accordingly: ($i$) the need to smooth and interpolate the ISRF parameters is explained in the text. Figures on the difference between parameters of the local ISRF and the smoothed ISRF are added; ($ii$) the order of the bi-variate polynomial fit used, is determined by the variance of the residuals (= ISRF parameters - ISRF smoothed parameters). The lowest order is used for which the variance stabilizes, because then we can identify contribution by the detector PSF; ($iii$) we are aware that small local variations are neglected/ignored, any remaining residuals are checked by examining the results.

**GC-vi** The use of smoothing procedures gives the impression that the authors do not have much confidence in their technique. They regard the smoothed ISRF parameters as more accurate, and report the observation that the individual rms deviation from the measurements have increased as "counter-intuitive" (although seems to be expected).

**Adjusted.** It is certainly not our intention to leave the reader with this impression. This section has been rewritten, see also DC-36.

**GC-vii** It is understood that for practical purposes (operational Level-2 processing), calibration key data cannot take into account ISRF variations at the pixel-to-pixel level. However, the authors should clearly state where such compromises are made(e.g. between level of detail and computational resources) and justify them by quantifying or estimating the impact on Level-2 accuracy. Currently, the ISRF is defined such as to match the fit and smoothing procedures, effectively establishing a definition which is only valid for the presented approach. This compromises the ability to compare with other instrument calibrations and the applicability for future instruments.

**Adjusted.** We are confident that the delivered ISRF calibration key data are of high quality, based on the residuals seen and the simulations done with synthetic data. We have found no evidence for residual pixel-to-pixel effects, except for a correlation with photo-sensitivity of the SWIR detector, see discussion in Sect. 4. These residual pixel-to-pixel effects are small and can be ignored. Measurement artefacts have been identified by a comparison of the irradiance, radiance and diode-laser measurements. The improved results wrt the original manuscript, and the many more simulations done further underline the original conclusion.

**GC-viii** Any local variation is interpreted as resulting from not further specified laser artefacts, which are smoothed out by a bi-variant polynomial fitting. This needs to be further justified, answering the following questions:

- Are the laser artefacts repeatable? Have measurements been repeated for some of these "bad data"?
- Do they occur at given angles and wavelengths (patterns) or are they randomly distributed?
- What is the likely instrumental root cause (e.g. speckle or wavelength instability)?
- Why are such instrumental effects absent in the "good data"? Could these also be affected by "laser artefacts".

The paper lacks a discussion of error sources of the new method including instrument effects (only fit residuals are considered). A true error analysis of the technique would involve a rigorous analysis of instrumental error sources, such as: - SNR of the laser measurements; - laser stability; - speckle amplitude (of integration spheres, diffusers); - straylight correction efficiency; - non-linearity (resp. knowledge thereof); - pixel-to-pixel cross talk variation.

**Adjusted** in the revised paper. Answers to the questions listed in this comment: ($i$) Laser artifacts are explained, measurements affected by saturation due to too strong laser signals are rejected from analysis. Detailed data analysis was performed after the measurement campaign was finalized, denying the possibility to improve or redo certain measurements. However, measurement artefacts have been identified by a comparison of the irradiance, radiance and diode-laser measurements; ($ii$) Mostly random, hoewever, at some wavelength ranges the OPO was better behaved, compared to other ranges. Then measurement artefacts are present in the results of both radiance and irradiance measurements at nearly the same wavelength ranges; ($iii$) There is no evidence that speckle is present in the radiance and irradiance measurements. In the original manuscript, we use "laser artefacts" for too strong laser signals, wavelength instabilities and intensity instabilities; ($iv$) Yes, but only minor effects are seen in ISRF fits with a small rms of less than 0.003.

All detector calibrations, such as memory, offset, dark current, background signal, PRNU are corrected using the operational level 1b processor. Examples of "good data" affected by wavelength instabilities are presented in the revised paper.

GC-ix The methodology for evaluating the suitability of the approach is questionable. The only figure-of-merit considered is "good fit quality". The amplitude of fit residuals is interpreted as the accuracy of the method, compared with the requirement of 1% of the ISRF peak. However, fit residuals only provide information about the consistency between the mathematical model and the measurement. If the measurement is affected by systematic instrument error (say, from stray light), the fit absorbs this error into the estimated parameters. This does, however, not mean that the true ISRF was determined more accurately. In fact, the authors identify "different treatment of stray light" as the likely cause of discrepancies between the ISRF for radiance and irradiance ports (although the correction technique and its accuracy is not presented), which in theory should be identical. But instead of interpreting this as a limitation, the two different stray light backgrounds are fitted by the model and both ISRFs are regarded as true ones. It is clear that instrument effects are fundamentally unavoidable (and obviously present here). They should be identified as such, and not "absorbed" into parameters of the model and declared part of the "true" ISRF.

**Adjusted.** In the revised paper, we present the rms values, not a flag for good fit quality.

Stray light is corrected with the operational level 1b processor. In irradiance measurements, where the light is imaged as a vertical line, the ISRF at one row could be affected by stray light from other rows. Hence, these data are corrected for stray light. In radiance measurements, where the light is imaged as a spot, there is no stray light from other rows and the stray-light correction is not applied. This was intended with the remark "different treatment of stray light". The fact that radiance measurements are not corrected for stray light is mentioned in Sect. 3.2, page 4, line 21.

GC-x Due to a poor (not further explained) laser performance, the irradiance measurements were used to compute key data, while the radiance fits classified as "good" were used for validation. Some validation results are reported, but only median values for selected columns, from which the conclusion is drawn, that radiance and irradiance ISRF are identical. This raises the question, if ISRF characterisation (which is typically a cost driver for imaging spectrometers), can be restricted to irradiance measurements only. This would greatly reduce effort (one wavelength scan versus 100 fpr each spatial sample) and cost. A discussion and quantitative analysis would greatly enhance the impact of the paper.

**Adjusted.** This comment has been addressed in the revised paper. Laser artefacts are explained and examples provided. Furthermore, Local ISRF data and fits are presented, not medians.

For SWIR, the spectral features introduced by the on-board diffuser in irradiance data have much larger periods than the ISRF range, so they are constant per ISRF. Because they are absent in radiance data, the radiance measurements were intended to be used for the key data, but the irradiance measurements turned out to be better suited. The difference found between the irradiance and radiance ISRF is gone with our improved method; it was an artefact of the fixed tail parameter $m$.

To characterize the ISRF only with irradiance data indeed saves a lot of measurement time, but denies the possibility for a proper comparison between the two measurements. For example, confirmation on presence of detector effects.

GC-xi   The paper includes a short section on in-flight calibration of the ISRF with Tropomi's on-board calibration system, comprising five tuneable laser diodes. This part of the paper has the potential to significantly raise its impact, as this aspect is relevant for several upcoming missions (see list above). However, this opportunity is missed as no comparison between on-board and external diffuser/laser ISRFs is provided. The authors merely the state, that "The ISRF measured with the diode lasers is in close agreement with the ISRF calibration data,...", without presenting any plot, table, or statistics. Since data using the on-board calibration system were acquired and are available, it is strongly suggested to extend this part of the paper by providing quantitative comparison.

**Adjusted.** The on-ground measurements with the on-board diode lasers are included in the revised paper. However, the method for in-flight monitoring the ISRF stability will be presented in a separate paper after in-flight data has become available and has been thoroughly analysed.

GC-xii   Finally, the authors do not give adequate credit to previous work. The reference list is rather short and limited to Tropomi-related publications. Tropomi/S5P not the first grating spectrometer for which extensive ISRF calibration has been performed, and also not the first covering the SWIR spectral range (e.g. SCIAMACHY, OCO-2). A more extensive discussion on previous work(actually needed for the introduction of a new method) should include a literature review covering the following missions (non-exhaustive list): -SCIAMACHY -GOME-2 -OMI (Dobber et al. 2004) -OCO-2 (Crisp et al., lasers used for ISRF calibration in SWIR and NIR) They should also mention upcoming missions for which a new technique may be come relevant, like e.g. -Sentinel-4 -Sentinel-5 -FLEX -MicroCarb -future Copernicus mission for antropogenic CO2 Also, the definition of ISRF and ISSF functions in the introduction, although accurate, needs to make reference to previous discussions in the literature (see detailed comments).

**Adjusted.** Thank you, for providing literature suggestions. A better overview of the field has been added, together with the suggested citations, where relevant.

GC-xiii   Overall, the draft paper in its current status cannot be considered for publication in a peer-reviewed scientific journal. However, as the topic under study is highly relevant for interpreting results from Tropomi/S5P, as well as for a wealth of future space-borne push-broom imaging spectrometers, they are encouraged to thoroughly rework the manuscript along the lines indicated in the above comments, and the more detailed ones below.

We agree with this conclusion and thank the referee for the general and detailed comments. The manuscript has been revised along the suggested lines.

**Detailed comments**

DC-1   Abstract: from the abstract it is not clear what is the difference be-tween "deriving" an ISRF, as proposed in this paper and applied to the SWIR band of the TROPOMI instrument on one hand, and "measuring an ISRF", which is reported to be done for all bands. Please rephrase or add a sentence, clarifying the topic of the work presented here. It is not clear from the abstract (and in large parts of the paper), if a new generic technique for determining an ISRF is proposed, or if the it is simply reported how it has been done for the Tropomi instrument. If the main topic is the introduction of a new technique for ISRF derivation, which can be used for future instruments, then the abstract shall contain quantitative statements about the advantages of the new technique. Will the proposed new method or be used to derive the in-flight ISRF?

**Adjusted.** We rephrased the text in the manuscript (not only the abstract) to clearly distinguish what has been measured, what is determined from these measurements, and what is delivered. Inconsistent usage led to confusion.

No, the method is TROPOMI-SWIR specific. The abstract has been updated accordingly.

DC-2   P.1 L.13: incomplete sentence "The latter..."

**Adjusted** according to the suggestion.

DC-3   P.1 L.14: "entrance slit"; also add that light from ground scene is collected by a common telescope (not mentioned)

**Adjusted** according to the suggestion.

DC-4 P1 L.19: Why does CO "have to be measured" with this acc. ? If this is a scientific requirement, please clarify and reference it (e.g. from a study)

**Adjusted.** Replaced text by: "The TROPOMI-SWIR band is used for the retrieval of the atmospheric trace gases carbon-monoxide, methane and water vapor. Simulations have shown that in particular the methane retrieval is very sensitive to errors in the instrument spectral response function (ISRF or instrument line shape). As a result, the requirement on the ISRF is formulated that it should be known with an accuracy of 1% of its maximum where the ISRF is greater than 1% of its maximum (Hu et al., 2016)."

(a) Add a sentence clarifying the term ISRF (in US literature instrument line shape) and how it is defined

**Adjusted.** The term ISRF is defined in the introduction, here we also refer to the term "instrument line shape".

(b) Better clarify the link between the requirement for XCH4 accuracy (1%) and the ISRF requirement. Is this error the only contributor to the budget?

**Not adjusted.** This is outside the scope of this manuscript. Please consult Hu et al. (2016).

DC-5 P.2 L. 2-17: The discussion of the difference between ISRF and ISSF on is important. It is basically paraphrasing a similar discussion in: J. Caron, et al.: THE CARBONSAT CANDIDATE MISSION: RADIOMETRIC AND SPEC-TRAL PERFORMANCES OVER SPATIALLY HETEROGENEOUS SCENES ICSO 2014 Tenerife, CanaryIslands, Spain International Conference on Space Optics, 7–10 October 2014. Please cite this paper in the introduction

**Not adjusted.** The CarboSat paper focuses on the distortion of the ISRF in spatially heterogeneous scenes, an important subject, but it does not provide a clear discussion on the ISSF and ISRF.

DC-6 P.2 L.5: It shall be pointed out what is the role of the ISSF, which is composed of ISRFs of neighbouring detector pixels. Is only the ISRF useful for ...

**Adjusted.** The sentence is indeed confusing: we have measured many ISSF samples, to determine high resolution ISRF. Rephrased in the revised manuscript.

DC-7 P.2 L.27-28: Previously it has not been mentioned that the instrument also measures solar irradiance. Please clarify that in the introduction, so that the reader understands the difference in "irradiance ISRF" and "radiance ISRF".

**Adjusted.** Added the following text to the introduction, page 1, lines 14–16: "The TROPOMI instrument measures sunlight reflected by the surface and atmosphere of the Earth via the radiance port. Direct sunlight is measured via the irradiance port and internal diffuser for calibration purposes."

DC-8 P.2 L.9: "In the spectral dimension, about 4–5 points have significant signal. This is the spread function of the instrument for this wavelength". If "This" refers to 4–5 points of signal, it must be replaced by "these"? It shall be added, that an ISSF cannot be measured continuously, but only sampled, while the ISRF can be measured continuously.

**Partly adjusted.** Added to the following text, page 2, line 18: "In the spectral dimension, about 5 pixels have significant signal as expected from the spectral oversampling." It is already explained in the text and Figure 1 that "an ISSF cannot be measured continuously, but only sampled, while the ISRF can be measured continuously."

DC-9 P.2 L.9: Recommendation: Cite a publication about the SCIAMACHY instrument, which pioneered space based mea-surements in the SWIR spectral range. Also cite the NASA mission OCO-2, which also deployed tuneable lasers in ISRF calibration.

**Adjusted.** Differences and advantages with previous analysis of instrument ISRF have been added to the introduction, page 2 lines 6–13.

DC-10 **GENERAL REMARK:** The introduction(and abstract)fail to motivate a new in the technique for ISRF determination, as is promised by the title. Why is a new approach needed ? Is the 1% accuracy requirement not reachable by previous techniques? What is the basic idea of the new approach? Please add a section motivating the new approach and its basic idea (and why it is only applied to the SWIR).

**Adjusted.** The usage of measure, determine and derive have been reconsidered and we have aimed to used them consistently throughout the document. For example, we measure an ISSF, from which we determine the wavelength and signal intensity of the laser for this measurement.

We dropped the claim to present a new technique, because we only adopted existing techniques to our measurements and 1% accuracy requirement. Updated the Title, Abstract, Introduction and Conclusions accordingly.

DC-11 P.2 L.31: Calibration measurements: Please provide a figure depicting the calibration setup (incl. OPO, integrating sphere, collimating optics, etc.). This would make it easier to follow the given description

**Adjusted** according to the suggestion.

DC-12 P.3 L.8: 1.1 deg. of the FoV corresponds indicates that more then one detector pixel is illuminated (216 pixels/(108 deg/1.1 deg) = 2.2 pixels). In the introduction it is stated that the ISRF is determined for individual pixels. How does image distortion (smile, frown) influence the ISRF measurements over >1 pixel?

**Not adjusted.** The image distortion is subpixel in both spatial and spectral direction and can therefore be neglected locally.

DC-13 P.3 L.11-12: The sequence of measurements is not clear. With 165 sec. of measurements at 10Hz we have 1650 acquisitions, so 20 for each wavelength?

**Adjusted** rephrased the sentence, see answer to next comment

What is the spectral sampling of the calibration measurement (2nm/80 = 0.025 nm, equally spaced)?

**Adjusted** as explained in Sect. 2.1, see answer to next comment

Is the wavelength adjusted in steps or continuously(by temperature variation?). Please clarify in the text.

**Adjusted.** Each automated wavelength scan of about 2 nm, or 20 spectral pixels, took about 165 seconds. The data is collected at 10 Hz, yielding 1650 ISSF samples in total and about 80 samples per pixel. The laser scan speed was not constant, despite the small adjustments of the piezo voltage during the wavelength scan. Due to the large number of samples taken, this has no negative impact on the ISRF determination.

DC-14 P.3 L.14: Please explain in the text why the "irradiance ISRF" is expected to be different from the "radiance ISRF" of the same spectrometer. Theoretically, the ISRF is a spectrometer property determined by the optics after the entrance slit. The only difference is the pre-slit illumination via the Sun diffuser (which btw. is not mentioned in the instrument description given in the introduction). The cited paper (Tol et al.2017) does not provide a sketch neither.

**Adjusted.** rephrased text on page 2, lines 30–32: "The ISRF is determined from measurements using the radiance port and the irradiance port. Although differences between both data sets are not expected, they could not be ruled out. Light entering via the irradiance port follows a different path, via a diffuser."

DC-15 P.3 L.14: "Calibration measurements via the radiance port and irradiance port have been performed to verify that they are identical." Were the irradiance and radiance ISRF measurements identical? To what extent? If this is reported later, indicate it in the text.

**Adjusted.** The explanation is moved to the introduction.

DC-16 P.3 L.16: The wording "100 wavelength scans per manual wavelength setting... to cover the swath..." is somewhat confusing. Is the manual setting a coarse adjustment which is then scanned in finer steps? Or is it that 100 swath positions are manually aligned and then scanned over wavelength?

**Adjusted.** During the radiance measurements each automated wavelength scan (of 2 nm) had to be repeated for each swath position. At each swath position the wavelength was scanned in the opposite direction to save time.

DC-17 P.3 L.18: "...was performed up and down" -> better "...was performed with increasing and decreasing wavelength scan direction."

**Adjusted** according to the suggestion.

DC-18 P.3 L.25: "block distribution" is not a commonly used expression, recommended to use the term "boxcar function" instead.

**Adjusted.** We do no longer refer to the block distribution. However, in our opinion, the term 'boxcar function' is associated with time series and is not a distribution. Therefore, we prefer the term 'uniform distribution'.

DC-19 P.3 L.28: "In the end, the Pearson type VII distribution resulted in the best fit". What criteria determines what is the best fit? In general, the criteria for the quality of the determined ISRF are not clearly defined at this stage. Define or reference "Pearson type VII distribution"

**Adjusted.** A justification why the Pearson VII distribution was used is added to the section "Methodology". Several other possible distributions have been tested, including the super-gauss, an asymmetric version of the exponential power distribution (Beirle et al. 2017). They do not represent the TROPOMI-SWIR ISRF with enough precision. The asymmetry of the super-gauss is provided by combining separate functions for two halves of the peak, which results in a continuous curve, but a discontinuous derivative, leading to larger residuals in the peak. The tails of the super-gauss are far too steep. Pearson type VII is defined as Eq. 4 in the text.

DC-20 Please clarify why the particular representation of the ISRF was chosen. The idea of the Peak function is sketched (convolution of a perfect slit image with a function representing image blur by the optics. However, a Gaussian is not a perfect representation of the spectrometer PSF and the detector cross talk, and other functions maybe used. Please comment on how the shape components were chosen (e.g. why a Pearson distribution). How robust are the results w.r.t. other representations?

**Adjusted.** this comment is elaborated in Sect. 3.1.

It is also not shown (nor stated) that eq. (2) and (3) represent a convolution of a skew-normal distribution with a boxcar function (which is suggested on P.4 L.1-2). Please clarify.

**Adjusted.** this comment is elaborated in Sect. 3.1.

DC-21 P.4 L.21: "The measurement data are corrected for... and straylight (irradiance only)". Why are the irradiance data stray-light corrected, and radiance data (apparently) not? Generally, the stray-light correction raises the question of definition of straylight, which is linked to the definition of the ISRF. In some ESA and NASA missions, stray light is defined by the spectral extent of the ISRF. E.g. for OCO-2, spectral stray-light is defined as the light detected at wavelengths beyond 6 times of the FWHM of the ISRF. Inside this range it is part of the ISRF. The authors should highlight the relation between stray light and ISRF definition, and explain the definition used for Tropomi.

**Adjusted.** Added to the introduction: "For Tropomi, spectral stray-light is defined as the light detected at wavelengths beyond the equivalent of 4.5 pixels. Inside this range it is part of the ISRF." From Sect. 2.3: "In irradiance measurements, where the light is imaged as a vertical line, the ISRF at one row could be affected by stray light from other rows. Hence, these data are corrected for stray light using the operational processor. In radiance measurements, where the light is imaged as a spot, there is no stray light from other rows and the stray-light correction is not applied."

DC-22 P.4 L.21: Have the data been corrected for detector non-linearity as well? This could be important for measurement of the far wings of the ISRF, where signal levels are low.

**Adjusted.** From Sect. 2.3: A non-linearity correction is not implemented in the operational processor. It is also not needed for the ISRF characterization, as the error is small: detector non-linearity was measured to be about 0.1% to 0.2% (Hoogeveen et al., 2013).

**DC-23** P.4 L.22: "Readouts from bad pixels are discarded in the analysis." The manuscript often uses terms from requirement definitions, which may not be familiar to all readers. Please briefly clarify the term "bad pixels" (e.g. reduced spectral detection efficiency).

**Adjusted.** A description of bad/dead pixels is added tot Sect. 2.3.

**DC-24** P.4; L.23-24: "Frames where the light source was off or very weak are discarded." Switched off light source is not expected in nominal calibration procedure. Specify how many data were lost or, if insignificant, remove sentence.

**Adjusted.** Indeed confusing, we have removed sentence.

**DC-25** P.4; L.24-25: "In each remaining frame, the column with the maximum average signal is determined and the columns up to 7 pixels from this peak column are selected, to include the faint signal of the tails." Are 7 pixels sufficient to capture the faint signal of the tails? The ISRF could be deliberately saturated to increase the sensitivity further out to the wings. The criteria should be the definition of the ISRF boundary (1% of the peak). Please comment (and add in text) that 7 pixels cover that range.

**Adjusted.** Too detailed information, not relevant. This paragraph has been removed.

**DC-26** P.5 L.2–18: This short (5 lines) section should be merged with the next one, or even the next two, which can be combined in a Section "ISSF and ISRF fitting".

**Adjusted** according to the suggestion.

**DC-27** P.5 L.4: "The ISSF is assumed to be the mirrored version of an ISRF,..." With growing wavelength distance (and changing optical PFS), the measured ISSF should differ from a mirrored ISRF centered at c0. Deviations from the symmetry assumptions could potentially affect the far wings of the ISRF. Please comment on the assumption and justify be demonstrating negligible error. If possible, present data (plots) supporting the validity of the assumption.

**Adjusted.** Following text has been added to the introduction: "By design, the ISRF and dispersion of the TROPOMI instrument should vary only smoothly in the spectral and spatial dimension. This assumption will be validated in this study by determining the local ISRF for many pixels of the SWIR detector."

**DC-28** P.5 L.4: "which can be modelled with the function AR(c;d,-s,w,.,.,m,c0) using Eq. (1); only its skew parameter s has the opposite sign". The introduction of the function AR in this sentence is confusing. Just define it by R with reversed skew parameter to reflect the mirror shape w.r.t. the ISRF. Maybe even add the defining equation AR(s) = R(-s).

**Adjusted.** Intended is "$a$R() where the normalized function "R()" is defined in Eq.1 and $a$ is a scaling parameter. Text in the manuscript is updated accordingly.

**DC-29** P.5 L.5: "In each frame, the ISSF of an illuminated row is fitted to the ISRF shape to normalise the signals and to find the wavelength peak position expressed in pixel units (Fig. 1b)". The process of fitting the ISSF to the ISRF shape involves the estimation of the 8 parameters indicated in Eq. 1-4. In a previous sentence, the number of useful pixels of a measured ISSF frame is only about 5: P.2 L.9: "In the spectral dimension, about 4–5 points have significant signal". Therefore the inversion of an ISRF profile seems under determined. Please explain how the ISRF parameters can be still be estimated (assumptions on constant parameters?).

**Not adjusted.** Rephrased this paragraph, the ISSF is fitted with at most 4 free parameters (stage 1) and in later stages with only two parameters: wavelength and intensity of the signal. At most 6 parameters are fitted in the ISRF fit.

**DC-30** P.5 L.18: "The square root of the fit variance is the rms value." This can be assumed to be known by the reader (recommended to be removed).

**Adjusted.** A definition and explanation of "rms" as used in the revised paper: "An alternative measure of the fit quality turns out to be an rms value calculated as the square root of the sum of the squared difference between the ISRF fit residuals, using points where the fit function is larger than 6% of its maximum, divided by the number of ISRF data points minus the number of free fit parameters. The threshold of 6% is arbitrary, but a lower value would include more

of the tails where the residuals are always very small, which would make this measure less sensitive. The advantage of this measure is that it is less sensitive to small outliers, and sensitive to large outliers which can corrupt the fit procedure. Therefore, we use this rms as a measure of the fit quality."

DC-31    In general, the description of the fit procedure is rather sparse. It reports processing steps, which are not further justified, leaving the reader wondering why a certain step is taken in a particular way: For example: 1) "As the laser-wavelength scan is not regular, the ISRF data points are not on a regular grid. Therefore, the points in the scan range are collected in bins of 1/32 of a spectral pixel and a median is applied to the data points in each bin. Empty bins are discarded". Does the sampling (1/32) automatically follow from the non-regularity of the grid? How irregular are the wavelength steps of the laser? Why can not the fit be performed on a non-regular grid, when the functional shape and the relative wavelength are prescribed anyway?

**Adjusted.** We removed irrelevant information from the description of the method.

2) The quality of the fit is determined by calculating the fit variance, the sum of the squared fit residuals where the fit function is larger than 6% of the maximum, divided by the number of degrees of freedom (number of points minus the free fit parameters)". Please explain (or reference) the fit quality parameter. In absence of explanation it appears to be an arbitrary choice (e.g.why 6% threshold). Since a new method shall be introduced here (according to the title), please extend the discussion of the fit procedure, explaining and justifying all steps.

**Adjusted.** The description is extended in item DC-30.

DC-32    P.5 L.21: "However, the ISRF fits are valid locally (at location (r, c)) and not available for all pixels". A previous sentence seems to indicate that the fit is done for all pixels: P.2 L.24: "A description of the method and algorithm used to derive the ISRF for all illuminated pixels is presented in Sect. 3" Please clarify.

**Adjusted.** It always has been our intention to measure the ISRF for all pixels, however, we could not determine the ISRF for all pixels due to measurements with too strong laser signal, bad pixels and laser artefact. Therefore, it is necessary to interpolate the local ISRFs.

DC-33    P.5; L.21-22: "It is expected that the fit parameters that define the local ISRF vary only smoothly over the surface of the detector as this is determined by the spectrometer optics". This is only true if detector effects can be neglected. This assumes that all pixel-to-pixel effects (like PRNU, DRNU) are perfectly calibrated. Cross-talk effects, which determine the detector PSF (one component of the ISRF and therefore not eliminated by calibration), may vary from pixel-to-pixel. Please justify the assumption that the ISRF is determined by optics only. If possible, provide empiric evidence for constant cross-talk.

**Adjusted.** All detector calibrations are performed with the operational level 1b processor. We have found no evidence for residual pixel-to-pixel effects, except for a correlation with photo-sensitivity of the SWIR detector, see discussion in Sec. 4 of the revised manuscript.

DC-34    P.5 L.25, Eq. 5: Similar to the previously described fit procedure, this particular smoothing function seems to "fall from the sky". Please provide justification for this particular function and how its parameters are determined (e.g. are the constants 255 and 999 the detector pixels in spatial and spectral dimension?

**Adjusted.** The function is given in a simpler form (but it still represents exactly the same shape). We have added some explanation and mention that 255 and 999 are the detector pixels. Order 'M' is the maximum of the sum of the exponents of variables 'r' and 'c'.

DC-35    P.6 L.1-3: "To obtain good results for the ISRF parameter fitting, obvious outliers in the individual ISRF-fit results should be rejected before the bi-variate polynomial fit is performed. Given the distribution of outliers (in columns at the same wavelength), it is judged that most of them are caused by laser artefacts". Typo: Replace "artifacts" by "artefacts"

**Adjusted.** Thanks to the reviewers question to explain "laser artefacts", we discovered that:

- a significant amount of irradiance measurements had to be rejected from the analysis due to an unexpectedly large variation of the laser signal yielding detector saturation.

- we rejected radiance measurements from partly illuminated rows.

Note that "artifact" is not a typo, depends on the language (US, British).

DC-36 Comment: Each ISRF has been determined by a multitude of ISSF measurements at many wavelengths, which have been used to fit a composite shape function with relatively few parameters. Random laser effects (which ones?) should already be smoothed out by this procedure. It is somewhat surprising, that these obtained parameters need to be further smoothing. Please elaborate (in the text) on: -the size and distribution of outliers -why it is judged that the outliers are caused by laser artefacts. The likely cause of "laser artefacts" (e.g. can better lasers improve the method?)

**Adjusted.** Smoothing is needed to derive ISRF for every pixel. Minor problems with the OPO are still present in the ISRF data points (see Figs. 3–5), which may affect the ISRF fit yielding minor deviations of the fit parameters. Assuming that these deviations are random and that the shape of the ISRF varies only smoothly over the surface of the detector (as it is determined by the spectrometer optics), then the quality of the determined ISRF would benefit when the ISRF parameters are smoothed and interpolated using bivariate polynomial fitting.

Measurement artifacts are identified by obvious outliers: $s > 5$, nearby saturation (measurements of a wavelength scan where saturation occurred), comparison with other measurements e.g. radiance and/or diode laser measurements.

DC-37 P.6; L.8: "unrealistic curve-fit solutions are rejected..." Please report the fraction of rejected fits.

**Adjusted.** We provide these numbers in Sect. 4

DC-38 P.6; L.9-10: Please report the fraction of rejected fits. And again, please justify the numbers of the rejection filter (why rms <= 0.0065 and not any other number?) Question: What was the impact of "bad" and "dead" pixels in the procedure? This may provide important guidelines for detector cosmetics requirements.

**Adjusted.** The rms is arbitrary indicator, see DC-30

We do not have "dead" pixels and only 260 "bad" pixels.

DC-39 P.6 L.12: "all automated scans are performed twice: scanning up and down in wavelength". Please report if a systematic difference was observed between the two scan directions (hysteresis effect).

**Adjusted.** No hysteresis effect was found in the measurements.

DC-40 P. 6 L.16-17: "The irradiance data has a better coverage in both spectral and spatial directions,..." Why is this the case? The difference between Sun and Earth ports is the diffuser before the slit. This may extend the illumination in the spatial direction (full slit illuminated), but not in the spectral. "...so a higher order M=7 could be applied on the parameters d and s, which show much more structure than the other fit parameters.". It would be useful to compare(in plots and by statistics) the variability of the different ISRF parameters.

**Adjusted.** The radiance measurements were intended to be used for the determination of the calibration key data, but the irradiance measurements turned out to be better suited (Sect. 4). In the updated method, there is no difference between order $M$ used for radiance and irradiance measurements.

DC-41 P.6 L.21-25: "The quality of the parameter fitting is determined by comparing the measured ISRF data points with the ISRF that results from the parameter model". This implies that the measurements are independent from the chosen mathematical model of the ISRF. However, each "measurement point" is already the result of fitting the ISRF shape model to the measured ISSF. "In general, the parameter smoothing will result in better and smoother ISRF calibration key data due to averaging and interpolation". This seems to imply that "smoother" automatically means "better". However, exaggerated smoothing could systematically affect the accuracy of the derived ISRF shapes. Please comment on the chosen balance between smoothing and accuracy. Have measurements been repeated at outlier positions to verify they are due to random instrumental artefacts? "Possibly counter intuitive, the rms value will be slightly larger as the ISRF

data points are now compared with a smoothed ISRF instead of an optimized local ISRF that might be influenced by measurement imperfections". What does "slightly" mean? Please provide numbers. This "counter-intuitive" observation actually may indicate exaggerated smoothing. Since the individual fits are closer to the truth, I would certainly expect the rms to increase.

**Adjusted.** As stated in the revised manuscript (Sect. 3.3) "Minor problems with the OPO are still present in the ISRF data points (see Figs. 3–5), which may affect the ISRF fit yielding minor deviations of the fit parameters. Assuming that these deviations are random and that the shape of the ISRF varies only smoothly over the surface of the detector (as it is determined by the spectrometer optics), then the quality of the ISRF would benefit when the ISRF parameters are smoothed and interpolated using bivariate polynomial fitting". Smoother only means better, when the parameters of the true ISRF are smooth. Then a fit through noisy data will result in a better estimate of this parameter. The results presented in Sect. 4 show that small local deviations are present in the data.

DC-42  Please report on the overall statistics of the smoothing procedure. What is the scatter of the original ISRF parameters around the smoothed value used in the key data ? Please provide a table for all ISRF parameters.

**Adjusted.** Figures of the residuals of all ISRF parameters are provided in the revised manuscript

DC-43  P.6 L.31: "block width" -> "boxcar width"

**Adjusted.** We do no longer refer to the block distribution. However, in our opinion, the term 'boxcar function' is associated with time series and is not a distribution. Therefore, we prefer the term 'uniform distribution'.

DC-44  ISRF parameter iteration: This section provides some explanation for the question raised above: How can 8 parameters be estimated from ISSF of 4-5 significant pixels? 1)"Once the ISRF has been fitted, the skew and tails are known approximately, and can be included as fixed properties...". 2) "Therefore, the refitted block-width as a function of row and column is smoothed and used as a fixed property in the final ISRF fit". However, the approach of the "passes or stages" cannot solve an under-determined measurement problem. Parameters are estimated in stages, limiting the number of unknowns in each step to avoid under-determination. However, the results of each stage impact the ones of the next one (by fixing parameters estimated from an incomplete model). Each stage necessarily yields errors (incomplete description of then shape profile by fixing parameters), and these are propagated into the next stage. In addition, smoothing seems to be applied between the 4 steps (see 2)), which also introduces systematic errors, propagated into the next step. Please include a discussion, justifying the choice of the sequence, in which parameters are estimated.

**Adjusted.** We never fit the ISSF with many parameters, max 4. We use many ISSF fits to estimate the (local) ISRF, which can be determined with many (6-7) parameters. We do only fix one of the ISRF parameters ($\eta$ or $w$). The method converges as is shown with synthetic data, see Sect. 3.2 of the revised manuscript.

DC-45  P.7 L.3-8: How have the "realistic ISRFs" been simulated, using the same parametric model? This paragraph seems to describe a convergence test of the fit procedure for ideal data (no noise, no parameterization errors, same ISRF). What are the start values chosen before stage1? How far from the known true values can they be? Please perform a test to demonstrate the convergence range of the technique.

**Adjusted.** Thanks to this request, we have extended the simulations and updated the method presented in the revised paper (Sect. 3.2).

DC-46  P.7 L.9-14 The discrepancy of the derived ISRF from the known true one may indeed indicate the numerical problem highlighted above and result from the approach (under-determined problem "solved" by fixing parameters and repeated iterations).

**Not adjusted** In our opinion, there is no under-determination problem. It is not surprising that the determined ISRF does not always match the true ISRF perfectly, although the true ISRF is generated with the same ISRF model used for the fit. This is because details of the true ISRF are lost in the poorly sampled ISSF fit. According to an extra simulation, the determined ISRF matches the true ISRF nearly perfectly when the ISSF signal is measured with 10 instead of 5 spectral pixels.

**DC-47** P.7 L.14: "However, the differences between the true ISRF and the derived ISRF are less than 0.25% and are considered acceptable". Acceptability could be stated if this were the maximum possible error. However, it seems to be assumed that the true ISRF is computed by Eq. 1-4, so it is consistent with the mathematical model and no modelling error is included. Please test the approach with ISRF profiles deviating from the chosen mathematical model to demonstrate robustness of the approach. Please also quantify the impact/sensitivity to measurement noise. This latter would give useful information on the required quality of the calibration system.

**Adjusted.** Simulations with noisy data have been performed, and using a different mathematical model: the super-gauss (Beirle et al. 2017). The noise simulations were available for the original manuscript, but because the impact of noise is very small, it was not included in the discussion. Simulations with other values for tail parameter $m$ revealed the sensitivity of the method to fixing this parameter and we have updated our method, now fitting both tail parameters. Changing the mathematical model by using a different function for the main peak still shows that convergence is very good, but the residuals are slightly larger wrt our model and wrt to the truth. Which is no surprise, and therefore not included in the revised manuscript.

**DC-48** P.7 L.22: "A median has been taken over all rows illuminated." What does this mean (a medium of what parameter)?

**Adjusted.** A median has been taken of the ISRF data-points of all illuminated rows. And an ISRF fit is performed on these point.

**DC-49** P.7; L.22: "From visual inspection of the displayed ISRFs, one can conclude that: (i) the ISRF is sharper and higher at higher column number (longer wavelength)". A large (.20% change) in ISRF width (2.3 - 2.7 pixels) should have been predicted by the optical design analysis. Is the magnification changing in spectral direction? See also comment on Figure 3 and 4. "(ii) the ISRF fit resembles the ISRF data very well, e.g. the residuals are very small,except where small artifacts can be identified in the ISRF data". The log plots show significant discrepancy in the wings and the residual plots show periodic structures, whose peak-to-peak amplitude correspond to almost half of the requirement (1%). I would change "very well" to "satisfactory" to "compliant". Since a new method is proposed (according to the title), the question arises if it provides superior performance over previous calibration campaigns. "(iii) the fit residuals of the irradiance ISRF are nearly a factor 2 smaller compared with the radiance ISRF". Please provide explanation.

**Adjusted.** ($i$) This is a visual inspection, but on the next page first bullet is the explanation (original paper): "Block width w of the ISRF is determined by the projection of the slit onto the detector and therefore decreases as a function of wavelength. As expected, no variation is seen over the spatial dimension of the detector (swath angle)".

This article is not about the requirements on the instrument.

($ii$) The method is updated to perform a more accurate fit on the tails. Note that the significant discrepancy in the wings, observed by the reviewer, has no impact on the ISRF knowledge as defined by the requirement on the ISRF. It is agreed that we should have used "satisfactory" instead of "very well". With the updated method, the results have improved, therefore, in the revised manuscript we use "agrees well". (We never claimed to have a superior method, only a new method.)

($iii$) We only observe this and cannot explain it.

**DC-50** P.7 L.27: "The difference is likely due to differences in stray light in these measurements". It was stated before that all measurements (both for radiance and irradiance) are based on stray light corrected data. Now stray light is identified as the cause for a discrepancy between radiance and irradiance ISRFs. This needs to be commented to avoid confusion. How accurate is the stray light correction? Is the observed discrepancy (apparently averaged across the entire detector) explainable by the limitation of stray-light correction. All this would be part of an accuracy analysis of the "new method", which is currently missing.

**Adjusted.** The radiance measurements are not corrected for stray light, as is mentioned in Sect. 3.2, page 4, line 21. It is agreed that the quoted sentence is confusing and it is rephrased.

**DC-51**  P.7 L.28: "In all subsequent fitting, shape parameter m is fixed to 1.25 to enhance convergence of the curve-fitting routine...". Why is this value fixed whereas all others are (partially, sequentially) fitted? If it represents the "stray light level", as suggested in the text, why should it be constant? How much do the fitted values of m vary and deviate from the median value? In absence of this discussion this appears an arbitrary reduction of parameters for better convergence. Please comment and justify.

**Adjusted.** This is argued in the two sentences before the quoted text. And yes, this value of $m$ is rather arbitrary and the only parameter fixed in the ISRF fit.

We should have mentioned in the original manuscript, that for the computations to determine the TROPOMI-SWIR ISRF, the tail parameter m was fixed to 1.25, based on the shape of the tails found in stray-light measurements.

In the revised manuscript, we present the results from an updated iteration scheme where both tail parameters are fitted to the ISRF data.

**DC-52**  P.7 L.30: "It has to be noted that the contribution of the tail to the ISRF is small ($< 10\%$) and only significant 1-2 pixels away from the peak". This seems in contrast to studies analysing the impact of ISRF shapes on CH4 retrieval, for which ISRF far wings are very important.

**Not adjusted.** Indeed, far wings are important for methane retrieval. However, the far wings are part of the stray light, not ISRF.

**DC-53**  P.7 L.31: "It has been verified that fixing the m parameter has negligible effect on the resulting ISRF and the fit residuals expressed in the rms value". Please provide evidence. Level 2 processing (not visual inspection) defines when an effect on the resulting ISRF is negligible. Has this been verified by retrieval simulations?

**Not adjusted.** No, level 2 is outside the scope of article.

**DC-54**  P.8 L.10: "Block-width of the ISRF is determined by the projection of the slit on to the detector and therefore decreases as a function of wavelength. "Again, please explain why a 20% reduction of slit width image is expected. In fact, the spectral sampling requirement (>2.5 pixel) seems to be violated across a wide spectral range.

**Adjusted.** Updated the text in the revised document: "The peak width due to the projection of the slit on the detector is constant when expressed as a wavelength interval, but expressed as a column distance it decreases towards larger columns (longer wavelengths), because the spectral dispersion changes.". The spectral resolution is about 2.5 nm, which is consistent with the range we have determined of 2.3–2.7 nm.

**DC-55**  P.8; L.19: "However, width parameter has been designed such that no errors are introduced by the ISRF parameter fit". What does it mean to "design" a parameter? Is it the choice of xi in Eq. 2?

**Adjusted.** The skew-normal width $d$ is defined (not designed) such that no errors are introduced by the ISRF parameter fit. With sigma the interpolation of $s$ would introduce errors, however, with $d$ defined as a function of sigma and $s$ such errors are not introduced.

**DC-56**  P.8; L.28: "The quality of the ISRF fits as determined with the parameters from the bivariate parameter-fitting models shown in Fig.6b. "Typo, "is" is missing.

**Adjusted** according to suggestion.

**DC-57**  P.8 L.30: "There area few small regions which coincide with the fine-scale structures visible in the skew-normal width, see for example around row 50 at columns 525 and 610". It is very difficult (if not impossible) to see the described fine-scale structures in Fig. 5 and 6 (a single row is not visible).

**Adjusted.** We replaced "fine-scale structures" with "patches". There are a few small regions in Fig. 6b such around row 64 column 600 and which corresponding to similar regions visible in Fig. 5b. See also Fig. 6, Hoogeveen et al. 2013.

You should be able to see single pixels (rows and columns) in the figures of the revised manuscript.

DC-58   P.9 L.3-4: "In general, the laser performed worse during the radiance measurements, yielding radiance ISRF measurements of poorer quality than the irradiance measurements". Please provide details as to why and how much the laser has performed worse.

**Not adjusted.** This was a general impression.

DC-59   P.9 L.13-14: "On the left side of the detector, the block width of the radiance ISRF tends to be smaller than that of the irradiance ISRF. This subtle difference is attributed to the non-optimal scanning of the laser at these wavelengths." Please indicate in which way the laser-scanning was non-optimal. Provide recommendation for optimum laser scanning.

Based on the poorer quality of the radiance fits, irradiance measurements are used for key data generation, while the radiance measurements merely serve for validation. What was the calibration time partition between irradiance and radiance measurements? Noting that - significantly more time was spent on radiance (100 scans versus 1) and - the differences are stated to be negligible one conclusion could be that ISRF calibration can be reduced to measuring irradiance only. Please comment.

**Adjusted.** We think that it was a very good decision to perform radiance as well as irradiance measurements, see also GC-x. Only afterwards, we could conclude that the stray-light correction performs very well on the irradiance measurements and the different light path has no significant impact on the determined ISRF. The way the measurements were implemented was the most efficient way to collect the data. However, we had to reject a significant number of irradiance measurements due to potential saturation, and reject a significant number of radiance measurements with partly illuminated rows. Therefore, we recommend to reserve more time for testing of the laser before the actual measurements. And for the radiance measurements: a FOV to 2.5–3 degrees (about 4–5 swath pixels) would be optimal, and perform successive swath scans with about one swath pixel overlap.

DC-60   It is appreciated that a section on in-flight calibration is included in this paper. However, more detailed information shall be given here. As a minimum, the text should provide - type of laser diodes (Distributed Feedback (DFB) - their distribution over the SWIR range This is particularly important since the ISRF changes significantly over the spectral range - scan range in nm (not roughly pixels) - mention of a negligible laser bandwidth Also, reference to publications shall be made here, which describe the instrument design of Tropomi.

**Adjusted** according to the suggestions of the reviewer.

DC-61   The comparison between ISRFs from the ISRF calibration campaign with external laser sources and measurements using the on-board lasers is of high interest in the context of future missions (e.g. Sentinel-5). Therefore, the authors should elaborate on the quantitative comparison between the ISRFs derived form the two sources. The current discussion is too qualitative and does not allow an evaluation of on-board ISRF monitoring. Adding a quantitative discussion (with plots) would significantly enhance the impact of this paper.

**Adjusted.** The measurements with the on-board diode-lasers are included in the analysis presented in the revised manuscript.

DC-62   P.9 L.26-27: "The scanning range is about 6 spectral pixels so that the ISRF can be monitored for one or two wavelength pixels per laser." This is confusing, since the term "wavelength pixels" is not defined, or at least not discriminated to "spectral pixels" in the same sentence. I assume the authors mean the spectral range corresponding to two FWHM of the ISRF. Please rephrase.

**Adjusted** rephrased the sentence as suggested by the reviewer.

DC-63   P.9 L.25-26: "The laser wavelengths are scanned by tuning the temperature of the laser using a built-in thermo-electric cooler". Builtin what: the (DFB?) laser or the calibration unit? Please describe more precisely.

**Adjusted** rephrased the sentence as suggested by the reviewer. The TE cooler is inside the laser housing.

DC-64   P.9 L.28: "As the diffuser is not moved during the measurements, there will be speckle". In adequate wording for a science paper: replace "there will be speckle" by more precise formulation, e.g. "...the measurements will be affected

by speckle patterns due to the coherent laser light". Also, the connection to "moving diffusers" might not be clear to every user. Please add a sentence like: "Such patterns can be reduced by moving (e.g. rotating) the detector during the acquisition. However, this is not foreseen for in-flight calibration due to mechanical constraints (e.g. micro-vibrations)."

**Adjusted.** A monochromanic laser has speckle which can be suppressed by using a moving diffuser however, moving the diffuser is life limited, therefore, we take a median over the illuminated rows. The suggestion of reviewer is accepted with small modification.

DC-65   P.9 L.29: "Most speckle is removed by taking the median of the data of all illuminated rows". This is not understood and needs more explanation: I assume that the illumination of the on-board diffuser illuminates the Is the median taken over all rows (swath direction) to yield only one ISRF for the entire focal plane? Why the median and not the mean (affected by outliers)? The latter makes more sense for reasons of energy conservation. Should not the on-board calibration enable the determination of the ISRF across the entire swath width at 5spectral positions?

**Adjusted.** We have added some rationale here. Median is taken per laser. The data are already normalized, therefore a median can be used to reduce outliers (i.e. speckle).

DC-66   P.9 L.29: "During the commissioning phase, in-flight measurements with the on-board lasers will be performed with a moving and a fixed diffuser" Before it was mentioned that the on-board diffuser cannot be moved in flight. This may be different during commissioning phase, but deserves a sentence of explanation. Please provide details to improve clarity.

**Adjusted** according to the suggestion.

DC-67   P.9; L.31: "The ISRF obtained from these measurement scan be compared with the ISRF measured on-ground using the external laser and the on-board diode lasers to detect any possible changes". Remove "any", as this suggests infinite accuracy.

**Adjusted.** In this context, "any" means "it does not matter which type", not "every". The sentence is rephrased.

DC-68   P.9 L.31: "The monitoring ISRF is of sufficient quality to check for any degradation of the instrument but cannot be applied in trace-gas retrieval". Remove "any", as this suggests infinite accuracy. Please explain why the on-board ISRF can not be applied in tracegas retrieval. It may actually be useful to correct for launch effects and thermo-mechanical effects (de-focus). If such correction is made, it will be indirectly used in Level-2 processing. In general, be more quantitative in the comparison and evaluation of the on-board ISRF measurements. What is the expected and obtained accuracy, and over which spectral range?

**Adjusted.** The in-flight measurements with the diode-lasers using a fixed diffuser are intended for monitoring of ISRF stability. Details on the monitoring ISRF will be provided in a separate paper. An improved description of the monitoring ISRF is added to the revised manuscript, see 'Introduction' and 'In-flight monitoring of ISRF' (Sec. 5).

DC-69   P.10 L.3: "With an oscillating diffuser...". Please explain what is meant by "oscillating". I suppose that a calibration disk is moved back and forth by a few degrees (which is not quite oscillation), but the reader has to guess. Provide more details (see comment above).

**Adjusted** according to the suggestion.

DC-70   P.10 L.4-5: "...except that ISRF parameter smoothing (Sect. 3.5) is calculated from the ISRF fits of the few columns scanned per diode laser". Why is smoothing necessary here? I would assume that the ISRF is determined for the five ISRFs corresponding to the center wavelengths of the diode laser scan ranges. The ISRF fit procedure probably takes the on-ground parameters as start values, so large outliers should not be expected.

**Adjusted.** We can confirm that smoothing of the ISRF parameters is not needed for diode laser measurements with a moving diffuser. Therefore, no smoothing is used on these measurements presented in the revised manuscripts.

DC-71   P.10 L.4-5: "The column dependence of the shape parameters is neglected and the row dependence is smoothed by a second order polynomial". Does a square law (second order polynomial) describe the variation of optical effects in swath direction?

**Adjusted.** Yes, it is correct that the ISRF shape changes across the spatial dimension. However, these measurements are intended for monitoring of ISRF stability (see also DC-68).

DC-72 P.10 L.6-7: "Then the median ISRF is calculated from all ISRF data of the central one/two fully-scanned columns, neglecting any row dependence". Replace "one/two" by "one to two". Why is row dependence neglected (it has even been "smoothed" by a polynomial anyway)? Why taking a median ISRF, not a mean? Why are the 5 in-flight ISRFs not determined for all spatial samples across the swath? The approach to in-flight ISRF characterisation appears somewhat immature.

**Partly adjusted.** Please understand that we are discussing monitoring ISRF, not in-flight ISRF characterisation. Replaced by "one or two".

DC-73 P.10 L.9-10: Insert "the" between "moving" and "on-board".

**Adjusted** according to the suggestion.

DC-74 P.10 L.10-11: "The ISRF measured with the diode lasers is in close agreement with the ISRF calibration data, thus proving the usability of the method and validating the calibration data". Be more quantitative here. No plot nor table is provided for this important comparison. Please plot the 5 ISRFs measured with the on-board diffuser together with the on-ground ISRF, corresponding to the same detector pixels. Perform the comparison for both, moved and stationary on-board diffuser to quantify the impact of speckle patterns.

**Adjusted.** We have added the suggested figures. More information will be provided in the upcoming in-flight paper.

DC-75 P.10 L.12: "The monitoring ISRF deviates from the ISRF calibration data as could be expected". This is in contradiction to the sentence before.

**Not adjusted.** The definition of "monitoring ISRF" has to be taken into account. It is given at the end of page 9, end of line 30: the currently planned in-flight measurements are only suitable for monitoring the ISRF, because for deriving the ISRF a moving diffuser and longer spectral scans are required.

DC-76 P.10 L.12-13: "However, it is believed the method is sensitive enough to be used on board for long-term monitoring, being able to distinguish between changes in the real instrument ISRF and changes in the speckle pattern". Again, please provide a quantitative comparison, to substantiate this "believe".

**Not adjusted.** Details on the monitoring ISRF will be provided in a separate paper.

DC-77 Editorial: Straylight is written inconsistently in two ways: "stray-light" and "stray light". Any of them is fine (as well as one word), but be consistent.

**Not adjusted.** As a compound noun (the term on its own), we use "stray light". When used as a compound adjective, a hyphen needs to be added, e.g. "stray-light measurement".

DC-78 P.10 L.15-16: "A new and accurate method using a scanning OPO has been developed and applied to characterize the TROPOMI-SWIR ISRF". Remove "accurate", as this is a qualitative statement, that's need to be substantiated (How accurate? More accurate than other methods ?). In fact, its accuracy should be a results reported quantitatively here.

**Adjusted.** This sentence is removed, we no longer present a new method.

DC-79 P.10 L.18-20: "An iterative scheme to derive the SWIR ISRF has been developed, where the ISRF determined in a previous iteration is used to improve the ISSF model in the current iteration. The required accuracy of the ISRF is obtained within 4 iterations". -> It shall added here that the "iterative scheme developed" is not estimating free parameters in every iteration (as would be expected for an over-determined problem selectively fixing part of the parameters in every iteration is characteristic to the proposed new approach, so it has to be repeated here (and the consequences as well).

**Adjusted.** Rephrased the conclusions according to the reviewers comment.

DC-80 P.10 L.18-20: "The ISRF measured through the irradiance port using the solar diffuser has been compared with the equivalent ISRF measured via the radiance port. The differences between the ISRFs derived from both data sets are very small,...". Please be quantitative here "...and largely due to differences in stray-light treatment and laser scan imperfections". The statement that the discrepancy in ISRFs is "largely due to differences in stray-light treatment and laser scan imperfections", is an assertion, not a finding. It is suspected, but not demonstrated in this paper (it is even unclear, in which way stray light was treated differently).

**Adjusted.** The difference found between the irradiance and radiance ISRF is nearly gone with our improved method. It was an artefact of the fixed tail parameter m. Updated the conclusions accordingly.

DC-81 P.10 L.23-25: "The derived ISRF meets the requirement on ISRF knowledge and should thus be sufficient for methane retrievals". The claim that the derived ISRF meets the requirement on ISRF knowledge is based only on the claim that the fit residuals are smaller than 1% of the ISF peak. However, this only means that the parameters of the chosen mathematical representation can tuned to match the observed shape. It does not mean that the observed shape is accurate. An example is the stray light, which apparently affects the measurements differently in the radiance and irradiance ports. Does it mean that the true ISRF of a system depends on the quality of stray-light correction? By fitting the stray light into the line shape (parameter m), it becomes a feature of the true ISRF. It is proposed to include a critical appraisal of the approach and results in the conclusions. This should outlined also the limitations of the approach. Accuracy (in terms of deviation from a true ISRF) shall clearly be distinguished from consistency between a fit and a measured curve.

**Partly adjusted.** We are confident that this claim is valid, based on the residuals seen and the simulations done with synthetic data. The improved results wrt the original manuscript, and the many more simulations done only underline the original conclusion. True validation will be performed in-flight by level-2 processing.

As already explained and mentioned in the original manuscript, the radiance measurements are not corrected for stray light, while the irradiance measurements have to be corrected for stray light (as there is hardly any relevant stray light). Therefore, stray light is treated differently, and result in nearly equal residual stray light in both data sets. It is shown in the revised manuscript that the ISRF derived from both data sets are nearly equal.

**1.2 Figures and Tables**

1. Table 1: Why is the parameters m kept at 1.25

   **Adjusted.** Obsolete now that there is an updated iteration scheme, see also DC-51.

2. Table 3: Reference (Beers et al.). Please explain why a a publication on "Measures of Location and Scale for Velocities in Clusters of Galaxies" is relevant, resp,. applicable to describing the ISRF variation.

   **Adjusted.** This table is obsolete. A reference has been chosen that is easily available and that not only defines these general measures of location and scale, but also describes them well. The application for which the article describes them is indeed not relevant.

3. Fig. 3 and 4: Reporting the parameter values in the caption is difficult to associate to the 15 plots and does not provide useful information. Better include in them into the plot legends.The fit residuals (bottom row) clearly exhibit systematic (periodic) structure. This indicates a shortcoming of the ISRF shape model, which does not allow for periodic components. Are there physical reasons why periodic components in the ISRF shape are ruled out? Please comment on this and possibly propose an improvements. - Please include plots showing the difference between radiance and irradiance ISRF and discuss the reasons for differences.

   **Adjusted.** We agree with the reviewer and different plots are provided. Residuals of a non-perfect fit will always result in periodic residuals (and noise). The residuals in these figures are small, thus the model agrees satisfactorily with the true ISRF.

4. Fig. 5: Unclear Fig. caption: "In the white area, the ISRF fit failed (vertical stripes), the light is blocked by the entrance slit of the spectrometer (top and bottom) or a shield at the detector (left and right)" It does not seem logical, that an

entrance slit blocks light. Improve clarity by adding "white area at the edge" or changing color. The term "white areas" is confusing with most of the middle panel being white (not only the edge). Proposed to change color scale.

**Adjusted** according to suggestion.

5. Fig. 7: The number of "good" fits is drastically lower for radiance than for irradiance. Please provide explanation why this is the case.

**Adjusted.** An explanation has been added to the caption.

6. Fig. A1 and A2: - Axis labels are missing ("Pixel No.") - Figure captions should be understandable without reading the text. Please extend the Figure captions, briefly explaining the difference between "ISRF fit" and "ISRF parameter fit". Comment: These plots indicate that there are systematic (not random) features being smoothed by the polynomial fit procedure, especially in the spectral dimension. Without evidence it is not obvious that they result from "laser artefacts". Speckle effects (not mentioned in the text) should affect the spatial component stronger due to smoothing by spectral dispersion.

**Adjusted.** All captions have been updated and additional explanation has been added.

7. The lower panel of Fig. A2 suggests that the "block width", representing the image width of the entrance slit, varies from 2.3 to 2.7 (pixels ?). This 20% change over the spectral range(for the entire swath) should be readily verifiable by optical analysis (diffraction and spot size PSF). Please check(and report) the plausibility of the result with the optical performance analysis.

**Adjusted.** Yes, this is consistent with the design as suggested by the reviewer.

8. Fig. A3 and A4: The plots show large variability of the resulting tail fraction and width from the ISRF fits. However, the ISRF parameter fit ("model") seems to assume a single value across the detector. Has this also been fixed to the median value (as parameter m)?

What is the justification, given the large, systematic variability? Convergence? It should be clarified (already in Section X) which parameter have been fixed to avoid the impression that the ISRF shape model has 8 free parameters.

**Adjusted.** We thought that it was clear that a fixed value for the tail parameter $m$ was used for all results presented in the original paper. This discussion is no longer relevant for the revised manuscript.

**1.3 References**

Buscaglione: ESA-SRDs should have no author name on Atmos. Meas.Tech. Discuss., doi:10.5194/amt-2017-438, 2017.

**Adjusted.** we do no longer refer to the SRD.

**1  *Author comment on* "New method to determine the instrument spectral response function, applied to TROPOMI-SWIR" *by Richard M. van Hees et al.*, manuscript amt-2017-438, Anonymous Referee #2**

We would like to thank Referee #2 for the very useful comments to improve our manuscript. In this document, we provide our reply to the comments. The original comments made by the referee are numbered and typeset in red. Page, line and figure numbers refer to the old version of the manuscript. After the reply we provide a revised version of the manuscript, each section is adjusted according to the review as described below. In this process, the original text has been substantially rewritten and/or reorganized. The output of 'latexdiff' is considered confusing and nevertheless provided. The revised version of the manuscript is part of this authors comment.

A brief overview of the major changes of the revised manuscript compared to the original manuscript:

– Title has been changed to "Determination of the TROPOMI-SWIR instrument spectral response function". We do no longer claim to present a new method.

– Clear distinction is made between what has been measured, what is determined from these measurements, and what is delivered. Inconsistent usage led to confusion.

– Thanks to the reviewer question to explain "laser artefacts", we discovered that:

  – a significant amount of irradiance measurements had to be rejected from the analysis due to an unexpectedly large variation of the laser signal yielding detector saturation.

  – we rejected radiance measurements from partly illuminated rows.

– The Pearson VII shape parameter $m$ was fixed to 1.25 in our analysis presented in the original manuscript, because then the shape of the model matched the far wings observed in TROPOMI SWIR stray light very well. However, simulations – requested by the reviewer – showed that this assumption introduces significant errors in case the true ISRF has wings with a different shape. A new approach has been developed and verified with simulations that requires two successive ISRF fits per stage: first only fixing the tail fraction (from the previous stage), then only fixing $w$ (improved guess from the previous ISRF fit). The method requires an initial guess of the tail fraction, which can be obtained from stray light or ISRF data (determined in stage 1).

– As a result of the above 2 points the analysis was improved such that the irradiance ISRF and radiance ISRF are more consistent.

– Analysis of the ISRF as determined from on-ground measurements with the on-board diode-lasers is included in the manuscript.

**1.1  General comments**

The paper "New method to determine the instrument spectral response function, applied to TROPOMI-SWIR" by R. M. van Hees et al. addresses the determination of the Sentinel-5p/TROPOMI instrument spectral response function (ISRF) in the SWIR spectral region. The authors claim that the accuracy of the derived ISRF is well within the requirements for accurate trace-gas retrievals, which is stated to be known with an accuracy of 1% of its maximum where the ISRF is greater than 1% of its minimum.

The paper addresses an important topic, as accurate knowledge of the ISRF shape and FWHM is essential to avoid systematic errors in trace-gas retrievals, especially for missions with stringent requirements on small systematic errors, e.g. greenhouse gas missions such as OCO-2/3 MicroCarb, GeoCarb, and the upcoming future European CO2 monitoring missions. The paper describes an iterative approach to accurately retrieve the ISRF shape from a series of measurements performed with an optical parametric oscillator (OPO) during the TROPOMI on-ground spectral calibration measurements at CSL. The topic is in general of high interest to be published in AMT. However, my impression is, that this paper resembles in large parts a technical document or an ATBD describing the applied mathematical algorithm without explanation or deeper analysis of the applied

steps. I agree in this regard with anonymous Referee #1 that this Manuscript should only be published in AMT after substantial revision. As a very comprehensive review is already given by Referee #1 addressing most of the issues I found in this paper, I will only briefly address some additional issues and my major points of concern.

To avoid further confusion, I will use in the following review the terms ISRF and ISSF as defined in this manuscript.

GC-i    The authors claims to introduce a new method for ISRF characterization without giving any evidence for the case. To underline the issue, the authors should perform a more comprehensive literature review on the topic and should cite for instance previous literature like K. Sun et al. 2017, R.A.M. Lee et al. 2017, J.O. Day et al. 2011, Beirle et al. 2017, Liu et al. 2015, Dirksen et al. 2006 and others.

**Adjusted.** We do no longer claim to present a new method, updated the title, abstract, section "Methodology" and conclusions accordingly. Thank you, for providing literature suggestions, citations have been added where relevant.

GC-ii   The authors are representing an iterative approach to derive the high resolution ISRF from a series of ISSF measurements, claiming that the high resolution ISRF could not be measured directly.
        However this is only true, if the spectral accuracy, line-width and "intensity" of the used optical stimulus (in this case the OPO) is insufficiently known. I have the impression, that this the case for the used OPO setup as the authors stated in two cases: "During the on-ground calibration measurements, the absolute wavelength of the source is not measured accurately enough" and "The ISRF parameters cannot be retrieved directly from the measurements, because the wavelength and intensity of the signal are unknown and have to be determined via the ISSF". If this is the case, the authors should clearly state in the introduction section of the manuscript, that the iterative approach presented in this paper is required due the insufficient accuracy of the used spectral calibration equipment and is therefore the novelty of the described procedure. However a carefully design of the calibration stimulus should be able to overcome this problem and should allow a direct high resolution measurement of the ISRF for each detector pixel. Nevertheless, construction of the ISSF from such measurements could be tricky, as in addition, detector issues, as for instance differences in pixel to pixel cross-talk, PRNU etc. needed to be considered.

**Adjusted.** A description and rationale has been added to the introduction and details are presented in the section "Calibration measurements". The calibration measurements had to be performed within a given time slot, which did not allow to perform the measurements with a very accurate wavelength meter and wait for a stable laser at a given wavelength. Instead, the laser slowly scanned during data taking and each frame is treated as the measurement of an ISSF per row, where the column index of the fitted center is used as a wavelength label for the row data. The line-width of the OPO laser is negligible in comparison to the instrument line-width and only smeared by approximately 1/80 px due to the scanning speed.

We are aware that small local variations are neglected/ignored. Any remaining residuals are checked by examining the results, presented in the discussion section, see page 8, lines 12–15 in the original manuscript.

GC-iii  The authors fail to justify, why for instance the Pearson VII distribution is used or why the given iterative approach is chosen. There is no comparison with other possible distributions, see for instance Beirle et al. 2017. Also the use of filter parameters is not sufficiently justified. For instance RMS filtering of ISRF fits with an RMS larger than 0.0065 is applied. Why not 0.005 or 0.008?

**Adjusted.** The iterative approach is now supported by a longer discussion of synthetic data (Section 3.2). A justification why the Pearson VII distribution was used is added to the section "Methodology". Several other possible distributions have been tested, including the super-gauss, an asymmetric version of the exponential power distribution (Beirle et al. 2017). They do not represent the TROPOMI-SWIR ISRF with enough precision. The asymmetry of the super-gauss is provided by combining separate functions for two halves of the peak, which results in a continuous curve, but a discontinuous derivative, leading to larger residuals in the peak. The tails of the super-gauss are far too steep. A justification of the rms and its threshold is added to Sect. 3.1 and Sect. 4.

GC-iv   The authors claim in the abstract and the conclusion that "The accuracy of the derived ISRF is well within the requirement for accurate trace-gas retrievals". However, the method described in this paper presents only a fit procedure able

to fit the measurements with a given accuracy. The authors lack to provide a comprehensive error budget, including effects on PRNU, detector non-linearity and other mostly detector related effects to underline that claim. The paper also lacks to provide an independent verification for that claim, see for instance Frankenberg et al. 2015 (doi:10.5194/amt-8-301-2015) for comparison.

**Partly adjusted.** We are confident that this claim is valid, based on the residuals seen and the simulations done with synthetic data. The improved results wrt the original manuscript, and the many more simulations done only underline the original conclusion. True validation will be performed in-flight by level-2 processing.

Measurement artefacts have been identified by a comparison of the irradiance, radiance and diode-laser measurements. Detector non-linearity has been explicitly addressed in Sect. 2.3. We have found no evidence for residual pixel-to-pixel effects, except for a correlation with photo-sensitivity of the SWIR detector, see discussion in Sect. 4.

GC-v   The use of the terms ISRF and ISSF as defined by the authors is confusing and not consisted with the paper of Hasekamp et al. 2016. cited for justification of the ISRF knowledge requirements. Typically the (derived) ISRF function is used to convolute a theoretical high resolution RTM spectrum to lower spectral resolution in the retrieval. The (optical) ISRF is typically defined as the spectrometer response to a uniform monochromatic stimulus and approximated as the convolution of the slit image (typically represented by a boxcar function) with the PSF (or more accurately, the LSF) if the detector properties are neglected, see for comparison also Caron et al. 2017. This definition of the ISRF function is the mirror function of the ISRF function as defined in this paper, when assuming the changes of the (optical) ISRF are smooth over the image plane.

**Not adjusted.** The mapping between object and image can be performed in two directions. The terms 'spread function' and 'response function' are often used interchangably, but within the TROPOMI project they were coupled to the two different mapping directions. In this paper, the terms are applied to the spectral behaviour of the instrument. The difference is clearly stated, using also a diagram.

The ISSF is measured and the ISRF is based on measured data, so a term like 'instrument spectral measured function' is not distinctive. Both are line-shaped, so the term 'instrument line shape' is also not distinctive. By defining the two abstract terms 'spread' and 'response', the two functions are given distinctive names, but still clearly related. For the purposes of this paper, these terms help explain the algorithm.

The ISRF as determined in this algorithm is the function used by Hasekamp et al., including the orientation (checked with the authors).

GC-vi   Section 5 needs a deeper analysis to justify statements given in this section. For instance statements as: "However, it is believed the method is sensitive enough to be used on board for long-term monitoring, being able to distinguish between changes in the real instrument ISRF and changes in the speckle pattern" needs to be justified by analysis or removed.

**Not adjusted.** We agree that deeper analysis is needed. Therefore, the analysis presented in this paper are on the on-ground measurements. In-flight monitoring of the ISRF will be addressed in a separate paper.

**1.2   Specific comments**

SC-1   Section 2, P.3 L.1-6: (calibration measurements): Is a Wavemeter and a (spectral response and linearity) calibrated monitoring detector used in the setup? Is there any other type of direct laser wavelength monitoring integrated in the setup. If yes, what is the accuracy? What is the laser linewidth?

**Adjusted.** A wavelength meter was available, which was only used by the operators to set the start wavelength of each automated wavelength scan. The relative intensity was checked by examining raw images. There was no other useful monitoring detector.

The line-width of the OPO laser is negligible in comparison to the instrument line-width and only smeared by approximately 1/80 px due to the scanning speed.

**SC-2** P.2 L.23-24: "using the wavelength assignment derived from an independent wavelength calibration measurement." State how accurate the independent wavelength measurements are, as this has impact on the accuracy of the derived ISRF shape and how is it done? Is a different setup used?

**Adjusted.** We have used a CO spectrum for the wavelength calibration, using a white light source, a CO gas cell and a molecular line list. This provides a smooth wavelength dependence over the whole detector. It is only used in the last step, when the ISRF has already been determined, to convert wavelengths in pixel units to wavelengths in nm. The absolute wavelength is not needed for this, only relative wavelength distances to the line center.

**SC-3** P.3 L.25: convolution of block distribution should be exchanged by convolution with a boxcar function in the entire manuscript.

**Adjusted.** We do no longer refer to the block distribution. However, in our opinion, the term 'boxcar function' is associated with time series and is not a distribution. Therefore, we prefer the term 'uniform distribution'.

**SC-4** P.3 L.26: The optics is "blurring" the image by the spectrometer PSF (LSF) which could be asymmetric and often has also oscillations in the wings. Using a normal distribution for approximation of the PSF/LSF is only a first order approximation.

**Adjusted** according to suggestion, see Sect. 3.

**SC-5** P.4 L.21: The measurement data is corrected for background, PRNU and stray light. Why is the data not corrected for detector non-linearity as the used MCT detector can have non-linearity in the % range over the dynamic range?

**Adjusted.** We refer now to the paper of Hoogeveen et al. 2013. Non-linearity of the SWIR detector is measured to be very small and, therefore, neglected (also in the operational processor).

**SC-6** P.5 L.2-3: Wavelength and intensity of the signal are unknown: are they really completely unknown? What is the wavelength accuracy/knowledge and stability of the used OPO?

**Adjusted.** See answers SC-1 and GC-ii.

**SC-7** P.5 L.11-12: The laser wavelength scan is not regular->why?

**Adjusted.** Added description, revised manuscript, Sect. 2. The wavelength variation involves changing the temperature of a laser crystal, which is not a linear process. It can only be made regular by using negligibly sized steps and waiting longer, but there was no time for that. It is also not needed.

**SC-8** P.5 L.17: larger than 6% -> justify 6%, why not 5% or 7%?

**Adjusted.** This value is arbitrary, and explained more in the revised manuscript, Sect. 3.1.

**SC-9** P.5 L.21-23: "It is expected that the fit parameters that define the local ISRF vary only smoothly over the surface of the detector as this is determined by the spectrometer optics. Therefore, a bivariate polynomial fitting is used to smooth and to interpolate the ISRF fit parameters". This can only be expected for the optical system in case additional detector effects are neglected. However in a real life scenario, the effective ISRF is additionally compromised by insufficiently corrected detector effects. I guess, if all detector effects could be corrected to the required level, the resulting derived ISRF would be a smooth function. Therefore this statement contradicts the claim by the authors that the accuracy of the derived ISRF is well within the requirement for accurate trace-gas retrievals. Also the statement that most of the outliers are at same wavelengths and are caused by laser artefacts / scan imperfections contradicts that claim, as laser artefacts and scan imperfections need to be considered in the total error budget of the ISRF. See also discussion on P9. L.10-13. For a better understanding it would be helpful to show ISRF cross sections measurements from compromised rows, where the fit procedure fails.

**Adjusted.** The pixel-to-pixel variations of the gain (PRNU) are only about 1 % before correction and orders of magnitude smaller after correction. Outliers (due to saturated measurements and bad laser behavior) are rejected and have no impact on the fit residuals, but not all effects from saturated measurements could be removed. The remaining high-frequency

features are ignored by the ISRF parameter smoothing, but they can be seen in the residuals, which are small. They are presented in the revised manuscript and analyzed looking at the differences between the irradiance, radiance and diode-laser measurements (Fig. 7), and at pixel-to-pixel effects in the residuals before and after smoothing (Fig. 8, 14, 15).

SC-10  P.9 L.3-7: A sketch of the radiance and irradiance calibration setup would be very helpful. Is the on-board diffuser illuminated via the integrating sphere during irradiance measurements or directly by the OPO? I would expect from the text and as a ND filter is used in front of the OPO (P3,L10), that the OPO is directly illuminating the on-board diffuser. This would result in a better SNR as more light is entering the instrument but also in more spectral structures introduced by the laser + on board diffuser combination. The authors state that the integration sphere in combination with the spinning mirror is used to avoid speckles. So please justify, why the irradiance measurement which obviously should have more spectral structures is used for the key-data. This furthermore triggers the discussion what is physically more meaningful, a smaller spread in the data as observed in figure 8 and also stated on P.9 L9-10 for the radiance measurements or a better RMS of the fit as observed for the irradiance measurements, which can for instance be caused by a larger number of fit parameters defining the degree of freedom of the fit.

**Adjusted.** A sketch of the setup has been added (Fig. 2 in the revised manuscript), the description of the setup for irradiance and radiance measurements have been separated. In both cases there is an integrating sphere between the OPO and the instrument.

For SWIR, the spectral features introduced by the on-board diffuser in irradiance data have much larger periods than the ISRF range, so they are constant per ISRF. Because they are absent in radiance data, the radiance measurements were intended to be used for the key data, but the irradiance measurements turned out to be better suited. The difference found between the irradiance and radiance ISRF is gone with our improved method; it was an artefact of the fixed tail parameter $m$.

SC-11  P.9 L.14: How is it judged that the observed difference is attributed to the non-optimal scanning of the laser? What is an optimal scan? In fact, the wavelength accuracy and intensity of the laser are imperfectly known as previously stated.

**Adjusted.** The sentence was found confusing and is rephrased.

SC-12  P.9 L.15: Is a difference of the block width between radiance and irradiance measurements of $\approx 5\%$ as shown in Fig 8c for the left side of the detector really negligible in comparison to the requirements?

**Adjusted.** See answer SC-11. The difference found between the irradiance and radiance ISRF is gone with our improved method. It was an artefact of the fixed tail parameter $m$.

SC-13  P.9 L.29: Would taking the median over an entire row not imply the assumption, that there could not be a relative ISRF change along the row?

**Adjusted.** The sentence was found confusing and is rephrased. We take the median over all data along a swath (all rows). Yes, it is correct that the ISRF shape changes across the spatial dimension. However, these measurements are intended for monitoring of ISRF stability. Details on the monitoring ISRF will be provided in a separate paper.

SC-14  P.10 L.1: The statement is contradicting to the previous statement. In fact, if the laser can be used to recalibrate the ISRF for a significant part of the detector, they can be used for trace gas retrievals.

**Adjusted.** Improved the description of the monitoring ISRF, see the revised manuscript 'Introduction' and 'In-flight monitoring of ISRF' (Sect. 5).

SC-15  P.13 Figure1(a): it should be added to the axis caption that "column" is the detector spectral direction and wavelength is the wavelength derived for each ISSF measurement by the fitting procedure.

**Not adjusted.** The wavelength is that of the incoming light from the source. That it is determined in the algorithm is not relevant in the description of the difference between ISSF and ISRF. The column index is indeed in the spectral direction, but in this context it is more important that it corresponds to detector pixels. Since a pixel distance could be both along a column or row, the word 'column' seemed slightly more informative.

SC-16   P.17 Figure 5: The figure caption should be clearer. For instance "white areas on top and bottom of the detector blocked by the slit for stray light correction and DC monitoring" or something similar.

**Adjusted.** The figure caption(s) are adjusted. The blocked edges of the SWIR detector are not useful for any correction or monitoring.

SC-17   P.22 Table 3: How meaningful is the fit of the skew parameter $s$ with $> 100\%$ error?

**Adjusted.** This was indeed confusing. The range of each parameter across the detector was listed, not the error. This table has been removed from the revised manuscript.

[revised manuscript text omitted]